# Balancing Understanding and Generation in Discrete Diffusion Models

Yue Liu [1]   Yuzhong Zhao [1]   Zheyong Xie [2]   Qixiang Ye [1]   Jianbin Jiao [1]   Yao Hu [2]   Shaosheng Cao [2]   Yunfan Liu [1]

## Abstract

In discrete generative modeling, two dominant paradigms demonstrate divergent capabilities: Masked Diffusion Language Models (MDLM) excel at semantic understanding and zero-shot generalization, whereas Uniform-noise Diffusion Language Models (UDLM) achieve strong few-step generation quality, yet neither attains balanced performance across both dimensions. To address this, we propose XDLM, which bridges the two paradigms via a stationary noise kernel. XDLM offers two key contributions: it provides (1) a principled theoretical unification of MDLM and UDLM, recovering each paradigm as a special case; and (2) an alleviated memory bottleneck enabled by an algebraic simplification of the posterior probabilities. Experiments demonstrate that XDLM advances the Pareto frontier between understanding capability and generation quality. Quantitatively, XDLM surpasses UDLM by 5.4 points on zero-shot text benchmarks and outperforms MDLM in few-step image generation (FID 54.1 vs. 80.8). When scaled to tune an 8B-parameter large language model, XDLM achieves 15.0 MBPP in just 32 steps, effectively doubling the baseline performance. Finally, analysis of training dynamics reveals XDLM's superior potential for long-term scaling. Code is available at https://github.com/MzeroMiko/XDLM.

## 1. Introduction

Diffusion models have achieved remarkable success in continuous domains, particularly in image and audio generation (Ho et al., 2020; Dhariwal & Nichol, 2021; Rombach et al., 2022; Kong et al., 2020). Inspired by this potential, Discrete Denoising Diffusion Probabilistic Models (D3PMs) have emerged as a promising paradigm shift for the discrete

state space (Austin et al., 2021). Notably, these Discrete Diffusion Models (DDMs) are now demonstrating strong performance in language modeling, a field long dominated by auto-regressive (AR) architectures.

Within this landscape, DDM studies have diverged into two distinct branches: Masked Diffusion Language Models (MDLMs) (Sahoo et al., 2024) and Uniform-noise Diffusion Language Models (UDLMs) (Schiff et al., 2024). While MDLMs achieve superior performance in likelihood modeling and zero-shot generalization, they often struggle to generate coherent, contextually consistent outputs with limited inference steps. Conversely, UDLMs excel at low-step generation but frequently lag behind MDLMs when the number of inference steps increases. Despite their individual merits, neither achieves balanced performance across these dimensions.

This performance imbalance has critical practical implications, particularly in domains like image synthesis where the required inference steps are typically far fewer than the total number of patches. In these few-step regimes, UDLM (and its hidden Gaussian counterparts (Schiff et al., 2024; Sahoo et al., 2025)) consistently outperforms the masked paradigm. Empirically, UDLM outperforms MDLM by 17.6% when generating ImageNet-1K images in an 8-step regime. Consequently, balancing the robust few-step generation of UDLMs with the superior semantic understanding of MDLMs remains a critical open challenge.

To this end, we propose miXed Diffusion Language Modeling (XDLM), a balanced theoretical formulation that bridges uniform and masked noise distributions. Unlike GIDD (von Rütte et al., 2025), which relies on a computationally expensive time-inhomogeneous process where noise characteristics shift at every timestep, XDLM enforces a stationary noise kernel where incremental noise structurally matches the marginal noise, as shown in Figure 1 (left). This consistency allows our training objective to be efficiently factorized into static constants and dynamic schedules, avoiding the complex re-computation of noise distributions required by GIDD. We theoretically prove that MDLM and UDLM are limiting cases of our formulation, and introduce a memory-efficient implementation algebraically simplifying posterior calculations, allowing XDLM to scale to large vocabulary sizes without prohibitive computational costs.

---

[1]UCAS  [2]Xiaohongshu Inc.. Correspondence to: Yunfan Liu <liuyunfan@ucas.ac.cn>, Shaosheng Cao <caoshaosheng@xiaohongshu.com>.

*Proceedings of the 43$^{rd}$ International Conference on Machine Learning*, Seoul, South Korea. PMLR 306, 2026. Copyright 2026 by the author(s).

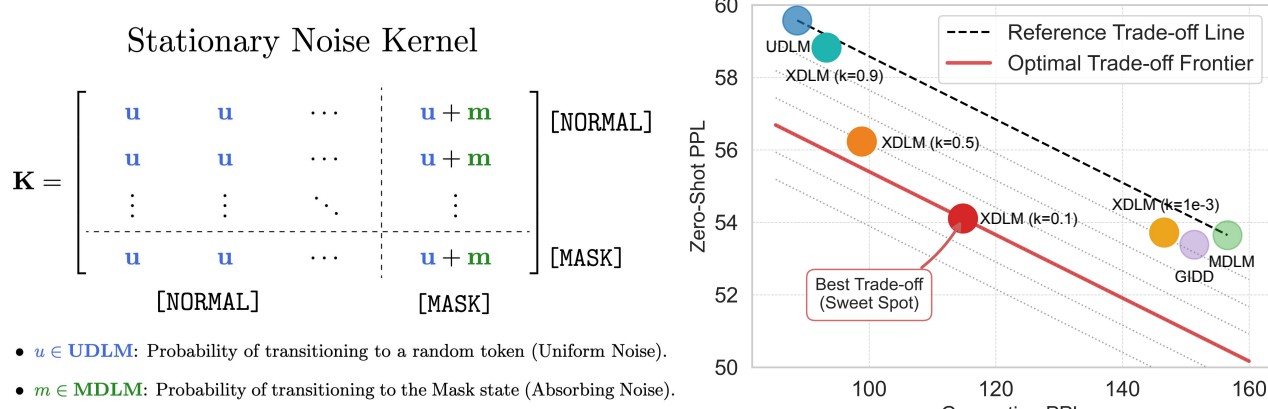

*Figure 1.* `Left`: XDLM combines the noise kernel of UDLM ($u$) and MDLM ($m$) to achieve a favorable trade-off between the two methods. `[NORMAL]` denotes normal tokens, while `[MASK]` represents the mask token. `Right`: The trade-off between understanding capability (zero-shot perplexity; lower is better) and generation capability (generation perplexity in 32 sampling steps; lower is better). The proposed XDLM with a mixing ratio of $k = 0.1$ achieves the optimal balance, labeled as the 'Sweet Spot'.

Experimental results demonstrate that XDLM consistently achieves superior performance across diverse modalities, validating the efficacy of our balanced discrete diffusion approach. As illustrated in Figure 1 (right), XDLM advances the Pareto frontier between understanding capability and generative quality. Notably, as the interpolation parameters approach their respective boundaries, XDLM recovers the performance of UDLM and MDLM, empirically validating our theoretical foundation. Concretely, in zero-shot language benchmarks, XDLM surpasses UDLM by 5.4 points in averaged metrics and trails MDLM by only 0.45 points. In conditional image generation, XDLM excels in both efficiency and quality: it significantly outperforms MDLM in few-step generation (reducing FID from 80.8 to 54.1 at 4 steps) and ranks first at 16 steps, surpassing UDLM by 0.4 points. Furthermore, large-scale continual pretraining on 8B-parameter LLMs (Nie et al., 2026) yields an MBPP score of 15.0 in only 32 sampling steps, improving upon the vanilla LLaDA baseline by over 120% (15.0 vs. 6.8). Finally, analysis of training dynamics reveals a distinct performance crossover: while masked baselines converge quickly but plateau early, XDLM sustains steady improvement throughout training, demonstrating superior long-term scaling.

## 2. Preliminary

Discrete Diffusion Models (DDMs) are latent generative models defined by two Markov processes: a *forward* process and a *reverse* process. We consider a single categorical variable $\mathbf{x}_0$, represented as a one-hot vector in $\{0, 1\}^N$, where $N = |\mathcal{V}|$ denotes the vocabulary size. Let $\mathbf{z}_t$ represent the latent state at time $t \in [0, 1]$, with $\mathbf{z}_0 := \mathbf{x}_0$. The forward process $q$ progressively corrupts the data via

transition kernels $q(\mathbf{z}_t \mid \mathbf{z}_s)$ for $s < t$, and the backward process $p_\theta(\mathbf{z}_s \mid \mathbf{z}_t)$ parameterized by $\theta$ iteratively denoises the latent states to recover the clean data $\mathbf{x}_0$.

**Forward Process** The forward transition probabilities are governed by row-stochastic matrix $\mathbf{Q}_{t|s} \in \mathbb{R}^{N \times N}$:

$$q(\mathbf{z}_t \mid \mathbf{z}_s) = \text{Cat}(\mathbf{z}_t; \mathbf{Q}_{t|s}^\top \mathbf{z}_s), \qquad (1)$$

where $\mathbf{z}_t$ and $\mathbf{z}_s$ are one-hot vectors due to the discreteness of the latent states, and $\text{Cat}(\mathbf{x}; \mathbf{p})$ is a categorical distribution. Common structural choices for $\mathbf{Q}_{t|s}$ include uniform transitions (UDLM (Schiff et al., 2024)) and absorbing-state transitions (MDLM (Sahoo et al., 2024)).

**Reverse Process** Using Bayes' rule and the Markov property, the posterior probability of $\mathbf{z}_s$ given $\mathbf{z}_t$ and $\mathbf{x}_0$ can be expressed as (Austin et al., 2021):

$$q(\mathbf{z}_s \mid \mathbf{z}_t, \mathbf{x}_0) = \frac{(\mathbf{Q}_{t|s}\mathbf{z}_t) \odot (\mathbf{Q}_{s|0}^\top \mathbf{x}_0)}{\mathbf{z}_t^\top \mathbf{Q}_{t|0}^\top \mathbf{x}_0}. \qquad (2)$$

In the literature, this reverse transition is approximated via $\mathbf{x}_0$-parameterization (Austin et al., 2021; Sahoo et al., 2024; Schiff et al., 2024), where a neural network $f_\theta$ is adopted to predict the distribution of clean data $\tilde{\mathbf{x}}_0 = f_\theta(\mathbf{z}_t)$. Thus, the parameterized reverse transition can be written as $p_\theta(\mathbf{z}_s \mid \mathbf{z}_t) = q(\mathbf{z}_s \mid \mathbf{z}_t, \tilde{\mathbf{x}}_0)$.

**Training Objective** The variational lower bound (VLB, ELBO) is optimized during training, which can be decomposed into three components: prior, diffusion, and reconstruction terms. Formally, by defining $s = (i - 1)/T$ and

$t = i/T$, the objective $\mathcal{L}_{\text{vb}}$ can be formulated as:

$$\mathcal{L}_{\text{vb}} = \mathbb{E}_{q(\mathbf{x}_0)} \Bigg[ \underbrace{D_{\text{KL}}(q(\mathbf{z}_1 \mid \mathbf{x}_0) \parallel p(\mathbf{z}_1))}_{\mathcal{L}_{\text{prior}}}$$

$$+ \sum_{i=2}^{T} \underbrace{\mathbb{E}_{q(\mathbf{z}_t \mid \mathbf{x}_0)}[D_{\text{KL}}(q(\mathbf{z}_s \mid \mathbf{z}_t, \mathbf{x}_0) \parallel p_\theta(\mathbf{z}_s \mid \mathbf{z}_t))]}_{\mathcal{L}_{\text{diffusion}}}$$

$$- \underbrace{\mathbb{E}_{q(\mathbf{z}_{1/T} \mid \mathbf{x}_0)}[\log p_\theta(\mathbf{x}_0 \mid \mathbf{z}_{1/T})]}_{\mathcal{L}_{\text{recons}}} \Bigg]. \quad (3)$$

In the continuous time limit ($T \to \infty$), the prior and reconstruction terms become negligible, simplifying the objective to the diffusion term alone (Schiff et al., 2024):

$$\mathcal{L}_{\text{vb}} = \mathbb{E}_{t \sim \mathcal{U}(0,1)} \left[ T D_{\text{KL}}(q(\mathbf{z}_s \mid \mathbf{z}_t, \mathbf{x}_0) \parallel p_\theta(\mathbf{z}_s \mid \mathbf{z}_t)) \right]. \quad (4)$$

# 3. miXed Diffusion Language Modeling via Stationary Noise Kernels

This section details the proposed miXed Diffusion Language Modeling (XDLM) framework. It begins by introducing the forward process with a stationary noise kernel in Section 3.1, and then derives a scalar formulation for efficient sampling and training in Section 3.2. Finally, Section 3.3 establishes that UDLM and MDLM can be viewed as limiting cases of XDLM under particular settings of the hybrid parameter.

## 3.1. The Forward Process with Stationary Kernels

The forward diffusion process of DDMs from time $s$ to $t$ is governed by a transition matrix $\mathbf{Q}_{t|s}$, defined as a convex combination of the signal-preserving identity matrix $\mathbf{I}$ and a stationary noise kernel $\mathbf{K}$:

$$\mathbf{Q}_{t|s} = \alpha_{t|s} \mathbf{I} + \beta_{t|s} \mathbf{K}, \quad (5)$$

where $\alpha_{t|s}, \beta_{t|s} \in [0, 1]$ represent a scalar schedule satisfying $\alpha_{t|s} + \beta_{t|s} = 1$.

A critical design choice in our method is the stationarity of $\mathbf{K}$. Unlike GIDD (von Rütte et al., 2025), which utilizes time-dependent mixing distributions that inherently couple the noise structure with the diffusion timeline, we enforce $\mathbf{K}$ to remain invariant across all time steps. This stationarity decouples the noise characteristics from the scheduling dynamics, ensuring that the diffusion process follows a geometrically consistent trajectory toward a fixed target distribution, which aligns with the optimal transport path proposed in the Flow Matching (Lipman et al., 2023).

Under this constraint, the noise kernel $\mathbf{K}$ must map every input state toward a target noise distribution $\pi$, regardless of the starting state. This requirement is formalized next.

**Lemma 3.1** (Construction of Mixing Kernel). *Let $\pi \in \mathbb{R}^N$ be a stationary target distribution defined as a mixture of the uniform distribution $\mathbf{u}$ and point-masses on special tokens $i \in \mathcal{S}$. If the noise kernel $\mathbf{K} \in \mathbb{R}^{N \times N}$ satisfies the property of instantaneous mixing (i.e., convergence to $\pi$ in a single step), it admits the decomposition:*

$$\mathbf{K} = \frac{k}{N} \mathbf{J} + \sum_{i \in \mathcal{S}} \mu_i \mathbf{M}_i, \quad (6)$$

*where $\mathbf{J} \in \mathbb{R}^{N \times N}$ is the all-ones matrix, $\mathbf{M}_i \in \mathbb{R}^{N \times N}$ is the absorbing matrix for state $i$, and the weights satisfy $k + \sum_{i \in \mathcal{S}} \mu_i = 1$.*

We now specialize Lemma 3.1 (derivation in Appendix A) to a standard MDLM setting with a single mask token, denoted by the one-hot vector $\mathbf{m}$. By assigning $k$ and $\mu$ as the weights of the uniform and masking components, respectively, the resulting noise kernel matrix naturally interpolates between uniform noise and the mask state as introduced by (Austin et al., 2021):

$$\mathbf{K} = \frac{k}{N} \mathbf{J} + \mu \mathbf{M}. \quad (7)$$

Figure 1 illustrates the corresponding transition dynamics within the noise kernel $\mathbf{K}$ driven by uniform noise (labeled $\mathbf{u}$) and absorbing noise (mask state, labeled $\mathbf{m}$).

## 3.2. Efficient Sampling and Training via Scalar Formulation

By substituting the definitions of the noise kernel $\mathbf{K}$ (Eq. 6) and transition matrix $\mathbf{Q}_{t|s}$ (Eq. 5) into Eq. 2 and Eq. 4, the explicit formulations for the posterior $q(\mathbf{z}_s \mid \mathbf{z}_t, \mathbf{x})$ and the corresponding KL divergence can be derived. However, direct evaluation of these expressions via matrix operations is prohibitively expensive in terms of both memory and computation, particularly for large vocabularies. In this section, we demonstrate that both the posterior and KL divergence can be reformulated into an equivalent scalar expressions (Lemma 3.3 and Lemma 3.4). By further incorporating the continuous-time asymptotic limit (Lemma 3.5), we establish a tractable and numerically stable training objective that circumvents explicit computation of large vocabulary size.

Concretely, letting $\mathbf{e}$ denote an arbitrary one-hot basis vector, the posterior probability of transitioning to state $\mathbf{e}$ can be expressed as

$$q(\mathbf{z}_s = \mathbf{e} \mid \mathbf{z}_t, \mathbf{x})$$
$$= \mathbf{e}^\top \frac{(\alpha_{t|s} \mathbf{z}_t + \beta_{t|s} \mathbf{K} \mathbf{z}_t) \odot (\alpha_s \mathbf{x} + \beta_s \mathbf{K}^\top \mathbf{x})}{\mathbf{z}_t^\top (\alpha_t \mathbf{x} + \beta_t \mathbf{K}^\top \mathbf{x})}, \quad (8)$$

where we observe that Eq. 8 can be reformulated into an equivalent scalar form by defining a set of helper primitives,

which can significantly reduce the computational complexity for both the posterior and the KL divergence.

**Definition 3.2** (Scalar Primitives). Let $p_{\mathbf{x},\mathbf{e}} := \mathbf{e}^\top \mathbf{x}$ represent the probability mass of distribution $\mathbf{x}$ at token $\mathbf{e}$. We define the noise rate function $r(\mathbf{e})$ and the forward diffusion map $f_t(\mathbf{x}, \mathbf{e})$ as:

$$r(\mathbf{e}) = \frac{k}{N} + \mu \delta_{\mathbf{e},\mathbf{m}} \tag{9}$$

$$f_t(\mathbf{x}, \mathbf{e}) = \alpha_t p_{\mathbf{x},\mathbf{e}} + \beta_t r(\mathbf{e}) \tag{10}$$

These helper functions yield the following efficient expressions for the posterior and KL divergence, with full derivations deferred to Appendix B.

**Lemma 3.3** (Scalar Posterior). *The posterior probability of transitioning to state* $\mathbf{e}$ *can be formulated as:*

$$q(\mathbf{z}_s = \mathbf{e} \mid \mathbf{z}_t, \mathbf{x}) = \frac{f_s(\mathbf{x}, \mathbf{e}) f_{t|s}(\mathbf{e}, \mathbf{z}_t)}{f_t(\mathbf{x}, \mathbf{z}_t)}. \tag{11}$$

*The model's reverse transition* $p_\theta$ *follows the same form by substituting* $\mathbf{x}$ *with the predicted distribution* $\tilde{\mathbf{x}}_0$.

**Lemma 3.4** (Scalar KL Divergence). *The term* $D_{KL}(q(\mathbf{z}_s \mid \mathbf{z}_t, \mathbf{x}) \parallel p_\theta(\mathbf{z}_s \mid \mathbf{z}_t))$ *can be written as:*

$$D_{KL} = \frac{\beta_{t|s} \alpha_s r(\mathbf{z}_t)}{f_t(\mathbf{x}, \mathbf{z}_t)} h_t(\mathbf{x}, \mathbf{z}_t, \tilde{\mathbf{x}}_0), \tag{12}$$

*where* $h_t$ *is an auxiliary function collecting the logarithmic difference terms:*

$$h_t(\mathbf{x}, \mathbf{z}_t, \tilde{\mathbf{x}}_0) = \frac{f_s(\mathbf{x}, \mathbf{z}_t)}{r(\mathbf{z}_t)} \frac{\alpha_{t|s}}{\beta_{t|s} \alpha_s} \log \frac{f_s(\mathbf{x}, \mathbf{z}_t) f_t(\tilde{\mathbf{x}}_0, \mathbf{z}_t)}{f_t(\mathbf{x}, \mathbf{z}_t) f_s(\tilde{\mathbf{x}}_0, \mathbf{z}_t)}$$
$$- \frac{1}{\alpha_s} \log \frac{f_t(\mathbf{x}, \mathbf{z}_t)}{f_t(\tilde{\mathbf{x}}_0, \mathbf{z}_t)} + \log \frac{f_s(\mathbf{x}, \mathbf{x})}{f_s(\tilde{\mathbf{x}}_0, \mathbf{x})}$$
$$+ \frac{k\beta_s}{N\alpha_s} \sum_{\mathbf{e} \in \mathcal{V}} \log \frac{f_s(\mathbf{x}, \mathbf{e})}{f_s(\tilde{\mathbf{x}}_0, \mathbf{e})}. \tag{13}$$

As for the limiting case where $s \to t$, approximating continuous time, the auxiliary function $h_t$ simplifies, thereby circumventing the numerical instability associated with the first logarithmic term in Eq. 13.

**Lemma 3.5** (Limiting Case). *As* $s \to t$, *the function* $h_t$ *converges to:*

$$h_t(\mathbf{x}, \mathbf{z}_t, \tilde{\mathbf{x}}_0) = \frac{p_{\mathbf{x},\mathbf{z}_t} - p_{\tilde{\mathbf{x}}_0,\mathbf{z}_t}}{f_t(\tilde{\mathbf{x}}_0, \mathbf{z}_t)} - \frac{1}{\alpha_t} \log \frac{f_t(\mathbf{x}, \mathbf{z}_t)}{f_t(\tilde{\mathbf{x}}_0, \mathbf{z}_t)}$$
$$+ \log \frac{f_t(\mathbf{x}, \mathbf{x})}{f_t(\tilde{\mathbf{x}}_0, \mathbf{x})} + \frac{k\beta_t}{N\alpha_t} \sum_{\mathbf{e} \in \mathcal{V}} \log \frac{f_t(\mathbf{x}, \mathbf{e})}{f_t(\tilde{\mathbf{x}}_0, \mathbf{e})}. \tag{14}$$

Consequently, by substituting the asymptotic behavior derived in Lemma 3.4 and 3.5 into the training objective defined in Eq. 4, we arrive at a tractable scalar formulation for

the training loss:

$$\lim_{T \to \infty} \mathcal{L}_{\text{vb}} = \mathbb{E}_{t,\mathbf{x}_0,\mathbf{z}_t} \left[ \frac{-\alpha_t' r(\mathbf{z}_t)}{f_t(\mathbf{x}, \mathbf{z}_t)} \cdot h_t(\mathbf{x}, \mathbf{z}_t, \tilde{\mathbf{x}}_0) \right]. \tag{15}$$

This substitution bypasses the expensive explicit computation of posterior distributions and their KL divergence, facilitating an efficient and stable implementation for XDLM training and sampling.

The core pseudocode is provided in Appendix D.

### 3.3. Relationship to MDLM and UDLM

The proposed XDLM establishes a generalized framework for discrete diffusion. By modulating the uniform noise weight $k$ (Eq. 7) and consequently adjusting the absorbing strength, XDLM seamlessly reduces to MDLM and UDLM at its parameter limits. The relationship between XDLM and these models is outlined in this section, with a detailed analysis deferred to Appendix C.

**Connection with MDLM.** Setting $k = 0$ (which implies $\mu = 1$) reduces the noise kernel to a pure masking operation. Thus, the posterior probability simplifies to:

$$p_\theta(\mathbf{z}_s = \mathbf{e} \mid \mathbf{z}_t)$$
$$= \delta_{\mathbf{z}_t,\mathbf{m}} \delta_{\mathbf{e},\mathbf{m}} \frac{\beta_s}{\beta_t} + \delta_{\mathbf{z}_t,\mathbf{m}} \delta_{\mathbf{e} \neq \mathbf{m}} \frac{\beta_{t|s} \alpha_s p_{\theta,\mathbf{e}}}{\beta_t} + \delta_{\mathbf{z}_t \neq \mathbf{m}} \delta_{\mathbf{e},\mathbf{z}_t}$$
$$= \begin{cases} \frac{\beta_s}{\beta_t} \delta_{\mathbf{m},\mathbf{z}_t}, & \mathbf{e} = \mathbf{m} \\ \frac{(\alpha_s - \alpha_t)}{1 - \alpha_t} p_{\theta,\mathbf{e}}, & \mathbf{z}_t = \mathbf{m}, \mathbf{e} \neq \mathbf{m} \\ \delta_{\mathbf{e},\mathbf{z}_t}, & \mathbf{z}_t \neq \mathbf{m}, \mathbf{e} \neq \mathbf{m} \end{cases} \tag{16}$$

Consequently, the KL divergence collapses to the standard cross-entropy loss on masked tokens, matching the MDLM objective (Sahoo et al., 2024):

$$D_{\text{KL}} = -\frac{\beta_{t|s} \alpha_s}{\beta_t} \log p_\theta(\mathbf{x} \mid \mathbf{z}_t). \tag{17}$$

**Connection with UDLM.** Setting $k = 1$ (implying $\mu = 0$) yields the uniform noise kernel. In this limit, the posterior probability becomes:

$$p_\theta(\mathbf{z}_s = \mathbf{e} \mid \mathbf{z}_t)$$
$$= \frac{\delta_{\mathbf{e},\mathbf{z}_t}(N\alpha_t p_{\theta,\mathbf{e}} + \beta_s \alpha_{t|s}) + (\alpha_s - \alpha_t) p_{\theta,\mathbf{e}} + \beta_s \beta_{t|s}/N}{N\alpha_t p_{\theta,\mathbf{z}_t} + \beta_t} \tag{18}$$

By defining $\bar{x}_j := N f_t(\mathbf{x}, \mathbf{e}_j)$ and $(\bar{x}_\theta)_j := N f_t(\tilde{\mathbf{x}}_0, \mathbf{e}_j)$, and taking the continuous-time limit $s \to t$, our derived KL divergence aligns identically with the UDLM loss from in (Schiff et al., 2024):

$$D_{\text{KL}} = \frac{-\beta_{t|s}\alpha_s}{N\alpha_t} \left( \frac{N}{(\bar{x}_\theta)_i} - \frac{N}{\bar{x}_i} - \sum_{j:(\mathbf{z}_t)_j=0} \frac{\bar{x}_j}{\bar{x}_i} \log \frac{(\bar{x}_\theta)_i \bar{x}_j}{(\bar{x}_\theta)_j \bar{x}_i} \right)$$
(19)

where $i$ is the index of the active state in $\mathbf{z}_t$.

## 4. Experiment

The experimental setup is first introduced in Section 4.1, with additional details provided in Appendix G. Section 4.2 then reports the performance of XDLM on zero-shot likelihood, language generation quality, and image generation tasks, demonstrating its superiority over existing methods (extended results are available in Appendices H, I and J) . To evaluate scalability to larger model sizes, XDLM is further extended to LLaDA-XDLM via continual pretraining on LLaDA, which achieves strong performance on standard benchmarks as detailed in Appendix K. Finally, Section 4.3 presents comprehensive analysis and visualizations across different tasks, showing that XDLM achieves a better balance between understanding and generation than other approaches (supplementary speed tests, training dynamics, and qualitative samples are compiled in Appendices L, M, E and F).

### 4.1. Experimental Setup

For all core configurations, models were trained on a cluster of 8×H800 GPUs. We standardized key hyperparameters across baselines, utilizing the AdamW optimizer ($\beta_1 = 0.9, \beta_2 = 0.999$, weight decay 0), a mixing ratio of $k = 0.1$, a global batch size of 512, and an EMA rate of 0.9999.

Our experimental codebase is built upon the Duo framework (Sahoo et al., 2025), which provides the core implementations for MDLM and UDLM, with the latter utilizing the Rao–Blackwellized NELBO formulation from (Sahoo et al., 2025). For a rigorous and fair comparison, XDLM is integrated into this identical framework. Meanwhile, the GIDD baseline (von Rütte et al., 2025) is integrated into our codebase adhering strictly to its official specifications while preserving its default configurations.

To ensure architectural consistency, we omit the log-linear time schedule input for MDLM, UDLM, and XDLM, while keeping the AdaLN branch stationary by fixing its conditioning input to zero. For GIDD, we follow the configuration described above. Complete implementation details and task-specific configurations are provided in Appendix G.

### 4.2. Performance

**Evaluation of Zero-Shot Likelihood**. To assess the generalization capabilities of XDLM, we evaluate zero-shot

*Table 1.* Validation PPL on the OWT dataset. XDLM achieves performance comparable to MDLM and outperforms UDLM under the same training steps.

| Model | MDLM | GIDD | XDLM | UDLM |
|-------|------|------|------|------|
| **PPL** | 23.321 | 23.136 | 24.097 | 25.937 |

perplexity (PPL) on seven external datasets after training on OpenWebText (OWT). Following prior work (Sahoo et al., 2024; 2025), our evaluation suite includes the validation sets of AG News (Zhang et al., 2015), LAMBADA (Paperno et al., 2016), LM1B (Chelba et al., 2013), Penn Treebank (Marcus et al., 1993), Scientific Papers from ArXiv and PubMed (Cohan et al., 2018), and WikiText (Merity et al., 2017). The validation PPL is estimated via the negative Evidence Lower Bound (ELBO) with Monte Carlo sampling. To ensure a fair comparison, we adopt the identical sampling configuration in (Sahoo et al., 2024; 2025).

As shown in Table 1, XDLM achieves a PPL of 24.097 on the OWT validation set, comparable to MDLM and GIDD and significantly outperforming UDLM (25.937). This retention of in-domain capability extends to zero-shot benchmarks. As summarized in Table 2, XDLM obtains an average PPL of 54.110 across seven datasets, performing similarly to MDLM (53.650) and GIDD (53.384) while UDLM clearly lags behind (59.574).

**Language Generation Quality.** We evaluate performance using perplexity (quality) and token entropy (diversity) under *ancestral* sampling (Sahoo et al., 2024; Schiff et al., 2024; Sahoo et al., 2025). Figure 2 shows that XDLM strikes a robust balance between masked and uniform diffusion paradigms. In few–step regimes (8—32 steps on OWT), XDLM benefits from uniform noise dynamics, achieving generation quality comparable to UDLMs and clearly surpassing purely masked models. In contrast, under multi-step regimes (512-–1024 steps), masked noise becomes dominant, allowing XDLM to outperform UDLM and reach MDLM-level performance. A similar trend is observed on LM1B, confirming the efficacy of XDLM in securing the efficiency of uniform noise while preserving the rigorous structure of masking. Comprehensive numerical results analyzing the impact of the mixing ratio $k$ are provided in Appendix I.

**Image Generation Performance**. We evaluate domain generality via class-conditional generation on ImageNet-1K. Table 3 reports FID and IS scores without Classifier-Free Guidance (CFG). In few-step regimes, XDLM and UDLM consistently outperform mask-based baselines (*i.e.*, MDLM and GIDD). XDLM proves highly competitive with UDLM, achieving the lowest FID (25.77) at 16 steps, demonstrating its capacity for efficient, high-quality generation. When evaluating with CFG scale of 2.0, as shown in Table 4, the

*Table 2.* Zero-shot performance comparison across various benchmarks. XDLM maintains performance parity with MDLM and GIDD, significantly outperforming UDLM. Models were reimplemented under a unified setting using 8×H800 GPUs. The best results for each benchmark are highlighted in bold.

| Dataset | AG News | LAMBADA | LM1B-GPT2 | PTB | ArXiv | PubMed | WikiText | Average |
|---------|---------|---------|-----------|-----|-------|--------|----------|---------|
| MDLM | 61.374 | 47.967 | **65.629** | 89.049 | 37.457 | 41.981 | 32.093 | 53.650 |
| GIDD | **60.607** | 47.811 | 65.898 | **86.911** | 39.019 | 42.634 | **30.809** | 53.384 |
| XDLM | 62.768 | **45.608** | 68.229 | 90.796 | **37.232** | **41.391** | 32.748 | 54.110 |
| UDLM | 69.402 | 51.272 | 75.572 | 95.986 | 42.671 | 47.181 | 34.933 | 59.574 |

*Table 3.* Performance comparison of image generation on ImageNet-1K with standard conditioning. The best results are bolded.

| Model | FID ↓ | | | IS ↑ | | |
|-------|--------|--------|---------|--------|--------|---------|
| | step 4 | step 8 | step 16 | step 4 | step 8 | step 16 |
| MDLM | 80.752 | 47.732 | 28.785 | 16.287 | 29.178 | **44.656** |
| GIDD | 86.842 | 54.933 | 35.403 | 14.559 | 24.297 | 35.698 |
| XDLM | 54.085 | 34.109 | **25.774** | 24.829 | 36.964 | 43.903 |
| UDLM | **49.861** | **30.144** | 26.242 | **27.049** | **38.832** | 41.801 |

*Table 4.* Performance comparison of image generation on ImageNet-1K with CFG *scale* = 2.0. The best results are bolded.

| Model | FID ↓ | | | IS ↑ | | |
|-------|--------|--------|---------|---------|---------|---------|
| | step 4 | step 8 | step 16 | step 4 | step 8 | step 16 |
| MDLM | 33.468 | 11.144 | **6.725** | 54.740 | 119.150 | **172.664** |
| GIDD | 41.000 | 15.151 | 7.076 | 44.084 | 95.789 | 148.148 |
| XDLM | **13.550** | **8.956** | 8.625 | **107.403** | **148.723** | 165.916 |
| UDLM | 14.055 | 9.718 | 8.980 | 97.859 | 123.582 | 132.099 |

pure mask-based MDLM achieves the lowest overall FID (6.73) at 16 steps, suggesting that masking excels given sufficient steps. However, by partially inheriting these absorbing noise properties, XDLM significantly outperforms UDLM in the few-step regime, achieving the best FID scores at 4 steps (13.55) and 8 steps (8.96). Extended analysis is provided in Appendix J.

**Continual Pretraining of Large Language Models (LLMs).** To validate XDLM as a scalable upgrade for state-of-the-art LLMs, we adapt LLaDA (Nie et al., 2026), a pretrained 8B-MDLM, into the XDLM formulation with a compute-efficient pretraining of 600 steps (termed LLaDA-XDLM). To rigorously isolate the source of improvement, we compare against two controls: LLaDA-MDLM (standard continual pretraining for the same duration) and LLaDA-XDLM-infer (LLaDA coupled with our sampling strategy).

Evaluations on OpenCompass (32 steps) show that LLaDA-XDLM consistently outperforms the original LLaDA and both controls on GSM8k, MATH, and BBH (Figure 3 (a)). The impact is most visible in MBPP code generation, where LLaDA-XDLM doubles the performance of LLaDA (15.0 vs. 6.8) by minimizing non-compilable errors, thereby con-

firming enhanced structural coherence.

As for the control models, while LLaDA-MDLM suffers a performance drop compared to the original LLaDA, likely due to a distribution shift between the tuning and pretraining corpora, LLaDA-XDLM achieves significant improvements under identical conditions. This confirms that the gains stem from the efficacy of the XDLM formulation rather than the additional 600 training steps alone. Furthermore, LLaDA-XDLM remarkably outperforms LLaDA-XDLM-infer, demonstrating that the improvement is intrinsic to the learned model rather than an artifact of sampling strategies.

### 4.3. Analysis

**Performance Crossover in Training Dynamics.** Analysis of the training dynamics reveals a 'performance crossover' phenomenon, where the relative efficacy of competing models shifts significantly as training progresses. This trend is particularly evident in the LM1B text generation task, as shown by the Generation PPL curves in Figure 4 (a). While GIDD and MDLM demonstrate advantages in the initial phase ($<$200k steps, pink region), they suffer from rapid saturation and are eventually outperformed by UDLM (yellow region). Leveraging the uniform nature inherited from UDLM, XDLM demonstrates substantial late-stage improvement, eventually matching or surpassing MDLM after 700k steps (green region). This trajectory is in sharp contrast to GIDD, which exhibits early saturation and only marginal PPL gains throughout the training process.

A consistent trajectory manifests in class-conditional image generation on ImageNet-1K (Figure 4 (b)). Here, XDLM demonstrates superior performance from the start, consistently achieving the lowest FID scores. While UDLM eventually converges with XDLM in the final training phase ($>$ 500k steps), both models significantly outperform the MDLM and GIDD baselines throughout the process.

**Impact of Mixing Ratio.** The mixing ratio $k$ serves as a pivotal control parameter governing the interpolation between two distinct paradigms: UDLM ($k \to 1$) and MDLM ($k \to 0$), characterized by their divergent capabilities in generation and understanding. As shown in Figure 1 (right), these boundary cases establish a linear *Reference Trade-off*

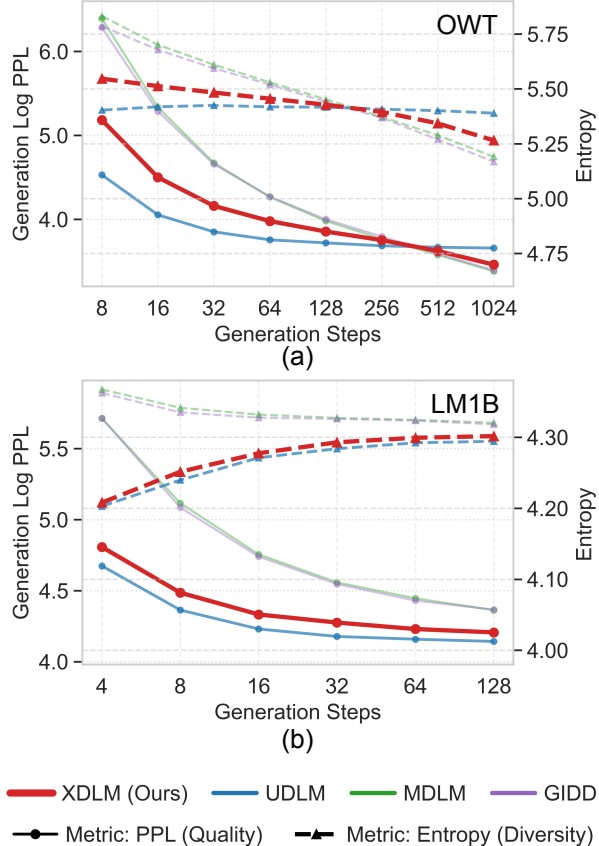

*Figure 2.* Language Generation Quality. Results on OWT (top) and LM1B (bottom) demonstrate that XDLM achieves a better balance across both few-step and multi-step regimes. For clarity, PPL and Generation Steps are reported in logarithmic scale.

*Line* (dashed black line), representing a zero-sum scenario where gains in generation quality (lower Generation PPL) come at the cost of understanding capability (higher Zero-Shot PPL).

However, the empirical behavior of XDLM reveals that the model does not merely traverse this linear path. Crucially, varying $k$ shifts the model's operating point along a continuous, smooth trajectory. This coherent evolution, as opposed to stochastic fluctuations or random walks, validates the architectural design, confirming that XDLM successfully synthesizes the mechanical properties of both UDLM and MDLM into a unified framework. Moreover, this trajectory bends concavely to form a superior *Optimal Trade-off Frontier* (solid red line), demonstrating a synergistic effect where the mixed objective yields performance gains that exceed the sum of the parts. By navigating this frontier, we identify an empirical 'sweet spot' at $k = 0.1$. As shown by the red marker, this configuration effectively breaks the Pareto frontier of the baselines, achieving robust understanding capabilities without the severe degradation in generation

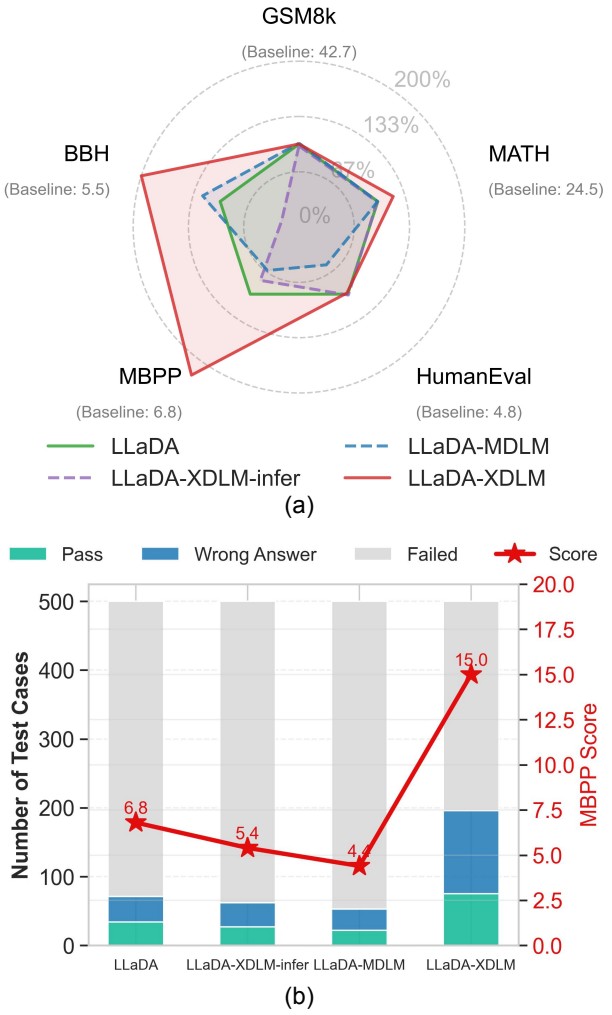

*Figure 3.* Evaluation of adapting LLaDA-8B to our XDLM formulation (LLaDA-XDLM): (a) LLaDA-XDLM consistently outperforms baselines across diverse benchmarks; (b) Improvements are particularly pronounced in code generation (MBPP), where the model substantially reduces generation failures.

performance typical of pure masked models.

**Computational Efficiency of XDLM.** We analyze the computational efficiency across inference, training, and sampling. While MDLM remains the fastest baseline due to its simple absorbing kernel, XDLM emerges as the most efficient model among those involving uniform noise. Benefiting from our scalar reformulation strategy, XDLM achieves a forward throughput of 396,398 tokens/s, nearly double GIDD's 199,516 tokens/s, and a training throughput of 137,372 tokens/s. In generation tasks, XDLM maintains a sampling speed of 7,108 tokens/s, significantly surpassing UDLM (2,882 tokens/s). This efficiency extends to memory consumption, where XDLM requires only 31.4 GB compared to UDLM's 59.7 GB and GIDD's 40.9 GB. These results confirm that XDLM effectively mitigates large-

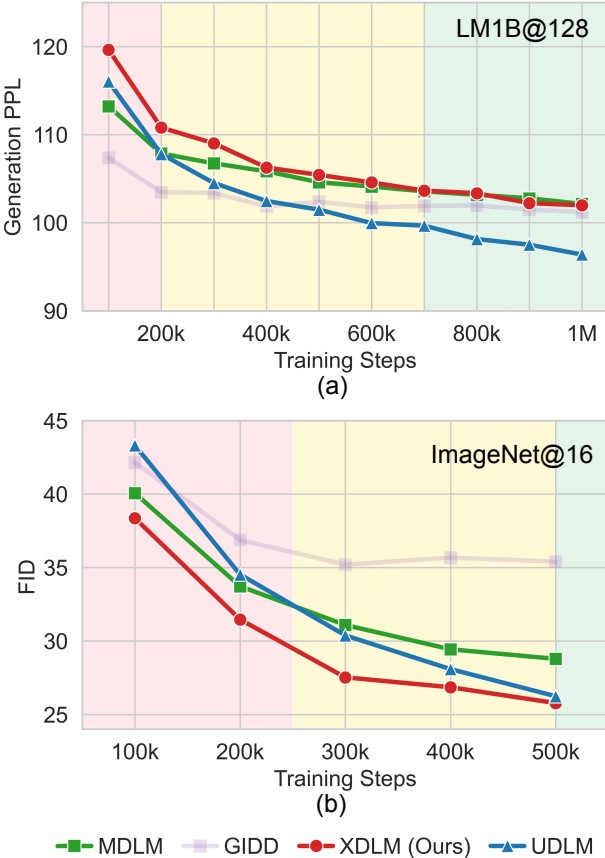

*Figure 4.* Training dynamics for (a) text and (b) image generation tasks. Colored regions indicate the dominant model during each phase, while transitions between colors mark points of performance crossover.

vocabulary bottlenecks without compromising the expressivity of complex noise kernels. For more detailed results, please refer to Appendix L.

**Inference Dynamics.** To investigate the internal generation mechanism of XDLM, we closely inspect a single generation trajectory over a budget of $T = 32$ steps. The result shows that XDLM effectively leverages a hybrid noise process: it constructs semantic structure via absorbing noise ([MASK] → Token) while refining content through uniform noise (Token → Token). Crucially, XDLM distinguishes itself with a re-masking strategy (transforming tokens back to [MASK]). Unlike standard iterative models that exhibit high sensitivity to initialization noise, this mechanism allows XDLM to actively reject low-probability tokens derived from uniform noise, effectively escaping local optima. For a detailed analysis, please refer to Appendices E and F.

**LLM-as-Judge Evaluation.** Following GIDD (von Rütte et al., 2025), we further assess unconditional generation quality on OWT using an LLM-as-judge protocol. We gen-

erate 5,000 samples with 128 decoding steps and use the GPT-5.2 API as the evaluator. Each sample is scored along five dimensions: clarity, grammaticality, factuality, writing style, and creativity, with each dimension rated on a 1–10 scale. To ensure evaluation consistency and facilitate automated parsing, the evaluator is prompted to provide a brief rationale before returning the final scores in JSON format. We use the same evaluation prompt as GIDD, and report the results in Table 5.

*Table 5.* LLM-as-judge evaluation on OWT with 128 decoding steps. Scores are averaged over 5,000 generated samples. Bold font indicates the best result in each column.

| Model | clarity | grammaticality | factuality | style | creativity |
|-------|---------|----------------|------------|-------|------------|
| MDLM | 2.02 | 2.06 | 1.93 | 2.43 | 4.24 |
| GIDD | 2.02 | 2.05 | **1.94** | 2.44 | 4.24 |
| XDLM | 2.02 | 2.10 | 1.92 | **2.48** | **4.37** |
| UDLM | 2.02 | **2.14** | 1.84 | 2.46 | 4.33 |

**Comparison with Loss Interpolation.** To distinguish XDLM from a heuristic combination of existing objectives, we compare it with a simple weighted mixture of MDLM and UDLM losses, denoted as SimpMix. Unlike Simp-Mix, which does not define a dedicated noise kernel or an explicit stationary distribution, XDLM is derived from a mathematically rigorous mixed noise process. As shown in Table 6, XDLM consistently achieves lower Generation PPL than SimpMix across all decoding budgets on LM1B, highlighting the importance of the proposed formulation beyond simple loss interpolation.

## 5. Related work

**Masked Diffusion Language Modeling**. Building on the general and flexible discrete diffusion framework introduced by D3PM (Austin et al., 2021), absorbing-state–based corruptions (often using a special [MASK] token) have emerged as a popular paradigm for discrete diffusion in language, frequently referred to as Masked Diffusion Language Models (MDLMs). MDLM (Sahoo et al., 2024) and MD4 (Shi et al., 2024) extend this framework and derive a continuous-time variational objective for masked diffusion as a simple weighted integral of cross-entropy losses, yielding a streamlined and general training recipe for discrete language models. From a continuous-time perspective, CTMC (Campbell et al., 2022) formalizes the forward process via continuous-time Markov chains, while SEDD (Lou et al., 2023) learns density ratios directly, providing a discrete analogue of score matching. These advances have facilitated scaling masked diffusion LMs to larger model sizes and datasets, leading to large diffusion language models such as LLaDA (Nie et al., 2026) and Dream (Ye et al., 2025), as well as extensions to multimodal settings (You et al., 2025; Yang et al., 2026; Yu et al., 2025).

*Table 6.* Generation PPL on LM1B for XDLM and the heuristic loss-weighting baseline SimpMix under different decoding budgets. Bold font indicates the best result in each column.

| Model | step 4 | step 8 | step 16 | step 32 | step 64 | step 128 |
|---|---|---|---|---|---|---|
| XDLM | **246.134** | **155.487** | **123.129** | **113.006** | **105.509** | **101.983** |
| SimpMix | 324.830 | 207.235 | 165.450 | 150.812 | 143.689 | 137.545 |

**Uniform-noise Diffusion Language Modeling**. The transition kernel of uniform-noise diffusion is a mixture of the identity and a uniform distribution over the vocabulary. This idea was introduced for discrete spaces in Argmax/Multinomial diffusion (Hoogeboom et al., 2021) and generalized by D3PM (Austin et al., 2021). Plaid (Gulrajani & Hashimoto, 2023) takes a notable step toward narrowing the likelihood gap between autoregressive models and diffusion-based language models. UDLM (Schiff et al., 2024) shows that, with appropriate guidance mechanisms, uniform-noise diffusion can yield stronger controllability. Duo (Sahoo et al., 2025) identifies a duality between uniform-noise diffusion and Gaussian diffusion language models, (*e.g.*, Plaid), leveraging this connection to distill a uniform-noise model from an auxiliary latent Gaussian model, while further enhancing UDLM training via a Rao–Blackwellized NELBO and a tailored curriculum learning strategy.

**Mixed Language Modeling**. D3PM (Austin et al., 2021) observes that BERT can be viewed as a one-step instance of mixed corruption with a transition matrix. However, BERT is trained with a cross-entropy objective on the masked positions only, whereas diffusion-based approaches optimize objectives integrated over a noise schedule (or time), leading to different training dynamics and inductive biases.

Recent research has further bridged and generalized these paradigms. (Fathi et al., 2025) unify autoregressive and diffusion models by assigning distinct noise schedules to individual token positions. Similarly, ReMDM (Wang et al., 2026) relaxes the strict absorbing constraint of the forward process, allowing [MASK] tokens to revert to their original states, thereby enabling plug-and-play editing capabilities.

The closest parallel to our approach is GIDD (von Rütte et al., 2025), which similarly explores the interpolation between masked and uniform-noise diffusion. Crucially, while GIDD relies on dynamically blending these mechanisms through time-dependent transition matrices, our XDLM framework enforces a stationary noise kernel. This design choice explicitly decouples the core noise characteristics from temporal trajectories, enabling the derivation of a highly efficient scalar formulation that fundamentally streamlines both training and sampling.

## 6. Conclusion

In this paper, we introduced XDLM, a unified approach that theoretically bridges the gap between Masked and Uniform-noise diffusion. By reformulating the forward process via a stationary, row-stochastic mixed transition kernel, we demonstrated that XDLM recovers existing paradigms (MDLM and UDLM) as special cases. To ensure practicality, we derived a memory-efficient implementation that reduces computational complexity, enabling training on large vocabularies.

Empirically, XDLM breaks the Pareto frontier between understanding and generation. We identified a mixing ratio of $k = 0.1$ as the optimal "sweet spot", where the model combines the robust zero-shot likelihoods of masking models with the superior sample diversity and few-step generation quality of uniform noise models. These advantages extend across domains, achieving state-of-the-art on ImageNet-1K and demonstrating significant scalability in the continual pretraining of 8B-parameter LLMs, where XDLM doubled performance on code generation benchmarks.

**Limitations** Despite these contributions, several avenues for future research remain. First, we have not yet trained XDLM from scratch at a large scale; such pre-training would likely allow for a more comprehensive exploration of the model's emergent properties. Second, we did not fully investigate the "performance crossover" phenomenon, wherein UDLM and XDLM appear to outperform MDLM in generation tasks involving large sampling steps (approaching autoregressive decoding). Third, domain-specific sampling strategies for XDLM in language modeling and image generation have not yet been optimized. Furthermore, while we confirmed that XDLM balances understanding and generation, the interaction and balance between textual and visual modalities within a single unified model remain uninvestigated. Finally, the development of post-training schemas and inference acceleration techniques for XDLM remains a subject for future work.

## Impact Statement

The primary objective of this research is to advance the technical state-of-the-art in machine learning. Consequently, it entails societal considerations parallel to those found in the general field of language and foundation models.

## Acknowledgment

This work was supported by National Natural Science Foundation of China (NSFC) under Grant No.62225208 and No.62406304.

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

## A. The Forward Process with Stationary Kernels

**Lemma 1** (Construction of Mixing Kernel). *Let $\boldsymbol{\pi} \in \mathbb{R}^N$ be a stationary target distribution defined as a mixture of the uniform distribution $\mathbf{u}$ and point-masses on special tokens $i \in \mathcal{S}$. If the noise kernel $\mathbf{K} \in \mathbb{R}^{N \times N}$ satisfies the property of instantaneous mixing (i.e., convergence to $\boldsymbol{\pi}$ in a single step), it admits the decomposition:*

$$\mathbf{K} = \frac{k}{N}\mathbf{J} + \sum_{i \in \mathcal{S}} \mu_i \mathbf{M}_i, \tag{20}$$

*where $\mathbf{J} \in \mathbb{R}^{N \times N}$ is the all-ones matrix, $\mathbf{M}_i \in \mathbb{R}^{N \times N}$ is the absorbing matrix for state $i$, and the weights satisfy $k + \sum_{i \in \mathcal{S}} \mu_i = 1$.*

*Proof.* The condition of instantaneous mixing implies that the transition probability to a target state $j$ is solely determined by the target marginal $\pi_j$, independent of the source state $s$. In terms of the stochastic matrix, this imposes row-identity: $\mathbf{K}_{sj} = \pi_j$ for all $s$. This is algebraically equivalent to the rank-1 factorization $\mathbf{K} = \mathbf{1}\boldsymbol{\pi}^\top$.

Substituting the mixture definition $\boldsymbol{\pi} = k\mathbf{u} + \sum_{i \in \mathcal{S}} \mu_i \mathbf{e}_i$ into this factorization yields:

$$\mathbf{K} = \mathbf{1}\left(k\mathbf{u} + \sum_{i \in \mathcal{S}} \mu_i \mathbf{e}_i\right)^\top \tag{21}$$

$$= k(\mathbf{1}\mathbf{u}^\top) + \sum_{i \in \mathcal{S}} \mu_i(\mathbf{1}\mathbf{e}_i^\top). \tag{22}$$

Identifying the resulting outer products with the canonical transition matrices $\mathbf{1}\mathbf{u}^\top = N^{-1}\mathbf{J}$ (uniform transition) and $\mathbf{1}\mathbf{e}_i^\top = \mathbf{M}_i$ (absorbing transition) concludes the proof. $\qquad\square$

## B. Efficient Sampling and Training via Scalar Formulation

Prior to establishing Lemma 3.3, we present an auxiliary definition for the vector-valued function $\mathbf{g}(s, t, \mathbf{x}, \mathbf{z}_t)$.

**Definition B.1.**

$$\mathbf{g}(s, t, \mathbf{x}, \mathbf{z}_t) \tag{23}$$

$$= \alpha_t \mathbf{z}_t \odot \mathbf{x} + \alpha_{t|s}\beta_s \mathbf{z}_t \odot \mathbf{K}^\top \mathbf{x} + \beta_{t|s}\alpha_s \mathbf{K}\mathbf{z}_t \odot \mathbf{x} + \beta_{t|s}\beta_s \mathbf{K}\mathbf{z}_t \odot \mathbf{K}^\top \mathbf{x} \tag{24}$$

$$= \alpha_{t|s}f_s(\mathbf{x}, \mathbf{z}_t)\mathbf{z}_t + \beta_{t|s}r(\mathbf{z}_t)\left(\alpha_s\mathbf{x} + \beta_s\left(\frac{k}{N}\mathbf{1} + \mu\mathbf{m}\right)\right) \tag{25}$$

Consequently, left-multiplying by the transpose of the one-hot vector $\mathbf{e}^\top$ yields:

$$\mathbf{e}^\top \mathbf{g}(s, t, \mathbf{x}, \mathbf{z}_t) \tag{26}$$

$$= \alpha_{t|s}f_s(\mathbf{x}, \mathbf{z}_t)\mathbf{e}^\top \mathbf{z}_t + \beta_{t|s}r(\mathbf{z}_t)\left(\alpha_s\mathbf{e}^\top\mathbf{x} + \beta_s\left(\frac{k}{N}\mathbf{e}^\top\mathbf{1} + \mu\mathbf{e}^\top\mathbf{m}\right)\right) \tag{27}$$

$$= f_s(\mathbf{x}, \mathbf{z}_t)\alpha_{t|s}\mathbf{e}^\top \mathbf{z}_t + \beta_{t|s}r(\mathbf{z}_t)\left(\alpha_s\mathbf{e}^\top\mathbf{x} + \beta_s r(\mathbf{e})\right) \tag{28}$$

$$= f_s(\mathbf{x}, \mathbf{e})\alpha_{t|s}\mathbf{e}^\top \mathbf{z}_t + \beta_{t|s}r(\mathbf{z}_t)f_s(\mathbf{x}, \mathbf{e}) \tag{29}$$

$$= f_s(\mathbf{x}, \mathbf{e})f_{t|s}(\mathbf{e}, \mathbf{z}_t) \tag{30}$$

**Lemma 2** (Scalar Posterior). *The posterior probability of transitioning to state $\mathbf{e}$ can be formulated as:*

$$q(\mathbf{z}_s = \mathbf{e} \mid \mathbf{z}_t, \mathbf{x}) = \frac{f_s(\mathbf{x}, \mathbf{e})f_{t|s}(\mathbf{e}, \mathbf{z}_t)}{f_t(\mathbf{x}, \mathbf{z}_t)}. \tag{31}$$

*The model's reverse transition $p_\theta$ follows the same form by substituting $\mathbf{x}$ with the predicted distribution $\tilde{\mathbf{x}}_0$.*

*Proof.* By expanding the transition relations, we have:

$$q(\mathbf{z}_s = \mathbf{e} \mid \mathbf{z}_t, \mathbf{x}) \tag{32}$$

$$= \mathbf{e}^\top \frac{(\mathbf{Q}_{t|s}\mathbf{z}_t) \odot (\mathbf{Q}_s^\top \mathbf{x})}{\mathbf{z}_t^\top \mathbf{Q}_t^\top \mathbf{x}} \tag{33}$$

$$= \mathbf{e}^\top \frac{(\alpha_{t|s}\mathbf{z}_t + \beta_{t|s}\mathbf{K}\mathbf{z}_t) \odot (\alpha_s\mathbf{x} + \beta_s\mathbf{K}^\top\mathbf{x})}{\mathbf{z}_t^\top(\alpha_t\mathbf{x} + \beta_t\mathbf{K}^\top\mathbf{x})} \tag{34}$$

$$= \mathbf{e}^\top \frac{\alpha_t\mathbf{z}_t \odot \mathbf{x} + \alpha_{t|s}\beta_s\mathbf{z}_t \odot \mathbf{K}^\top\mathbf{x} + \beta_{t|s}\alpha_s\mathbf{K}\mathbf{z}_t \odot \mathbf{x} + \beta_{t|s}\beta_s\mathbf{K}\mathbf{z}_t \odot \mathbf{K}^\top\mathbf{x}}{\alpha_t\mathbf{z}_t^\top\mathbf{x} + \beta_t\mathbf{z}_t^\top\mathbf{K}^\top\mathbf{x}} \tag{35}$$

$$= \mathbf{e}^\top \frac{\mathbf{g}(s, t, \mathbf{x}, \mathbf{z}_t)}{f_t(\mathbf{x}, \mathbf{z}_t)} \tag{36}$$

$$= \frac{f_s(\mathbf{x}, \mathbf{e})f_{t|s}(\mathbf{e}, \mathbf{z}_t)}{f_t(\mathbf{x}, \mathbf{z}_t)} \tag{37}$$

$\square$

**Lemma 3** (Scalar KL Divergence)**.** *The term $D_{KL}(q(\mathbf{z}_s \mid \mathbf{z}_t, \mathbf{x}) \parallel p_\theta(\mathbf{z}_s \mid \mathbf{z}_t))$ can be written as:*

$$D_{KL} = \frac{\beta_{t|s}\alpha_s r(\mathbf{z}_t)}{f_t(\mathbf{x}, \mathbf{z}_t)} h_t(\mathbf{x}, \mathbf{z}_t, \tilde{\mathbf{x}}_0), \tag{38}$$

*where $h_t$ is an auxiliary function collecting the logarithmic difference terms:*

$$h_t(\mathbf{x}, \mathbf{z}_t, \tilde{\mathbf{x}}_0) = \frac{f_s(\mathbf{x}, \mathbf{z}_t)}{r(\mathbf{z}_t)} \frac{\alpha_{t|s}}{\beta_{t|s}\alpha_s} \log \frac{f_s(\mathbf{x}, \mathbf{z}_t)f_t(\tilde{\mathbf{x}}_0, \mathbf{z}_t)}{f_t(\mathbf{x}, \mathbf{z}_t)f_s(\tilde{\mathbf{x}}_0, \mathbf{z}_t)}$$
$$- \frac{1}{\alpha_s} \log \frac{f_t(\mathbf{x}, \mathbf{z}_t)}{f_t(\tilde{\mathbf{x}}_0, \mathbf{z}_t)} + \log \frac{f_s(\mathbf{x}, \mathbf{x})}{f_s(\tilde{\mathbf{x}}_0, \mathbf{x})}$$
$$+ \frac{k\beta_s}{N\alpha_s} \sum_{\mathbf{e} \in \mathcal{V}} \log \frac{f_s(\mathbf{x}, \mathbf{e})}{f_s(\tilde{\mathbf{x}}_0, \mathbf{e})}. \tag{39}$$

*Proof.* By expanding the definition of the KL divergence, we obtain:

$$D_{KL} = \sum_{\mathbf{e}} q(\mathbf{z}_s = \mathbf{e} \mid \mathbf{z}_t, \mathbf{x}) \log \frac{q(\mathbf{z}_s = \mathbf{e} \mid \mathbf{z}_t, \mathbf{x})}{p_\theta(\mathbf{z}_s = \mathbf{e} \mid \mathbf{z}_t)} \tag{40}$$

$$= \sum_{\mathbf{e}} q(\mathbf{z}_s = \mathbf{e} \mid \mathbf{z}_t, \mathbf{x}) \log \frac{q(\mathbf{z}_s = \mathbf{e} \mid \mathbf{z}_t, \mathbf{x})}{q(\mathbf{z}_s = \mathbf{e} \mid \mathbf{z}_t, \tilde{\mathbf{x}}_0)} \tag{41}$$

$$= \sum_{\mathbf{e}} \frac{f_s(\mathbf{x}, \mathbf{e}) f_{t|s}(\mathbf{e}, \mathbf{z}_t)}{f_t(\mathbf{x}, \mathbf{z}_t)} \left( \log \frac{f_s(\mathbf{x}, \mathbf{e}) f_{t|s}(\mathbf{e}, \mathbf{z}_t)}{f_t(\mathbf{x}, \mathbf{z}_t)} - \log \frac{f_s(\tilde{\mathbf{x}}_0, \mathbf{e}) f_{t|s}(\mathbf{e}, \mathbf{z}_t)}{f_t(\tilde{\mathbf{x}}_0, \mathbf{z}_t)} \right) \tag{42}$$

$$= \frac{1}{f_t(\mathbf{x}, \mathbf{z}_t)} \sum_{\mathbf{e}} f_s(\mathbf{x}, \mathbf{e}) f_{t|s}(\mathbf{e}, \mathbf{z}_t) \log \frac{f_s(\mathbf{x}, \mathbf{e}) f_t(\tilde{\mathbf{x}}_0, \mathbf{z}_t)}{f_t(\mathbf{x}, \mathbf{z}_t) f_s(\tilde{\mathbf{x}}_0, \mathbf{e})} \tag{43}$$

$$= \frac{1}{f_t(\mathbf{x}, \mathbf{z}_t)} \sum_{\mathbf{e}} f_s(\mathbf{x}, \mathbf{e}) \left( \alpha_{t|s} \delta_{\mathbf{e}, \mathbf{z}_t} + \beta_{t|s} r(\mathbf{z}_t) \right) \log \frac{f_s(\mathbf{x}, \mathbf{e}) f_t(\tilde{\mathbf{x}}_0, \mathbf{z}_t)}{f_t(\mathbf{x}, \mathbf{z}_t) f_s(\tilde{\mathbf{x}}_0, \mathbf{e})} \tag{44}$$

$$= \frac{\alpha_{t|s} f_s(\mathbf{x}, \mathbf{z}_t)}{f_t(\mathbf{x}, \mathbf{z}_t)} \log \frac{f_s(\mathbf{x}, \mathbf{z}_t) f_t(\tilde{\mathbf{x}}_0, \mathbf{z}_t)}{f_t(\mathbf{x}, \mathbf{z}_t) f_s(\tilde{\mathbf{x}}_0, \mathbf{z}_t)} \tag{45}$$

$$+ \frac{\beta_{t|s} r(\mathbf{z}_t)}{f_t(\mathbf{x}, \mathbf{z}_t)} \log \frac{f_t(\tilde{\mathbf{x}}_0, \mathbf{z}_t)}{f_t(\mathbf{x}, \mathbf{z}_t)} \sum_{\mathbf{e}} f_s(\mathbf{x}, \mathbf{e}) \tag{46}$$

$$+ \frac{\beta_{t|s} r(\mathbf{z}_t)}{f_t(\mathbf{x}, \mathbf{z}_t)} \sum_{\mathbf{e}} f_s(\mathbf{x}, \mathbf{e}) \log \frac{f_s(\mathbf{x}, \mathbf{e})}{f_s(\tilde{\mathbf{x}}_0, \mathbf{e})} \tag{47}$$

$$= \frac{\alpha_{t|s} f_s(\mathbf{x}, \mathbf{z}_t)}{f_t(\mathbf{x}, \mathbf{z}_t)} \log \frac{f_s(\mathbf{x}, \mathbf{z}_t) f_t(\tilde{\mathbf{x}}_0, \mathbf{z}_t)}{f_t(\mathbf{x}, \mathbf{z}_t) f_s(\tilde{\mathbf{x}}_0, \mathbf{z}_t)} - \frac{\beta_{t|s} r(\mathbf{z}_t)}{f_t(\mathbf{x}, \mathbf{z}_t)} \log \frac{f_t(\mathbf{x}, \mathbf{z}_t)}{f_t(\tilde{\mathbf{x}}_0, \mathbf{z}_t)} \tag{48}$$

$$+ \frac{\beta_{t|s} r(\mathbf{z}_t)}{f_t(\mathbf{x}, \mathbf{z}_t)} \sum_{\mathbf{e}} \left( \alpha_s \delta_{\mathbf{x}, \mathbf{e}} + \beta_s r(\mathbf{e}) \right) \log \frac{f_s(\mathbf{x}, \mathbf{e})}{f_s(\tilde{\mathbf{x}}_0, \mathbf{e})} \tag{49}$$

$$= \frac{\alpha_{t|s} f_s(\mathbf{x}, \mathbf{z}_t)}{f_t(\mathbf{x}, \mathbf{z}_t)} \log \frac{f_s(\mathbf{x}, \mathbf{z}_t) f_t(\tilde{\mathbf{x}}_0, \mathbf{z}_t)}{f_t(\mathbf{x}, \mathbf{z}_t) f_s(\tilde{\mathbf{x}}_0, \mathbf{z}_t)} - \frac{\beta_{t|s} r(\mathbf{z}_t)}{f_t(\mathbf{x}, \mathbf{z}_t)} \log \frac{f_t(\mathbf{x}, \mathbf{z}_t)}{f_t(\tilde{\mathbf{x}}_0, \mathbf{z}_t)} \tag{50}$$

$$+ \frac{\beta_{t|s} \alpha_s r(\mathbf{z}_t)}{f_t(\mathbf{x}, \mathbf{z}_t)} \log \frac{f_s(\mathbf{x}, \mathbf{x})}{f_s(\tilde{\mathbf{x}}_0, \mathbf{x})} \tag{51}$$

$$+ \frac{k \beta_{t|s} \beta_s r(\mathbf{z}_t)}{N f_t(\mathbf{x}, \mathbf{z}_t)} \sum_{\mathbf{e}} \log \frac{f_s(\mathbf{x}, \mathbf{e})}{f_s(\tilde{\mathbf{x}}_0, \mathbf{e})} + \frac{\mu \beta_{t|s} \beta_s r(\mathbf{z}_t)}{f_t(\mathbf{x}, \mathbf{z}_t)} \log \frac{f_s(\mathbf{x}, \mathbf{m})}{f_s(\tilde{\mathbf{x}}_0, \mathbf{m})} \tag{52}$$

$$= \frac{\beta_{t|s} \alpha_s r(\mathbf{z}_t)}{f_t(\mathbf{x}, \mathbf{z}_t)} h_t(\mathbf{x}, \mathbf{z}_t) \tag{53}$$

The proof is completed by noting that the term $\log[f_s(\mathbf{x}, \mathbf{m})/f_s(\tilde{\mathbf{x}}_0, \mathbf{m})]$ vanishes since it evaluates to zero identically for all $\mathbf{x}$ and $\tilde{\mathbf{x}}_0$.

$\square$

**Lemma 4** (Limiting Case). *As $s \to t$, the function $h_t$ converges to:*

$$h_t(\mathbf{x}, \mathbf{z}_t, \tilde{\mathbf{x}}_0) = \frac{p_{\mathbf{x}, \mathbf{z}_t} - p_{\tilde{\mathbf{x}}_0, \mathbf{z}_t}}{f_t(\tilde{\mathbf{x}}_0, \mathbf{z}_t)} - \frac{1}{\alpha_t} \log \frac{f_t(\mathbf{x}, \mathbf{z}_t)}{f_t(\tilde{\mathbf{x}}_0, \mathbf{z}_t)} +$$

$$\log \frac{f_t(\mathbf{x}, \mathbf{x})}{f_t(\tilde{\mathbf{x}}_0, \mathbf{x})} + \frac{k \beta_t}{N \alpha_t} \sum_{\mathbf{e} \in \mathcal{V}} \log \frac{f_t(\mathbf{x}, \mathbf{e})}{f_t(\tilde{\mathbf{x}}_0, \mathbf{e})}. \tag{54}$$

*Proof.* Observe that as $s \to t$, the product $\beta_{t|s} \alpha_s \to 0$, which yields an indeterminate form of type $0/0$. By applying

L'Hôpital's rule through differentiation with respect to $\alpha_{t|s}$, we obtain:

$$\lim_{s \to t} \frac{1}{\beta_{t|s}\alpha_s} \log \frac{f_s(\mathbf{x}, \mathbf{z}_t) f_t(\tilde{\mathbf{x}}_0, \mathbf{z}_t)}{f_t(\mathbf{x}, \mathbf{z}_t) f_s(\tilde{\mathbf{x}}_0, \mathbf{z}_t)} \tag{55}$$

$$= \lim_{s \to t} \frac{-1}{\alpha_s} \frac{\partial}{\partial \alpha_{t|s}} \log \frac{f_s(\mathbf{x}, \mathbf{z}_t) f_t(\tilde{\mathbf{x}}_0, \mathbf{z}_t)}{f_t(\mathbf{x}, \mathbf{z}_t) f_s(\tilde{\mathbf{x}}_0, \mathbf{z}_t)} \tag{56}$$

$$= \lim_{s \to t} \frac{-1}{\alpha_s} \left( \frac{\partial \log f_s(\mathbf{x}, \mathbf{z}_t) f_t(\tilde{\mathbf{x}}_0, \mathbf{z}_t)}{\partial \alpha_{t|s}} - \frac{\partial \log f_t(\mathbf{x}, \mathbf{z}_t) f_s(\tilde{\mathbf{x}}_0, \mathbf{z}_t)}{\partial \alpha_{t|s}} \right) \tag{57}$$

$$= \lim_{s \to t} \frac{-1}{\alpha_s} \left( \frac{\partial \log f_t(\tilde{\mathbf{x}}_0, \mathbf{z}_t)}{\partial \alpha_{t|s}} - \frac{\partial \log f_t(\mathbf{x}, \mathbf{z}_t)}{\partial \alpha_{t|s}} \right) \tag{58}$$

$$= \lim_{s \to t} \frac{-1}{\alpha_s} \left( \frac{1}{f_t(\tilde{\mathbf{x}}_0, \mathbf{z}_t)} \frac{\partial (\alpha_t p_{\tilde{\mathbf{x}}_0, \mathbf{z}_t} + (1 - \alpha_t) r(\mathbf{z}_t))}{\partial \alpha_{t|s}} - \frac{1}{f_t(\mathbf{x}, \mathbf{z}_t)} \frac{\partial f_t(\mathbf{x}, \mathbf{z}_t)}{\partial \alpha_{t|s}} \right) \tag{59}$$

$$= \lim_{s \to t} \frac{-1}{\alpha_s} \left( \frac{1}{f_t(\tilde{\mathbf{x}}_0, \mathbf{z}_t)} (\alpha_s p_{\tilde{\mathbf{x}}_0, \mathbf{z}_t} - \alpha_s r(\mathbf{z}_t)) - \frac{1}{f_t(\mathbf{x}, \mathbf{z}_t)} \frac{\partial f_t(\mathbf{x}, \mathbf{z}_t)}{\partial \alpha_{t|s}} \right) \tag{60}$$

$$= -\frac{p_{\tilde{\mathbf{x}}_0, \mathbf{z}_t} - r(\mathbf{z}_t)}{f_t(\tilde{\mathbf{x}}_0, \mathbf{z}_t)} + \frac{p_{\mathbf{x}, \mathbf{z}_t} - r(\mathbf{z}_t)}{f_t(\mathbf{x}, \mathbf{z}_t)} \tag{61}$$

$$= \frac{(\alpha_t p_{\tilde{\mathbf{x}}_0, \mathbf{z}_t} + \beta_t r(\mathbf{z}_t))(p_{\mathbf{x}, \mathbf{z}_t} - r(\mathbf{z}_t)) - (\alpha_t p_{\mathbf{x}, \mathbf{z}_t} + \beta_t r(\mathbf{z}_t))(p_{\tilde{\mathbf{x}}_0, \mathbf{z}_t} - r(\mathbf{z}_t))}{f_t(\mathbf{x}, \mathbf{z}_t) f_t(\tilde{\mathbf{x}}_0, \mathbf{z}_t)} \tag{62}$$

$$= \frac{-\alpha_t p_{\tilde{\mathbf{x}}_0, \mathbf{z}_t} r(\mathbf{z}_t) + \beta_t p_{\mathbf{x}, \mathbf{z}_t} r(\mathbf{z}_t) + \alpha_t p_{\mathbf{x}, \mathbf{z}_t} r(\mathbf{z}_t) - \beta_t p_{\tilde{\mathbf{x}}_0, \mathbf{z}_t} r(\mathbf{z}_t)}{f_t(\mathbf{x}, \mathbf{z}_t) f_t(\tilde{\mathbf{x}}_0, \mathbf{z}_t)} \tag{63}$$

$$= \frac{r(\mathbf{z}_t)(p_{\mathbf{x}, \mathbf{z}_t} - p_{\tilde{\mathbf{x}}_0, \mathbf{z}_t})}{f_t(\mathbf{x}, \mathbf{z}_t) f_t(\tilde{\mathbf{x}}_0, \mathbf{z}_t)} \tag{64}$$

Taking the limit of the full leading term inside $h_t$ yields:

$$\lim_{s \to t} \left( \frac{f_s(\mathbf{x}, \mathbf{z}_t)}{r(\mathbf{z}_t)} \frac{\alpha_{t|s}}{\beta_{t|s}\alpha_s} \log \frac{f_s(\mathbf{x}, \mathbf{z}_t) f_t(\tilde{\mathbf{x}}_0, \mathbf{z}_t)}{f_t(\mathbf{x}, \mathbf{z}_t) f_s(\tilde{\mathbf{x}}_0, \mathbf{z}_t)} \right) = \frac{p_{\mathbf{x}, \mathbf{z}_t} - p_{\tilde{\mathbf{x}}_0, \mathbf{z}_t}}{f_t(\tilde{\mathbf{x}}_0, \mathbf{z}_t)} \tag{65}$$

$\square$

## C. Relationship to MDLM and UDLM

**Lemma 5.** *XDLM can be reduced to MDLM by setting $k = 0$, with the posterior probability:*

$$p_\theta(\mathbf{z_s} = \mathbf{e} \mid \mathbf{z}_t) = \delta_{\mathbf{z}_t, \mathbf{m}} \delta_{\mathbf{e}, \mathbf{m}} \frac{\beta_s}{\beta_t} + \delta_{\mathbf{z}_t, \mathbf{m}} \delta_{\mathbf{e} \neq \mathbf{m}} \frac{\beta_{t|s}\alpha_s p_{\theta, \mathbf{e}}}{\beta_t} + \delta_{\mathbf{z}_t \neq \mathbf{m}} \delta_{\mathbf{e}, \mathbf{z}_t} \tag{66}$$

$$= \begin{cases} \frac{\beta_s}{\beta_t} \delta_{\mathbf{m}, \mathbf{z}_t}, & \mathbf{e} = \mathbf{m} \\ \frac{(\alpha_s - \alpha_t)}{1 - \alpha_t} p_{\theta, \mathbf{e}}, & \mathbf{z}_t = \mathbf{m}, \mathbf{e} \neq \mathbf{m} \\ \delta_{\mathbf{e}, \mathbf{z}_t}, & \mathbf{z}_t \neq \mathbf{m}, \mathbf{e} \neq \mathbf{m} \end{cases} \tag{67}$$

*and KL divergence*

$$D_{KL} = -\frac{\beta_{t|s}\alpha_s}{\beta_t} \log p_{\theta, \mathbf{x}} \tag{68}$$

*Proof.* since $k = 0$, $\mu = 1$, we have

$$r(\mathbf{e}) = \delta_{\mathbf{e}, \mathbf{m}} \tag{69}$$

$$p_{\tilde{\mathbf{x}}_0, \mathbf{e}} = \delta_{\mathbf{e} \neq \mathbf{m}} p_{\tilde{\mathbf{x}}_0, \mathbf{e}} \tag{70}$$

hence

$$q(\mathbf{z_s} = \mathbf{e} \mid \mathbf{z}_t, \tilde{\mathbf{x}}_0) = \frac{f_s(\tilde{\mathbf{x}}_0, \mathbf{e}) f_{t|s}(\mathbf{e}, \mathbf{z}_t)}{f_t(\tilde{\mathbf{x}}_0, \mathbf{z}_t)} \tag{71}$$

$$= \frac{(\alpha_s p_{\tilde{\mathbf{x}}_0, \mathbf{e}} + \beta_s r(\mathbf{e}))(\alpha_{t|s} p_{\mathbf{e}, \mathbf{z}_t} + \beta_{t|s} r(\mathbf{z}_t))}{\alpha_t p_{\tilde{\mathbf{x}}_0, \mathbf{z}_t} + \beta_t r(\mathbf{z}_t)} \tag{72}$$

$$= \frac{\alpha_t p_{\tilde{\mathbf{x}}_0, \mathbf{e}} \delta_{\mathbf{e}, \mathbf{z}_t} + \alpha_s \beta_{t|s} p_{\tilde{\mathbf{x}}_0, \mathbf{e}} \delta_{\mathbf{z}_t, \mathbf{m}} + \alpha_{t|s} \beta_s \delta_{\mathbf{e}, \mathbf{z}_t} \delta_{\mathbf{e}, \mathbf{m}} + \beta_s \beta_{t|s} \delta_{\mathbf{e}, \mathbf{m}} \delta_{\mathbf{z}_t, \mathbf{m}}}{\alpha_t p_{\tilde{\mathbf{x}}_0, \mathbf{z}_t} + \beta_t \delta_{\mathbf{z}_t, \mathbf{m}}} \tag{73}$$

$$= \delta_{\mathbf{z}_t, \mathbf{m}} \frac{\alpha_s \beta_{t|s} p_{\tilde{\mathbf{x}}_0, \mathbf{e}} + \alpha_{t|s} \beta_s \delta_{\mathbf{e}, \mathbf{z}_t} + \beta_s \beta_{t|s} \delta_{\mathbf{e}, \mathbf{z}_t}}{\beta_t} \tag{74}$$

$$+ \delta_{\mathbf{z}_t \neq \mathbf{m}} \frac{\alpha_t p_{\tilde{\mathbf{x}}_0, \mathbf{e}} \delta_{\mathbf{e}, \mathbf{z}_t}}{\alpha_t p_{\tilde{\mathbf{x}}_0, \mathbf{z}_t}} \tag{75}$$

$$= \delta_{\mathbf{z}_t, \mathbf{m}} \left( \frac{\alpha_s \beta_{t|s}}{\beta_t} p_{\tilde{\mathbf{x}}_0, \mathbf{e}} + \frac{\beta_s}{\beta_t} \delta_{\mathbf{e}, \mathbf{z}_t} \right) + \delta_{\mathbf{z}_t \neq \mathbf{m}} \delta_{\mathbf{e}, \mathbf{z}_t} \frac{p_{\tilde{\mathbf{x}}_0, \mathbf{e}}}{p_{\tilde{\mathbf{x}}_0, \mathbf{z}_t}} \tag{76}$$

$$= \delta_{\mathbf{z}_t, \mathbf{m}} \frac{\alpha_s \beta_{t|s}}{\beta_t} p_{\tilde{\mathbf{x}}_0, \mathbf{e}} + \frac{\beta_s}{\beta_t} \delta_{\mathbf{z}_t, \mathbf{m}} \delta_{\mathbf{e}, \mathbf{z}_t} + \delta_{\mathbf{z}_t \neq \mathbf{m}} \delta_{\mathbf{e}, \mathbf{z}_t} \tag{77}$$

for KL, we have

$$h_t(\mathbf{x}, \mathbf{z}_t) = \frac{f_s(\mathbf{x}, \mathbf{z}_t)}{r(\mathbf{z}_t)} \frac{\alpha_{t|s}}{\beta_{t|s} \alpha_s} \log \frac{f_s(\mathbf{x}, \mathbf{z}_t) f_t(\tilde{\mathbf{x}}_0, \mathbf{z}_t)}{f_t(\mathbf{x}, \mathbf{z}_t) f_s(\tilde{\mathbf{x}}_0, \mathbf{z}_t)} \tag{78}$$

$$- \frac{1}{\alpha_s} \log \frac{f_t(\mathbf{x}, \mathbf{z}_t)}{f_t(\tilde{\mathbf{x}}_0, \mathbf{z}_t)} + \log \frac{f_s(\mathbf{x}, \mathbf{x})}{f_s(\tilde{\mathbf{x}}_0, \mathbf{x})} + \frac{k\beta_s}{N\alpha_s} \sum_{\mathbf{e}} \log \frac{f_s(\mathbf{x}, \mathbf{e})}{f_s(\tilde{\mathbf{x}}_0, \mathbf{e})} \tag{79}$$

$$= \frac{f_s(\mathbf{x}, \mathbf{z}_t)}{r(\mathbf{z}_t)} \frac{\alpha_{t|s}}{\beta_{t|s} \alpha_s} \log \frac{(\alpha_s p_{\mathbf{x}, \mathbf{z}_t} + \beta_s \delta_{\mathbf{z}_t, \mathbf{m}})(\alpha_t p_{\tilde{\mathbf{x}}_0, \mathbf{z}_t} + \beta_t \delta_{\mathbf{z}_t, \mathbf{m}})}{(\alpha_t p_{\mathbf{x}, \mathbf{z}_t} + \beta_t \delta_{\mathbf{z}_t, \mathbf{m}})(\alpha_s p_{\tilde{\mathbf{x}}_0, \mathbf{z}_t} + \beta_s \delta_{\mathbf{z}_t, \mathbf{m}})} \tag{80}$$

$$- \frac{1}{\alpha_s} \log \frac{\alpha_t p_{\mathbf{x}, \mathbf{z}_t} + \beta_t \delta_{\mathbf{z}_t, \mathbf{m}}}{\alpha_t p_{\tilde{\mathbf{x}}_0, \mathbf{z}_t} + \beta_t \delta_{\mathbf{z}_t, \mathbf{m}}} - \log p_{\tilde{\mathbf{x}}_0, \mathbf{x}} \tag{81}$$

$$= \frac{f_s(\mathbf{x}, \mathbf{z}_t)}{r(\mathbf{z}_t)} \frac{\alpha_{t|s}}{\beta_{t|s} \alpha_s} \log \left( \delta_{\mathbf{z}_t, \mathbf{m}} \frac{\beta_s \delta_{\mathbf{z}_t, \mathbf{m}} \beta_t \delta_{\mathbf{z}_t, \mathbf{m}}}{\beta_t \delta_{\mathbf{z}_t, \mathbf{m}} \beta_s \delta_{\mathbf{z}_t, \mathbf{m}}} + \delta_{\mathbf{z}_t \neq \mathbf{m}} \frac{\alpha_s p_{\mathbf{x}, \mathbf{z}_t} \alpha_t p_{\tilde{\mathbf{x}}_0, \mathbf{z}_t}}{\alpha_t p_{\mathbf{x}, \mathbf{z}_t} \alpha_s p_{\tilde{\mathbf{x}}_0, \mathbf{z}_t}} \right) \tag{82}$$

$$- \frac{1}{\alpha_s} \log \left( \delta_{\mathbf{z}_t, \mathbf{m}} \frac{\beta_t \delta_{\mathbf{z}_t, \mathbf{m}}}{\beta_t \delta_{\mathbf{z}_t, \mathbf{m}}} + \delta_{\mathbf{z}_t \neq \mathbf{m}} \frac{\alpha_t p_{\mathbf{x}, \mathbf{z}_t}}{\alpha_t p_{\tilde{\mathbf{x}}_0, \mathbf{z}_t}} \right) - \log p_{\tilde{\mathbf{x}}_0, \mathbf{x}} \tag{83}$$

$$= \frac{f_s(\mathbf{x}, \mathbf{z}_t)}{r(\mathbf{z}_t)} \frac{\alpha_{t|s}}{\beta_{t|s} \alpha_s} \log(1) - \frac{1}{\alpha_s} \delta_{\mathbf{z}_t \neq \mathbf{m}} \log \frac{p_{\mathbf{x}, \mathbf{z}_t}}{p_{\tilde{\mathbf{x}}_0, \mathbf{z}_t}} - \log p_{\tilde{\mathbf{x}}_0, \mathbf{x}} \tag{84}$$

so

$$D_{KL} = \frac{\beta_{t|s} \alpha_s r(\mathbf{z}_t)}{f_t(\mathbf{x}, \mathbf{z}_t)} h_t(\mathbf{x}, \mathbf{z}_t) \tag{85}$$

$$= -\frac{\beta_{t|s} \delta_{\mathbf{z}_t, \mathbf{m}}}{f_t(\mathbf{x}, \mathbf{z}_t)} \delta_{\mathbf{z}_t \neq \mathbf{m}} \log \frac{p_{\mathbf{x}, \mathbf{z}_t}}{p_{\tilde{\mathbf{x}}_0, \mathbf{z}_t}} - \frac{\beta_{t|s} \alpha_s \delta_{\mathbf{z}_t, \mathbf{m}}}{\alpha_t \delta_{\mathbf{x}, \mathbf{z}_t} + \beta_t \delta_{\mathbf{z}_t, \mathbf{m}}} \log p_{\tilde{\mathbf{x}}_0, \mathbf{x}} \tag{86}$$

$$= -\delta_{\mathbf{z}_t, \mathbf{m}} \frac{\beta_{t|s} \alpha_s}{\beta_t} \log p_{\tilde{\mathbf{x}}_0, \mathbf{x}} \tag{87}$$

$$\square$$

**Lemma 6.** *XDLM can be reduced to UDLM by setting $k = 1$, with the posterior probability:*

$$p_\theta(\mathbf{z_s} = \mathbf{e} \mid \mathbf{z}_t) = \frac{\delta_{\mathbf{e}, \mathbf{z}_t}(N\alpha_t p_{\theta, \mathbf{e}} + \beta_s \alpha_{t|s}) + (\alpha_s - \alpha_t) p_{\theta, \mathbf{e}} + \beta_s \beta_{t|s}/N}{N\alpha_t p_{\theta, \mathbf{z}_t} + \beta_t} \tag{88}$$

*and KL divergence*

$$D_{KL} = \frac{-\beta_{t|s}\alpha_s}{N\alpha_t} \left( \frac{1}{f_t(\tilde{\mathbf{x}}_0, \mathbf{z}_t)} - \frac{1}{f_t(\mathbf{x}, \mathbf{z}_t)} - \sum_{\mathbf{e}\neq\mathbf{z}_t} \frac{f_t(\mathbf{x}, \mathbf{e})}{f_t(\mathbf{x}, \mathbf{z}_t)} \log \frac{f_t(\tilde{\mathbf{x}}_0, \mathbf{z}_t)f_t(\mathbf{x}, \mathbf{e})}{f_t(\tilde{\mathbf{x}}_0, \mathbf{e})f_t(\mathbf{x}, \mathbf{z}_t)} \right) \tag{89}$$

*Proof.* since $k = 1$, $\mu = 0$, we have

$$\forall \mathbf{e}, r(\mathbf{e}) = r = \frac{1}{N} \tag{90}$$

Hence,

$$q(\mathbf{z_s} = \mathbf{e} \mid \mathbf{z}_t, \tilde{\mathbf{x}}_0) = \frac{f_s(\tilde{\mathbf{x}}_0, \mathbf{e})f_{t|s}(\mathbf{e}, \mathbf{z}_t)}{f_t(\tilde{\mathbf{x}}_0, \mathbf{z}_t)} \tag{91}$$

$$= \frac{(\alpha_s p_{\tilde{\mathbf{x}}_0, \mathbf{e}} + \beta_s/N)(\alpha_{t|s}p_{\mathbf{e}, \mathbf{z}_t} + \beta_{t|s}/N)}{\alpha_t p_{\tilde{\mathbf{x}}_0, \mathbf{z}_t} + \beta_t/N} \tag{92}$$

$$= \frac{N\alpha_t p_{\tilde{\mathbf{x}}_0, \mathbf{e}}\delta_{\mathbf{e}, \mathbf{z}_t} + \alpha_s\beta_{t|s}p_{\tilde{\mathbf{x}}_0, \mathbf{e}} + \alpha_{t|s}\beta_s\delta_{\mathbf{e}, \mathbf{z}_t} + \beta_s\beta_{t|s}/N}{N\alpha_t p_{\tilde{\mathbf{x}}_0, \mathbf{z}_t} + \beta_t} \tag{93}$$

$$= \frac{\delta_{\mathbf{e}, \mathbf{z}_t}\left(N\alpha_t p_{\tilde{\mathbf{x}}_0, \mathbf{e}} + \alpha_{t|s}\beta_s\right) + (\alpha_s - \alpha_t)p_{\tilde{\mathbf{x}}_0, \mathbf{e}} + \beta_s\beta_{t|s}/N}{N\alpha_t p_{\tilde{\mathbf{x}}_0, \mathbf{z}_t} + \beta_t} \tag{94}$$

Given $s \to t$, for the KL divergence, we have

$$h_t(\mathbf{x}, \mathbf{z}_t) \tag{95}$$

$$= \frac{p_{\mathbf{x}, \mathbf{z}_t} - p_{\tilde{\mathbf{x}}_0, \mathbf{z}_t}}{f_t(\tilde{\mathbf{x}}_0, \mathbf{z}_t)} - \frac{1}{\alpha_t}\log\frac{f_t(\mathbf{x}, \mathbf{z}_t)}{f_t(\tilde{\mathbf{x}}_0, \mathbf{z}_t)} + \log\frac{f_t(\mathbf{x}, \mathbf{x})}{f_t(\tilde{\mathbf{x}}_0, \mathbf{x})} + \frac{k\beta_t}{N\alpha_t}\sum_{\mathbf{e}}\log\frac{f_t(\mathbf{x}, \mathbf{e})}{f_t(\tilde{\mathbf{x}}_0, \mathbf{e})} \tag{96}$$

$$= f_t(\mathbf{x}, \mathbf{z}_t)\frac{p_{\mathbf{x}, \mathbf{z}_t} - p_{\tilde{\mathbf{x}}_0, \mathbf{z}_t}}{f_t(\mathbf{x}, \mathbf{z}_t)f_t(\tilde{\mathbf{x}}_0, \mathbf{z}_t)} - \frac{1}{\alpha_t}\log\frac{f_t(\mathbf{x}, \mathbf{z}_t)}{f_t(\tilde{\mathbf{x}}_0, \mathbf{z}_t)} + \log\frac{f_t(\mathbf{x}, \mathbf{x})}{f_t(\tilde{\mathbf{x}}_0, \mathbf{x})} \tag{97}$$

$$+ \frac{\beta_t}{N\alpha_t}\sum_{\mathbf{e}}\log\frac{f_t(\tilde{\mathbf{x}}_0, \mathbf{z}_t)f_t(\mathbf{x}, \mathbf{e})}{f_t(\tilde{\mathbf{x}}_0, \mathbf{e})f_t(\mathbf{x}, \mathbf{z}_t)} - \frac{\beta_t}{N\alpha_t}\sum_{\mathbf{e}}\log\frac{f_t(\tilde{\mathbf{x}}_0, \mathbf{z}_t)}{f_t(\mathbf{x}, \mathbf{z}_t)} \tag{98}$$

$$= \frac{f_t(\mathbf{x}, \mathbf{z}_t)}{\alpha_t}\left(\frac{1}{f_t(\tilde{\mathbf{x}}_0, \mathbf{z}_t)} - \frac{1}{f_t(\mathbf{x}, \mathbf{z}_t)}\right) - \log\frac{f_t(\mathbf{x}, \mathbf{z}_t)}{f_t(\tilde{\mathbf{x}}_0, \mathbf{z}_t)} + \log\frac{f_t(\mathbf{x}, \mathbf{x})}{f_t(\tilde{\mathbf{x}}_0, \mathbf{x})} \tag{99}$$

$$+ \frac{\beta_t}{N\alpha_t}\sum_{\mathbf{e}}\log\frac{f_t(\tilde{\mathbf{x}}_0, \mathbf{z}_t)f_t(\mathbf{x}, \mathbf{e})}{f_t(\tilde{\mathbf{x}}_0, \mathbf{e})f_t(\mathbf{x}, \mathbf{z}_t)} \tag{100}$$

$$= \frac{f_t(\mathbf{x}, \mathbf{z}_t)}{\alpha_t}\left(\frac{1}{f_t(\tilde{\mathbf{x}}_0, \mathbf{z}_t)} - \frac{1}{f_t(\mathbf{x}, \mathbf{z}_t)}\right) - \log\frac{f_t(\mathbf{x}, \mathbf{z}_t)}{f_t(\tilde{\mathbf{x}}_0, \mathbf{z}_t)} + \log\frac{f_t(\mathbf{x}, \mathbf{x})}{f_t(\tilde{\mathbf{x}}_0, \mathbf{x})} \tag{101}$$

$$+ \frac{1}{\alpha_t}\sum_{\mathbf{e}}f_t(\mathbf{x}, \mathbf{e})\log\frac{f_t(\tilde{\mathbf{x}}_0, \mathbf{z}_t)f_t(\mathbf{x}, \mathbf{e})}{f_t(\tilde{\mathbf{x}}_0, \mathbf{e})f_t(\mathbf{x}, \mathbf{z}_t)} - \log\frac{f_t(\tilde{\mathbf{x}}_0, \mathbf{z}_t)f_t(\mathbf{x}, \mathbf{x})}{f_t(\tilde{\mathbf{x}}_0, \mathbf{x})f_t(\mathbf{x}, \mathbf{z}_t)} \tag{102}$$

$$= \frac{f_t(\mathbf{x}, \mathbf{z}_t)}{\alpha_t}\left(\frac{1}{f_t(\tilde{\mathbf{x}}_0, \mathbf{z}_t)} - \frac{1}{f_t(\mathbf{x}, \mathbf{z}_t)}\right) + \frac{1}{\alpha_t}\sum_{\mathbf{e}}f_t(\mathbf{x}, \mathbf{e})\log\frac{f_t(\tilde{\mathbf{x}}_0, \mathbf{z}_t)f_t(\mathbf{x}, \mathbf{e})}{f_t(\tilde{\mathbf{x}}_0, \mathbf{e})f_t(\mathbf{x}, \mathbf{z}_t)} \tag{103}$$

So,

$$D_{KL} = \frac{\beta_{t|s}\alpha_s r(\mathbf{z}_t)}{f_t(\mathbf{x}, \mathbf{z}_t)}h_t(\mathbf{x}, \mathbf{z}_t) \tag{104}$$

$$= \frac{-\beta_{t|s}\alpha_s}{N\alpha_t}\left(\frac{1}{f_t(\mathbf{x}, \mathbf{z}_t)} - \frac{1}{f_t(\tilde{\mathbf{x}}_0, \mathbf{z}_t)} - \sum_{\mathbf{e}}\frac{f_t(\mathbf{x}, \mathbf{e})}{f_t(\mathbf{x}, \mathbf{z}_t)}\log\frac{f_t(\tilde{\mathbf{x}}_0, \mathbf{z}_t)f_t(\mathbf{x}, \mathbf{e})}{f_t(\tilde{\mathbf{x}}_0, \mathbf{e})f_t(\mathbf{x}, \mathbf{z}_t)}\right) \tag{105}$$

$\square$

# D. PyTorch-style Pseudo-code for Training and Sampling

This section provides PyTorch-style pseudocode for the core training and sampling procedures of XDLM. Algorithm 1 presents the one-step reverse transition used by our ancestral sampler. This sampler is used in all sampling experiments, except for LLaDA generation. Algorithm 2 presents the scalar-reformulated KL computation used for XDLM training, including the continual pre-training of LLaDA-XDLM. In the special case where $\alpha_s = \alpha_t$, the full KL routine get_kl can be reduced to the simplified implementation get_kl_simp, which is obtained by canceling terms that become identical under $\alpha_s = \alpha_t$. The simplified version is included in our released codebase.

---

**Algorithm 1** Scalar-reformulated one-step reverse transition for XDLM sampling

---

```python
def sample_one_step(
    cls,
    logits: torch.Tensor,
    inputs: torch.Tensor,
    k1: float = 0.1,
    mask_id: Optional[int] = None,
    alpha_t: torch.Tensor = 0.1,
    alpha_s: torch.Tensor = 0.1,
    probs: torch.Tensor = None,
):
    """Performs one reverse diffusion step from time t to s (where t > s).

    Comment:
        f_t(x, e) = a_t <e,x> + b_t (k1 / v + k2 * (e == m))
        out = f_s(x_th, e) / f_t(x_th, zt) * f_{t|s}(zt, e)

    Args:
        logits: Predicted token distributions of shape [batch, length, vocab].
        inputs: Noisy input tokens (z_t) of shape [batch, length].
        k1: mixing ratio of absorb and uniform noise.
        mask_id: Index of the [MASK] token in the vocabulary.
        alpha_t: Signal rate at current timestep t.
        alpha_s: Signal rate at previous timestep s (alpha_s >= alpha_t).
        probs: Optional pre-computed probability distribution [B, L, V].

    Returns:
        out_probs: The probability distribution for the next state z_s [B, L, V].
    """
    shape = logits.shape if logits is not None else probs.shape
    b, l, v = shape

    k2 = 1 - k1
    alpha_t = alpha_t.view(b, 1, 1)
    alpha_s = alpha_s.view(b, 1, 1)
    beta_t = 1 - alpha_t
    beta_s = 1 - alpha_s
    beta_ts = (alpha_s - alpha_t) / alpha_s

    zt_eq_m = (inputs == mask_id).unsqueeze_(-1)
    probs = cls.get_probs(logits, mask_id=mask_id, probs=probs)
    prob_zt = torch.gather(probs, dim=-1, index=inputs.unsqueeze(-1)) # (b, l, 1)

    # when current state zt is the mask token
    probs_from_mask = (alpha_s - alpha_t) * probs / beta_t + k1 * beta_ts * beta_s / beta_t / v
    if mask_id is not None:
        probs_mask_from_mask = (beta_s / beta_t - k1 * beta_ts * beta_s * (v - 1) / (beta_t * v))
        probs_from_mask[:, :, mask_id] = probs_mask_from_mask.squeeze(-1)

    # when current state zt is not the mask token
    vfprob_s = alpha_s * v * probs + beta_s * k1 # v * f_s(x_th,e)
    vfprob_t_zt = alpha_t * v * prob_zt + beta_t * k1 # v * f_t(x_th,zt)
    probs_from_token = k1 * beta_ts * vfprob_s / vfprob_t_zt / v # f_s(x_th,e) / f_t(x_th,zt) * b_(t|s) (k1 / v)
    probs_from_token_zt_add = 1 - k1 * beta_ts / vfprob_t_zt # f_s(x_th,e) / f_t(x_th,zt) * (e == zt) a_(t|s)
    probs_from_token = torch.scatter_add(probs_from_token, dim=-1, index=inputs.unsqueeze(-1), src=probs_from_token_zt_add)
    if mask_id is not None:
        probs_mask_from_token = (k1 * beta_ts * beta_s * (k1 / v + k2) / vfprob_t_zt)
        probs_from_token[:, :, mask_id] = probs_mask_from_token.squeeze(-1)

    out_probs = torch.where(zt_eq_m, probs_from_mask, probs_from_token)
    return out_probs
```

---

**Algorithm 2** Scalar-reformulated KL computation for XDLM training

```python
def get_kl(
    cls,
    logits: torch.Tensor,
    inputs: torch.Tensor,
    labels: torch.Tensor,
    mask_id: Optional[int] = None,
    k1: float = 0.1,
    alpha_t: torch.Tensor = 0.1,
    alpha_s: torch.Tensor = 0.1,
    delta_alpha_scale: Union[torch.Tensor, float] = 1.0,
    limit_case: bool = True,
    probs: torch.Tensor = None,
):
    """Computes the KL divergence between diffusion timesteps s and t.

    Comment:
        D_KL = delta_alpha_scale * rdivt * h_t(x, zt, x_th)
        h_t(x, zt, x_th) = party / rdivt
            - frac{1}{a_s} log frac{f_t(x,zt)} {f_t(x_th,zt)}
            + log frac{f_s(x,x)} {f_s(x_th,x)} + frac{k b_{s}}{a_{s}} partx

    Args:
        labels: Ground truth tokens (x_0) of shape [batch, length].
        alpha_s: Signal rate at previous timestep s (alpha_s >= alpha_t).
        delta_alpha_scale: Scaling factor (alpha_s - alpha_t) to align KL with NELBO loss.

    Returns:
        KL divergence of shape [batch, length].
    """
    b, l, v = logits.shape
    # b, l = inputs.shape
    # b, l = labels.shape

    alpha_t = alpha_t.view(b, 1, 1)
    alpha_s = alpha_s.view(b, 1, 1)
    beta_t = 1 - alpha_t
    beta_s = 1 - alpha_s

    if not isinstance(delta_alpha_scale, torch.Tensor):
        delta_alpha_scale = torch.full_like(alpha_t, fill_value=delta_alpha_scale)

    zt_eq_x = (inputs == labels).unsqueeze_(-1)
    zt_eq_m = (inputs == mask_id).unsqueeze_(-1)
    vratio = k1 + v * (1 - k1) * zt_eq_m
    probs = cls.get_probs(logits, mask_id=mask_id, probs=probs)

    prob_x0 = torch.gather(probs, dim=-1, index=labels.unsqueeze(-1)) # (b, l, 1)
    prob_zt = torch.gather(probs, dim=-1, index=inputs.unsqueeze(-1)) # (b, l, 1)
    vfprob_s_x0 = alpha_s * v * prob_x0 + beta_s * k1
    vfprob_s_zt = alpha_s * v * prob_zt + beta_s * vratio
    vfprob_t_zt = alpha_t * v * prob_zt + beta_t * vratio

    vfhard_s_x0 = alpha_s * v + beta_s * k1
    vfhard_s_zt = alpha_s * v * zt_eq_x + beta_s * vratio
    vfhard_t_zt = alpha_t * v * zt_eq_x + beta_t * vratio
    rdivt = torch.where(zt_eq_x, k1 / (v * alpha_t + beta_t * k1), 1 / beta_t) # r(z_t) / f_t(x,z_t)

    if limit_case or (alpha_s - alpha_t).min() < 1e-6:
        party = (v * (zt_eq_x.float() - prob_zt) / vfprob_s_zt) * (vratio / vfhard_t_zt)
    else:
        tmpx_zt_eq_x = (v * alpha_t * alpha_s + alpha_t * beta_s * k1) / (v * alpha_s * alpha_t + alpha_s * beta_t * k1)
        tmpx_zt_ne_x = (alpha_t * beta_s) / (alpha_s * beta_t)
        tmpx = torch.where(zt_eq_x, tmpx_zt_eq_x, tmpx_zt_ne_x)
        tmpy = (v * (zt_eq_x.float() - prob_zt) / vfprob_s_zt) * rdivt
        party = tmpx * torch.log1p(tmpy * (alpha_s - alpha_t)) / (alpha_s - alpha_t)

    if k1 > 0:
        partx_0 = - (v * alpha_s * probs + beta_s * k1).log().mean(dim=-1, keepdim=True)
        partx_1 = (beta_s * k1).log() * (v - 1) / v
        partx_2 = (v * alpha_s + beta_s * k1).log() / v
        partx = partx_0 + partx_1 + partx_2

    kl = delta_alpha_scale.view(-1, 1) * (
        party
        - rdivt / alpha_s * (vfhard_t_zt.log() - vfprob_t_zt.log())
        + rdivt * (vfhard_s_x0.log() - vfprob_s_x0.log())
        + (k1 * beta_s * rdivt * partx / alpha_s if k1 > 0 else 0)
    ).view(b, l)

    return kl
```

# E. LM1B Sampling Case

We investigate the relationship between the inference computational budget (defined by the number of sampling steps $T$) and the perceptual quality of the generated text. Table 7 presents randomly selected, non-curated samples from the XDLM model trained on the LM1B dataset. Each sample represents the final output of a distinct diffusion process constrained to a specific total step count $T \in \{4, 8, 16, 32, 64, 128\}$.

At the lower bound of the sampling budget ($T = 4$), the model fails to converge to the data manifold. The output is characterized by disjointed lexical retrieval (e.g., "*the advertising -tag prong*") where individual tokens are valid but lack syntactic binding. Increasing the budget to $T = 8$ yields the emergence of superficial syntactic structures (e.g., "*curt schilling is a man of newness*"), yet the semantic content remains illogical, indicating that the diffusion process resolves local grammatical dependencies prior to higher-order semantic meaning.

As the sampling steps increase to the intermediate regime ($T = 16$ and $T = 32$), we observe a transition from mere grammatical correctness to narrative plausibility. At $T = 16$, the text exhibits valid sentence structures but suffers from tautological repetition and thematic drift (e.g., "*middle east in the middle east*"). By $T = 32$, the model generates cohesive clauses with clear subject-verb-object relationships, although the specific entities often remain synthetic or hallucinatory (e.g., "*mr poundhead*"). This suggests that while 32 steps are sufficient for the model to learn the structural rules of language, the denoising process has not yet fully aligned the output with the specific factual distributions of the training corpus.

A distinct phase transition in sample quality is observed at $T = 64$, which appears to represent the saturation point for high-fidelity generation. At this stage, the model produces semantically robust text indistinguishable from natural news data, successfully handling complex entity lists (e.g., "*Congo, Benin, Ivory Coast*") and domain-specific terminology (e.g., "*National Transportation Safety Board*"). Extending the process to $T = 128$ yields marginal improvements in long-range consistency and the precision of quantitative figures, but the perceptual gain over $T = 64$ is significantly smaller than the leap observed from $T = 32$ to $T = 64$. This trajectory indicates that while XDLM requires a minimum threshold of approximately 64 steps to resolve fine-grained semantic details, further increases in computational cost yield diminishing returns in generation quality.

*Table 7.* **Uncurated samples generated by XDLM trained on LM1B.** Each sample represents the output of a distinct diffusion process constrained to a specific total step count $T$. The sequence length is fixed at 128 tokens.

| Steps ($T$) | Generated Text Sample |
|---|---|
| $T = 4$ | [CLS]n.  [CLS] the advertising -tag prong those former candidates is sept.  [CLS] new and aspiring contenders boasted a present, confusing not candidates not only for 15 - u running bonds.  [CLS] passengers on board the puerto rico jet jet out to the ash oftwa, hawaii, wednesday that afternoon.  [CLS] as a result of the fusion going from in horsepower and to okter and the ram and to gm, s rentalearing / thirds was the infall.  [CLS] times square may undergo restorations [CLS] and " old allah " is an homage to " the a, the " god " in snowy islamic faith.  [CLS] unfortunately christina sees, [CLS] |
| $T = 8$ | [CLS] be obese to go over.  [CLS] curt schilling is a man of newness.  [CLS] mr middle had, having not started the business for 15 years, wondered for one only anyone on board had once wanted from kraft.  [CLS] the new oft - appointed owner proposed wednesday that the administration should take a " punitive tax supplement from treasury " and insisted on indiana and michigan adhering to gm's contract terms.  [CLS] ms. cummings :  one at times decides to undergo restoration.  [CLS] and for now pakistani parliament is now resigned to a democratic, destabilization vote in parting zardari, which sees itself [CLS] |

| Steps ($T$) | Generated Text Sample (Continued) |
|---|---|
| $T = 16$ | `[CLS] " fails to clear advertising restrictions. [CLS] the network, which is the only known new middle east in the middle east, is expected for about 10 to 15 years. [CLS] i believe the people on board had called very pleased, how i was able to explain it in me. [CLS] the studio 86 is nearly a lot of contemporary televised – auto show – marquees – cars. the pre – gm bio tasted like two – thirds of iconic americans : one – third of the time that ford used and around 4 years the original was back to a building it made years of. [CLS] in contrast, zardari, who sees washington [CLS]` |
| $T = 32$ | `[CLS] me, to clear the books. [CLS] that sense of neutrality is likely to force some postponements of the dispute, which was resolved wednesday. [CLS] 15 told the telegraph that he always believed the board had convinced him it was time to take the train. [CLS] mr poundhead wants the city council to create a series of new visitors' areas. [CLS] marfor, according to le monde, " is the afghans to steal or take and foot out " – – money the taliban has often used to buy from pakistani taliban, not pakistan's. [CLS] though some of its friends in the west zardari said he sees k [CLS]` |
| $T = 64$ | `[CLS] me, was the crew of the xv230, which is carrying congo, benin, ivory coast, cape verde, manchester russians and coastguards at luanda. [CLS] the national transportation safety board had called the electrical life support vests to a safe spot above hawaii. [CLS] the daily herald said analysts expect earnings of $ 394 million, or 43 cents a share, due to better – than – expected growth in the mini market. [CLS] general cuddy writes that the controls could be used to prevent revenge attacks, but not to halt the protests by those suspected of disturbing network in the island, throwing up the theory. [CLS] [CLS]` |
| $T = 128$ | `[CLS] me to represent the cuban people they represent. [CLS] the leadership is still taking control of the company, but the problem has not been resolved. [CLS] to tony fernanda santos, of eey investments board, it thought up life's 328 million – and one – pound tax loss of $ 86 million and projected earnings of $ 37 million next year to 2011. [CLS] next up is " best of belies, " the first mini – series. [CLS] cusuoga, the nationalist method often used to describe revenge attacks, is not experienced in most protests against a presence of the tigers in the island's sinhala territories. [CLS] [CLS]` |

To investigate the internal generation mechanism of XDLM, we visualize a single generation trajectory with a total budget of $T = 32$ steps and a fixed sequence length of 128 tokens. Table 8 details the evolution of the sequence at steps $t \in \{0, 1, 8, 16, 24, 32\}$. We utilize a color-coded schema to distinguish the specific transition types inherent to the hybrid noise process. **Green** denotes transitions from absorbing state (`[MASK]`) to a token (typical of Masked DLMs), **Blue** denotes token-to-token refinement (typical of Uniform DLMs), and **Red** denotes the re-masking operation (token back to `[MASK]`). This red transition is particular to XDLM, where the forward process is a mixture of absorbing and uniform noise, allowing the model to stochastically backtrack by rejecting previously generated tokens.

The process initializes at $t = 0$ with a mixture of masks and random tokens derived from uniform noise, such as "*coffin*," "*slippery*," and "*trumpets*." A critical dynamic is observed immediately at the transition to $t = 1$. The model actively rejects the majority of these random initializations. We observe extensive **Red** markings where tokens like "*coffin*" and "*trumpets*" are reverted to `[MASK]`. This demonstrates that XDLM is not strictly bound by its random initialization from uniform noise. Unlike standard iterative refinement models that might struggle to escape local minima induced by poor random seeds, XDLM possesses the capacity to identify low-probability tokens and "clean the slate" via re-masking. Simultaneously, valid anchor tokens begin to emerge (**Green**), such as "*shillings*" and "*blocked*," establishing an initial semantic direction.

As the generation progresses through the intermediate steps ($t = 1 \rightarrow t = 16$), the model engages in a dynamic interplay of generation and error correction. The presence of all three transition types indicates a non-monotonic search process. The model generates new candidates (**Green**) and refines existing ones (**Blue**), but crucially, it continues to utilize re-masking (**Red**) to prune inconsistent branches. For instance, tokens generated tentatively in early steps are re-masked when they fail

to align with the evolving global context.

By the final phase ($t = 24 \rightarrow t = 32$), the dynamics shift characteristically. The **Red** re-masking transitions disappear entirely, indicating structural convergence. The process focuses exclusively on filling the remaining gaps (**Green**) and performing fine-grained lexical substitutions (**Blue**). Notable refinements occur during this phase; for instance, the phrase "*life infancy*" at step 24 is successfully updated to "*was time*" at step 32, while "*from pound*" is refined into "*from pound-land*." This trajectory confirms that XDLM successfully anneals from a high-temperature exploration state, which is characterized by active deletion and re-generation, to a low-temperature refinement state, ensuring the final output is both globally coherent and locally precise.

*Table 8.* **Step-wise evolution of a generated sequence** ($T = 32$). Text colors indicate the transition dynamics inherent to the hybrid noise process: **Green** represents new tokens generated from masks; **Blue** represents lexical refinement; and **Red** highlights the re-masking operation where previously generated tokens are rejected and reverted to [MASK].

| Steps ($T$) | Generated Text Sample |
| --- | --- |
| $T = 0$ | [MASK] [MASK] faltered [MASK] [MASK] coffin [MASK] [MASK] [MASK] [MASK] och [MASK] [MASK] [MASK] [MASK] [MASK] [MASK] [MASK] [MASK] [MASK] [MASK] [MASK] [MASK] [MASK] [MASK] [MASK] [MASK] [MASK] [MASK] , [MASK] [MASK] [MASK] [MASK] [MASK] [MASK] [MASK] ##agawa [MASK] [MASK] [MASK] [MASK] slippery [MASK] [MASK] [MASK] [MASK] [MASK] [MASK] [MASK] [MASK] [MASK] [MASK] [MASK] [MASK] [MASK] harrington [MASK] [MASK] [MASK] [MASK] [MASK] [MASK] [MASK] [MASK] [MASK] trumpets surpassing [MASK] [MASK] [MASK] [MASK] [MASK] [MASK] [MASK] [MASK] [MASK] [MASK] [MASK] [MASK] [MASK] [MASK] [MASK] [MASK] [MASK] [MASK] [MASK] [MASK] [MASK] [MASK] [MASK] [MASK] [MASK] [MASK] [MASK] refuse [MASK] [MASK] [MASK] [MASK] [MASK] [MASK] [MASK] [MASK] ##was veteran jan [MASK] [MASK] [MASK] [MASK] [MASK] [MASK] [MASK] [MASK] [MASK] [MASK] [MASK] [MASK] [MASK] [MASK] [MASK] [MASK] [MASK] begs [MASK] [MASK] |
| $T = 1$ | [MASK] [MASK] faltered [MASK] [MASK] **[MASK]** [MASK] [MASK] [MASK] [MASK] **[MASK]** **##fo** [MASK] **is** [MASK] [MASK] **shillings** [MASK] [MASK] [MASK] [MASK] [MASK] [MASK] [MASK] [MASK] [MASK] [MASK] **1879** [MASK] **[MASK]** [MASK] [MASK] [MASK] **##anda** [MASK] **blocked** [MASK] ##agawa [MASK] [MASK] [MASK] [MASK] **[MASK]** [MASK] [MASK] [MASK] [MASK] [MASK] [MASK] [MASK] [MASK] [MASK] [MASK] [MASK] [MASK] [MASK] **[MASK]** [MASK] **engined** [MASK] [MASK] [MASK] **a** [MASK] [MASK] **##itive** **[MASK]** **[MASK]** [MASK] [MASK] [MASK] **mar** [MASK] [MASK] [MASK] [MASK] [MASK] [MASK] **distributing** [MASK] [MASK] [MASK] [MASK] [MASK] [MASK] [MASK] [MASK] [MASK] [MASK] [MASK] [MASK] [MASK] [MASK] [MASK] [MASK] [MASK] **[MASK]** [MASK] [MASK] [MASK] **tertiary** [MASK] [MASK] [MASK] [MASK] **[MASK]** **[MASK]** **[MASK]** [MASK] [MASK] [MASK] **strode** [MASK] [MASK] [MASK] [MASK] [MASK] [MASK] [MASK] **whatever** [MASK] **##rda** [MASK] [MASK] [MASK] **[MASK]** [MASK] [MASK] |
| $T = 8$ | [MASK] [MASK] **[MASK] to clear** [MASK] [MASK] [MASK] **[CLS]** [MASK] [MASK] **of** [MASK] **is** [MASK] **congo** **[MASK]** [MASK] [MASK] [MASK] [MASK] [MASK] **##rians** **santa** **,** [MASK] [MASK] **[MASK]** [MASK] [MASK] [MASK] **15** [MASK] ##anda [MASK] **[MASK]** [MASK] **[MASK]** [MASK] [MASK] **board had** [MASK] **##ffi** [MASK] [MASK] [MASK] **##illo** [MASK] **the** [MASK] [MASK] **[CLS]** [MASK] **hawaii** [MASK] [MASK] **the [MASK] 86 to** [MASK] **a** [MASK] **of process** [MASK] [MASK] [MASK] [MASK] **[CLS]** mar [MASK] [MASK] [MASK] [MASK] [MASK] [MASK] **,** [MASK] [MASK] [MASK] [MASK] [MASK] [MASK] [MASK] [MASK] [MASK] [MASK] **brazil** [MASK] [MASK] [MASK] [MASK] [MASK] **the** [MASK] [MASK] [MASK] **used** **[MASK]** [MASK] [MASK] **pakistani** [MASK] [MASK] [MASK] [MASK] **' s** [MASK] **[MASK] ##west** [MASK] [MASK] [MASK] [MASK] **in** [MASK] **[MASK] za** ##rda [MASK] [MASK] [MASK] **sees** [MASK] [MASK] |

| Steps ($T$) | Generated Text Sample (Continued) |
|---|---|
| $T = 16$ | [MASK] [MASK] [MASK] to clear [MASK] [MASK] [MASK] [CLS] [MASK] [MASK] of [MASK] is [MASK] congo [MASK] [MASK] [MASK] [MASK] [MASK] [MASK] **the** **[MASK]** , **which** [MASK] **resolved obscene .** **[CLS]** 15 [MASK] ##anda [MASK] [MASK] [MASK] [MASK] [MASK] [MASK] board had **convinced** ##ffi [MASK] **life** [MASK] **[MASK] take** the [MASK] [MASK] [CLS] [MASK] hawaii [MASK] [MASK] the **city** 86 to [MASK] a [MASK] of **new** [MASK] [MASK] **areas** [MASK] [CLS] mar [MASK] **vice** [MASK] [MASK] **le** [MASK] , [MASK] [MASK] [MASK] **afghan** [MASK] **to steal** [MASK] [MASK] [MASK] **[MASK] out** [MASK] – [MASK] [MASK] the **taliban has often** used **to** [MASK] **from** pakistani [MASK] [MASK] **not pakistan** ' s [MASK] [MASK] **though** [MASK] **of its** [MASK] in [MASK] [MASK] za ##rda **##ri said** [MASK] sees [MASK] [MASK] |
| $T = 24$ | [MASK] [MASK] [MASK] to clear [MASK] [MASK] . [CLS] **that** [MASK] of **neutrality** is [MASK] congo **force some post ##pone ##ments of** the [MASK] , which [MASK] resolved **[MASK]** . [CLS] 15 [MASK] **the telegraph that** [MASK] [MASK] **believed the** board had convinced **him** [MASK] life **infancy** [MASK] take the **train** [MASK] [CLS] **from pound** [MASK] , the city **plans** to [MASK] a [MASK] of new [MASK] [MASK] areas [MASK] [CLS] mar [MASK] **, according** [MASK] le **monde ,** " **is the** afghan **##s** to steal **or** [MASK] **and foot** out " – **–** [MASK] the taliban has often used to [MASK] from pakistani **taliban** [MASK] not pakistan ' s [MASK] [MASK] though [MASK] of its **friends** in **the** [MASK] za ##rda ##ri said **he** sees [MASK] [MASK] |
| $T = 32$ | **[CLS] me** , to clear **the flag** . [CLS] that **kind** of neutrality is **likely to** force some post ##pone ##ments of the **dispute** , which **was** resolved **wednesday** . [CLS] 15 **told** the telegraph that **he always** believed the board had convinced him **it was time to** take the train . [CLS] from pound **##land** , the city plans to **build** a **series** of new **retail residential** areas . [CLS] mar **##for** , according **to** le monde , " is the afghan ##s to steal or **take** and foot out " – – **money** the taliban has often used to **buy** from pakistani taliban , not pakistan ' s **.** **[CLS]** though **some** of its friends in the **west** za ##rda ##ri said he sees **k [CLS]** |

## F. ImageNet Sampling Case

We conduct a qualitative comparison of XDLM against the baseline models MDLM, GIDD, and UDLM on ImageNet-1K. The results are analyzed across two settings: unguided generation (Figure 5) and generation with slight classifier-free guidance (Figure 6). The visual evidence underscores XDLM's unique ability to combine the strengths of absorbing and uniform noise strategies while mitigating their individual weaknesses.

The unguided setting serves as a stress test for structural priors. Here, the limitations of MDLM and GIDD are most apparent. These models struggle to form coherent geometries, resulting in chaotic texture blobs (e.g., the flamingo and ladybug columns) rather than distinct objects. Conversely, UDLM, which relies solely on uniform noise, successfully captures global shapes but fails to generate high-frequency details, resulting in blurry, out-of-focus images.

XDLM distinguishes itself by balancing these two extremes. By integrating the structural rigidity of MDLM with the refinement flexibility of UDLM, XDLM achieves structural coherence without sacrificing sharpness. For instance, in the rocket launch example, XDLM is the only model that renders a clear vertical fuselage and distinct smoke plumes without guidance.

As illustrated in Figure 6, introducing a guidance scale of 2.0 improves general sample quality, yet the characteristic behaviors of the models persist. MDLM continues to suffer from a lack of detail refinement, often producing over-saturated textures with poor semantic structure. UDLM remains overly smooth; while the images are clean, they lack the crispness required for photorealism.

XDLM leverages the guidance most effectively to enhance detail. This is evident in the flock of flamingos. XDLM renders distinct, individual birds with sharp feathers, whereas MDLM merges them into a mass and UDLM blurs the boundaries. Similarly, in the food categories (strawberries and pizza), XDLM generates realistic surface textures and lighting reflections that are absent in the UDLM samples and distorted in the MDLM samples. This confirms that XDLM's hybrid noise

mechanism provides a superior foundation for high-fidelity generation.

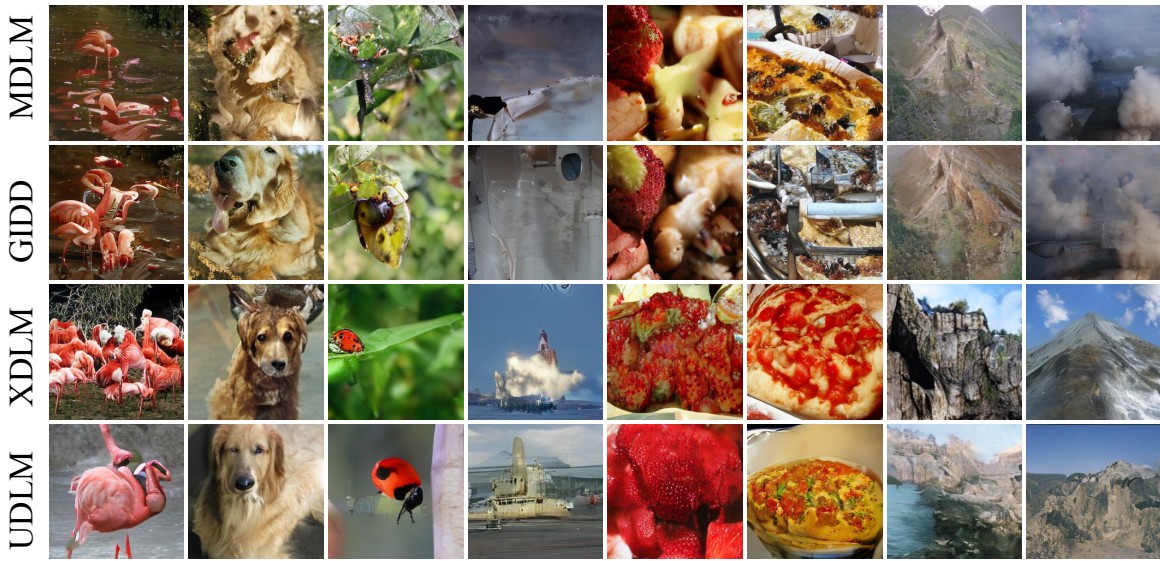

*Figure 5.* **Qualitative comparison of class-conditional generation on ImageNet-1K without CFG.** In the absence of guidance, baseline models struggle significantly: MDLM and GIDD produce chaotic artifacts with poor structural coherence, while UDLM yields recognizable but over-smoothed images. XDLM, by effectively balancing the characteristics of MDLM and UDLM, generates the most coherent and semantically correct samples (e.g., the distinct rocket structure and pizza toppings) even without guidance.

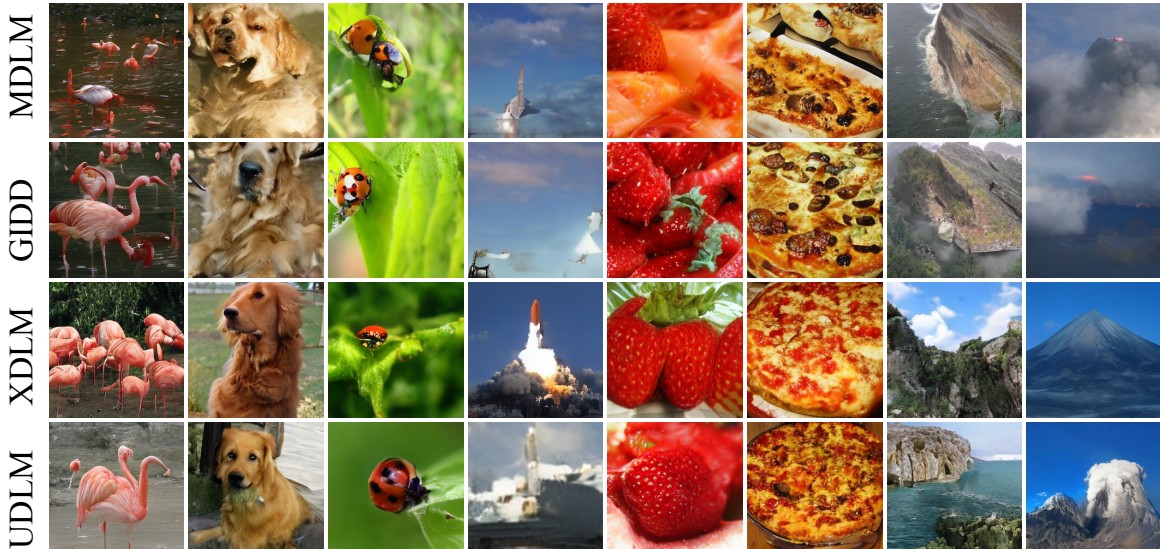

*Figure 6.* **Qualitative comparison of class-conditional generation on ImageNet-1K with a CFG scale of** 2.0. With the addition of guidance, all models improve, but quality disparities remain. MDLM and GIDD still exhibit texture distortion and "burn" artifacts. UDLM produces clean but blurry outputs, lacking fine-grained texture. XDLM demonstrates superior fidelity, combining sharp high-frequency details (seen in the strawberry and volcanoes) with accurate global structure.

We further analyze the sampling dynamics of XDLM ($k = 0.1$) over an 8-step generation process on ImageNet-1K with a CFG scale of 2.0, comparing it against MDLM, GIDD, and UDLM. The visual progression (Figure 7) highlights distinct behaviors in how each model handles noise and feature refinement.

A critical advantage of XDLM is its strong refinement capability, facilitated by the integration of uniform noise throughout

the sampling process. As observed in rows 3 and 4, both XDLM and UDLM begin generating key semantic features, specifically the dog's nose and eyes, at the very beginning of the generation cycle. In contrast, MDLM and GIDD adopt a more conservative approach. They exhibit little to no refinement of these specific features. Consequently, by the final step [8/8], MDLM and GIDD still fail to resolve the proper structure of the nose and eyes, leaving them distorted or missing. XDLM, however, establishes these features early and refines them continuously, resulting in an anatomically correct and aesthetically pleasing final output.

The models also differ significantly in their ability to generate fine textures, such as the dog's fur. Because MDLM lacks the ability to correct or refine previously generated tokens, the resulting fur texture appears rough and incoherent. While GIDD produces more delicate fur than MDLM, its performance is hindered by its reliance on time-variant noise, where the beneficial uniform noise mechanism only functions effectively during the very last step, limiting the potential for gradual improvement. Conversely, XDLM refines the fur texture consistently across all steps, ultimately producing the most realistic and high-fidelity texture among all evaluated models.

Crucially, the comparison demonstrates that relying exclusively on uniform noise, as seen in UDLM, is insufficient for optimal image quality. While UDLM shares XDLM's ability to locate features early, the final generated image is significantly more blurred. This lack of sharpness results in a loss of fine-grained details compared to XDLM, which successfully balances structural coherence with sharp, detailed textures.

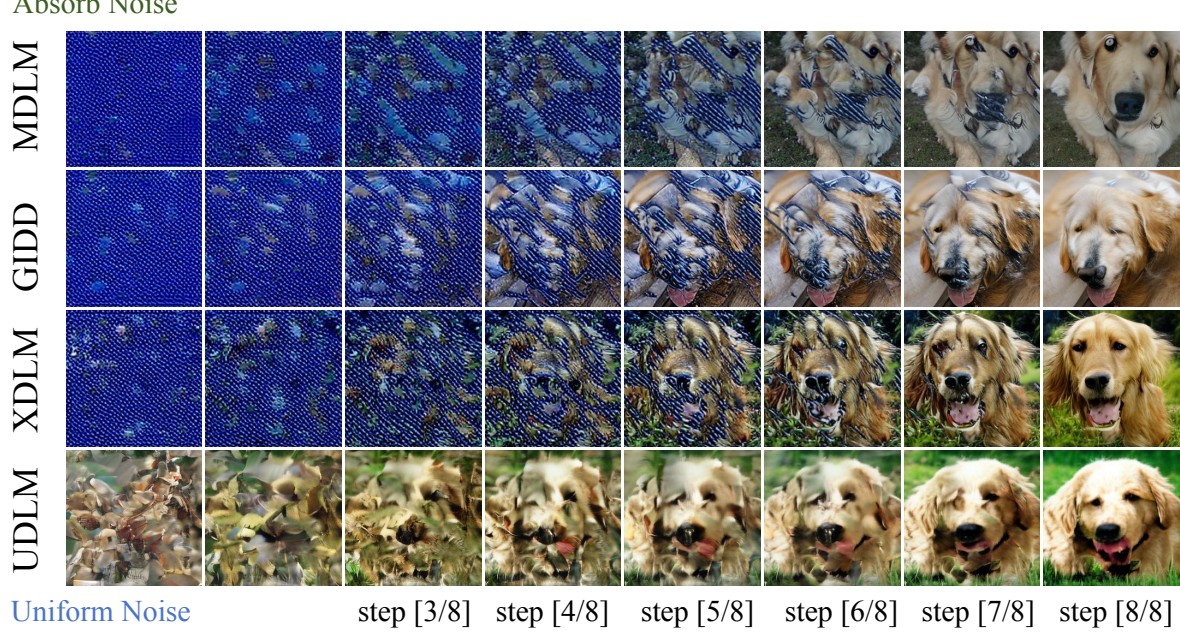

*Figure 7.* **Visual comparison of sampling dynamics over 8 steps on ImageNet-1K.** The figure contrasts the generation trajectories of MDLM, GIDD, XDLM ($k = 0.1$), and UDLM with CFG scale $= 2.0$. XDLM demonstrates superior structural coherence and detail refinement compared to baseline models, successfully transitioning from noise to a high-fidelity image.

## G. Detailed Configurations of Experiments

**Language Modeling** Following established methodologies (Sahoo et al., 2025; Schiff et al., 2024; Sahoo et al., 2024), we evaluate our model on standard benchmarks using the LM1B (Chelba et al., 2013) and OpenWebText (OWT) (Gokaslan & Cohen, 2019) datasets. For LM1B, we detokenize the dataset following (Lou et al., 2023; Sahoo et al., 2024; 2025) and evaluate with sequence packing following (Austin et al., 2021; Sahoo et al., 2025). Sequences are processed with a context length of 128 using the `bert-base-uncased` tokenizer (Devlin et al., 2019). For OWT, we use the GPT-2 tokenizer (Radford et al., 2019) with a context length of 1,024. Sequences are packed to a maximum length of 1,024 with an EOS token delimiter. As OWT lacks a standard validation split, we reserve the last 100k documents for validation, following (Sahoo et al., 2024).

Consistent with prior work (Sahoo et al., 2024; Schiff et al., 2024; Sahoo et al., 2025), we parameterize both XDLM and baselines using the modified Diffusion Transformer (DiT) architecture (Peebles & Xie, 2023) adapted from (Lou et al., 2023). All models employ 12 layers, 12 attention heads, a hidden dimension of 768, and a timestep embedding dimension of 128. The dropout rate is set to 0.1, and input/output word embeddings are untied. To ensure a fair comparison, we retain the AdaLN branch but fix the input to zero instead of using a log-linear scheduled time embedding. For the GIDD baseline, which is ported to our experimental codebase, we follow the default hyperparameter configuration specified in (von Rütte et al., 2025), including `loss_weighting=dynamic`, `min_loss_weight=0.0`, `max_loss_weight=2.0`, `p_uniform=0.0`, and `t_eps=`$10^{-4}$.

We train all models using AdamW ($\beta_1 = 0.9, \beta_2 = 0.999$, weight decay 0) with a global batch size of 512. The learning rate is warmed up to $3 \times 10^{-4}$ over the first 2,500 steps and held constant thereafter. Following prior work (Sahoo et al., 2024; 2025), we use an Exponential Moving Average (EMA) decay rate of 0.9999.

**Image Generation**    We evaluate on discretized CIFAR-10 (Krizhevsky et al., 2009) and ImageNet-1K (Deng et al., 2009).

On the ImageNet-1K dataset, We use the VQ-VAE tokenizer from LlamaGen (Sun et al., 2024), , which compresses images into sequences of length 256. The backbone architecture mirrors our language modeling transformer, augmented with class-conditional embeddings to enable class-conditional generation. Training hyperparameters follow the language modeling configuration, with the training duration set to 0.5M steps.

For CIFAR-10, we treat raw RGB pixel values as discrete tokens (with a vocabulary size of 256) and flatten the images into sequences of length $32 \times 32 \times 3$, following (Schiff et al., 2024). We employ a U-Net backbone (Ronneberger et al., 2015; Ho et al., 2020) with a discretized truncated logistic output distribution (Austin et al., 2021; Sahoo et al., 2024). Class conditioning is implemented by adding label embeddings to the timestep embeddings (Dhariwal & Nichol, 2021). The model is trained for 300k steps using AdamW ($\beta_1 = 0.9, \beta_2 = 0.999$, weight decay 0) with a batch size of 512. We use a learning rate warmup of 5,000 steps peaking at $2 \times 10^{-4}$ and maintain an Exponential Moving Average (EMA) rate of 0.9999.

**Large Language Model Tuning**    We scale XDLM to large language modeling by performing continual pretraining on LLaDA (Nie et al., 2026), a model originally trained via MDLM. We utilize a 10-billion-token subset of the FineWeb-Edu dataset (Penedo et al., 2024). The training configuration employs a sequence length of 4,096 and a global batch size of 512. The learning rate is warmed up to $3 \times 10^{-4}$ over 100 steps and held constant for the remaining 500 steps.

Unless otherwise specified, we set the mixing ratio to $k = 0.1$. All experiments were conducted on a node equipped with $8\times$ Nvidia H800 GPUs.

# H. Details of Zeroshot Capability of XDLM Trained on OWT

To comprehensively assess the generalization capabilities of XDLM, we evaluate zero-shot perplexity on seven external datasets following training on OWT. In this section, we provide the detailed experimental setup, an analysis of the hyperparameter sensitivity regarding the mixing ratio $k$, and a breakdown of the training dynamics.

Aligning with prior work (Sahoo et al., 2024; 2025), our evaluation suite includes the validation splits of AG News (Zhang et al., 2015), LAMBADA (Paperno et al., 2016), LM1B (Chelba et al., 2013), Penn Treebank (Marcus et al., 1993), Scientific Papers from ArXiv and PubMed (Cohan et al., 2018), and WikiText (Merity et al., 2017). The validation PPL is estimated via the negative Evidence Lower Bound (ELBO) with Monte Carlo sampling. We employ an evaluation batch size of 128 on a single GPU, and all text is pre-packed into a sequence length of 1024. To ensure a fair comparison, we adopt the identical sampling configuration used in (Sahoo et al., 2024; 2025).

We investigate the impact of the mixing ratio $k$ on model performance. Table 9 and Table 10 present the validation PPL on OWT and the zero-shot PPL on external benchmarks, respectively.

As shown in Table 9, XDLM exhibits a performance profile highly correlated with $k$. With smaller values (e.g., $k = 1 \times 10^{-3}$ and $k = 0.1$), XDLM achieves a PPL of 23.495 and 24.097, respectively, which is comparable to MDLM (23.321) and GIDD (23.136). Conversely, as $k$ increases towards 0.9, the performance degrades to 25.731, approaching the UDLM baseline (25.937).

This trend is further supported by the zero-shot benchmark results in Table 10. XDLM ($k = 0.1$) achieves an average PPL

of 54.110, performing similarly to MDLM (53.650) and significantly outperforming UDLM (59.574). This demonstrates that XDLM successfully retains the strong modeling capabilities of the masked diffusion paradigm.

*Table 9.* **Validation PPL on OWT after 1M training steps.** Performance is highly correlated with the mixing ratio $k$: lower values ($k \leq 0.1$) maintain parity with the strong MDLM baseline (23.321), while higher values ($k \rightarrow 0.9$) degrade towards the UDLM baseline (25.937).

| Model | MDLM | GIDD | XDLM(k=1e-3) | XDLM(k=0.1) | XDLM(k=0.5) | XDLM(k=0.9) | UDLM |
|---|---|---|---|---|---|---|---|
| **PPL** | 23.321 | 23.136 | 23.495 | 24.097 | 24.818 | 25.731 | 25.937 |

For completeness, we include additional baselines reported in Duo (Sahoo et al., 2025). In our experiments, UDLM is implemented with the Rao–Blackwellized NELBO introduced by Duo. We also report the vanilla UDLM baseline and Duo, where the latter further incorporates curriculum learning into UDLM; their results are adopted from (Sahoo et al., 2025). In addition, we compare XDLM with DiDi-Instruct (Zheng et al., 2025), a distillation-oriented approach that improves inference efficiency by aligning multi-step generation behavior with a few-step student model. However, its zero-shot perplexity suggests a potential trade-off in language understanding capability.

*Table 10.* **Zero-shot PPL evaluation on seven external benchmarks.** XDLM ($k = 0.1$) demonstrates robust generalization, achieving an average PPL (54.110) comparable to pure mask-based methods (MDLM/GIDD) and significantly outperforming UDLM (59.574). DiDi-Instruct[†] denotes results reported by (Zheng et al., 2025), which studies a distillation-based approach. The [‡] symbol denotes results reported by Duo (Sahoo et al., 2025). The UDLM-vanilla[‡] does not use the Rao–Blackwellized NELBO introduced in (Sahoo et al., 2025), whereas the UDLM row above is retrained with this objective. The best result for each dataset is highlighted in bold.

| Dataset | AG News | LAMBADA | LM1B-GPT2 | PTB | ArXiv | PubMed | WikiText | Average |
|---|---|---|---|---|---|---|---|---|
| MDLM | 61.374 | 47.967 | **65.629** | 89.049 | 37.457 | 41.981 | 32.093 | 53.650 |
| GIDD | **60.607** | 47.811 | 65.898 | **86.911** | 39.019 | 42.634 | **30.809** | 53.384 |
| XDLM(k=1e-3) | 62.247 | 45.982 | 65.944 | 89.575 | 38.294 | 42.206 | 31.809 | 53.722 |
| XDLM(k=0.1) | 62.768 | **45.608** | 68.229 | 90.796 | **37.232** | **41.391** | 32.748 | 54.110 |
| XDLM(k=0.5) | 67.352 | 47.966 | 71.252 | 91.683 | 38.460 | 43.373 | 33.570 | 56.236 |
| XDLM(k=0.9) | 71.791 | 49.094 | 74.116 | 97.266 | 39.658 | 44.701 | 35.175 | 58.829 |
| UDLM | 69.402 | 51.272 | 75.572 | 95.986 | 42.671 | 47.181 | 34.933 | 59.574 |
| UDLM-vanilla[‡] | 80.96 | 53.57 | 77.59 | 112.82 | 44.08 | 50.98 | 39.42 | 65.63 |
| Duo[‡] | 67.81 | 49.78 | 73.86 | 89.35 | 40.39 | 44.48 | 33.57 | 57.03 |
| DiDi-Instruct[†] | 78.36 | 53.62 | 80.58 | 107.03 | 45.35 | 47.56 | 35.20 | 63.96 |

Notably, the superior performance of XDLM is not merely an endpoint phenomenon but a consistent property observed throughout the training process. As illustrated in Figure 8, the PPL trajectory of XDLM closely aligns with that of MDLM, steadily decreasing from 64.85 to 54.11. In contrast, the UDLM curve remains notably higher across all training steps. Table 11 details the average zero-shot PPL at 100k-step intervals. The data confirms that XDLM ($k = 0.1$) maintains a learning dynamic nearly identical to MDLM from the early stages (100k steps) to convergence (1M steps), highlighting the inherent stability and efficiency of the proposed method.

*Table 11.* **Detailed evolution of Zero-Shot PPL throughout training.** Evaluated at 100k-step intervals, the data confirms that XDLM ($k = 0.1$) maintains a learning dynamic nearly identical to MDLM, highlighting the method's stability and efficiency from early training to convergence. The best results are bolded.

| Train Steps | 100k | 200k | 300k | 400k | 500k | 600k | 700k | 800k | 900k | 1M |
|---|---|---|---|---|---|---|---|---|---|---|
| MDLM | 64.209 | 59.315 | 57.411 | 56.129 | 55.434 | 54.943 | 54.466 | 54.201 | 53.839 | 53.650 |
| GIDD | **61.097** | **57.900** | **56.326** | **55.389** | **54.848** | **54.294** | **53.960** | **53.807** | **53.258** | **53.384** |
| XDLM(k=1e-3) | 63.748 | 59.097 | 57.319 | 56.374 | 55.576 | 54.937 | 54.577 | 54.336 | 53.977 | 53.722 |
| XDLM(k=0.1) | 64.847 | 60.187 | 58.253 | 57.201 | 56.288 | 55.797 | 55.271 | 54.872 | 54.500 | 54.110 |
| XDLM(k=0.5) | 68.074 | 62.524 | 60.489 | 59.353 | 58.396 | 57.881 | 57.437 | 56.976 | 56.530 | 56.236 |
| XDLM(k=0.9) | 70.264 | 65.384 | 63.147 | 61.758 | 60.977 | 60.263 | 59.723 | 59.438 | 58.915 | 58.829 |
| UDLM | 71.220 | 65.789 | 63.692 | 62.320 | 61.258 | 60.937 | 60.577 | 60.192 | 59.711 | 59.574 |

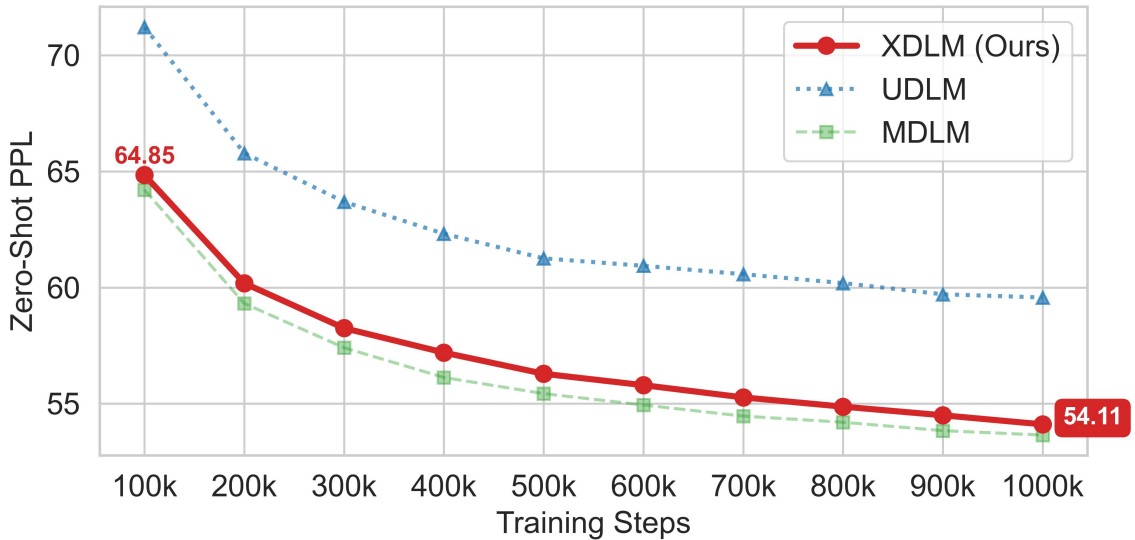

*Figure 8.* **Average Zero-Shot PPL trajectory throughout training.** Consistent with the final results, XDLM ($k = 0.1$) closely aligns with the MDLM baseline from early stages (64.85) to convergence (54.11), while the UDLM curve remains notably higher across all training steps.

## I. Detailed Language Generation Results

In this section, we present the comprehensive numerical results for the language generation experiments introduced in Section 4.2.

To systematically analyze the trade-off between the absorbing (MDLM-like) and uniform (UDLM-like) diffusion processes, we evaluate XDLM across varying mixing ratios $k \in \{10^{-3}, 0.1, 0.5, 0.9\}$. For generation, we adopt the *ancestral* sampling setting (Sahoo et al., 2024; Schiff et al., 2024; Sahoo et al., 2025), employing parallel token sampling from the estimated posterior via Gumbel noise. The resulting performance is evaluated based on sample quality, quantified by GPT-2 Large perplexity (PPL ↓), and diversity, measured by Token Entropy (↑).

Table 12 presents the comprehensive evaluation on the OWT dataset. The data quantitatively confirm that $k$ governs the model's performance across different numbers of sampling steps, effectively interpolating between the behaviors of UDLM and MDLM. In the few-step regime (e.g., 8 to 32 sampling steps), increasing $k$ significantly improves performance, aligning XDLM with the efficiency of UDLM. For instance, with only 8 sampling steps, XDLM ($k = 0.9$) yields a low perplexity of 189.750, which is highly competitive with UDLM (183.991) and vastly superior to the mask-based baselines, MDLM (711.382) and GIDD (654.841). Conversely, in the multi-step regime (e.g., 512 to 1024 sampling steps), retaining mask-like properties (lower $k$) remains advantageous for achieving high-fidelity generation. XDLM ($k = 0.1$) demonstrates an optimal balance. It dramatically improves upon MDLM when sampling budget is limited, while retaining the capacity to achieve

very low perplexity scores (52.609 at 1024 steps) when more sampling steps are available, all without sacrificing the diversity indicated by entropy.

*Table 12.* **Quality (PPL) vs. Diversity (Entropy) on OWT.** The results demonstrate that the optimal mixing ratio $k$ depends on the sampling budget. In the few-step regime (e.g., 8 steps), a higher $k$ is essential for efficiency, allowing XDLM to match the performance of UDLM. Conversely, in the multi-step regime, a lower $k$ becomes advantageous for achieving high fidelity. Notably, XDLM ($k = 0.1$) strikes an optimal balance, dramatically outperforming MDLM at low step counts while achieving superior perplexity at high step counts. The best results are bolded.

| Model | step 8 | step 16 | step 32 | step 64 | step 128 | step 256 | step 512 | step 1024 |
|---|---|---|---|---|---|---|---|---|
| | | | | *Perplexity* ↓ | | | | |
| MDLM | 711.382 | 288.157 | 156.576 | 106.098 | 80.198 | 64.376 | 51.756 | 41.545 |
| GIDD | 654.841 | 273.829 | 151.331 | 102.558 | 77.548 | **62.077** | **49.822** | **39.981** |
| XDLM(k=1e-3) | 613.647 | 253.897 | 146.567 | 104.090 | 80.301 | 65.245 | 52.650 | 42.797 |
| XDLM(k=0.1) | 301.640 | 161.724 | 114.795 | 94.837 | 83.205 | 73.824 | 63.888 | 52.609 |
| XDLM(k=0.5) | 218.572 | 127.335 | 98.817 | 86.956 | 81.051 | 77.155 | 73.503 | 66.136 |
| XDLM(k=0.9) | 189.750 | 117.234 | 93.239 | 84.068 | 80.614 | 79.456 | 78.707 | 78.238 |
| UDLM | **183.991** | **111.237** | **88.632** | **79.509** | **76.611** | 74.019 | 73.345 | 106.666 |
| | | | | *Entropy* ↑ | | | | |
| MDLM | **5.872** | **5.767** | **5.687** | **5.614** | 5.543 | 5.472 | 5.387 | 5.287 |
| GIDD | 5.832 | 5.748 | 5.675 | 5.605 | 5.534 | 5.462 | 5.370 | 5.269 |
| XDLM(k=1e-3) | 5.837 | 5.746 | 5.673 | 5.613 | 5.546 | 5.475 | 5.393 | 5.298 |
| XDLM(k=0.1) | 5.662 | 5.630 | 5.600 | 5.573 | **5.548** | 5.516 | 5.467 | 5.390 |
| XDLM(k=0.5) | 5.574 | 5.565 | 5.555 | 5.547 | 5.538 | 5.525 | 5.514 | 5.476 |
| XDLM(k=0.9) | 5.546 | 5.547 | 5.548 | 5.548 | **5.548** | **5.547** | **5.544** | 5.542 |
| UDLM | 5.536 | 5.539 | 5.543 | 5.535 | 5.536 | 5.531 | 5.524 | **5.769** |

*Table 13.* **Quality (PPL) vs. Diversity (Entropy) on LM1B.** Corroborating the OWT findings, a higher $k$ proves dominant when the sampling budget is strictly limited. At 4 steps, XDLM ($k = 0.5$) effectively matches UDLM. Across the full spectrum of sampling steps, XDLM bridges the gap between the two paradigms, capturing the efficiency of uniform noise for small budgets while preserving the high-quality potential of masking for larger budgets. The best results are bolded.

| Model | step 4 | step 8 | step 16 | step 32 | step 64 | step 128 |
|---|---|---|---|---|---|---|
| | | | *Perplexity* ↓ | | | |
| MDLM | 377.177 | 215.579 | 152.939 | 124.731 | 112.049 | 102.142 |
| GIDD | 392.002 | 215.873 | 151.824 | 124.014 | 110.247 | 101.211 |
| XDLM(k=1e-3) | 343.246 | 199.457 | 146.612 | 124.444 | 111.960 | 103.544 |
| XDLM(k=0.1) | 246.134 | 155.487 | 123.129 | 113.006 | 105.509 | 101.983 |
| XDLM(k=0.5) | 232.702 | 141.121 | 116.063 | 106.182 | 103.419 | 100.754 |
| UDLM | **232.403** | **136.914** | **110.575** | **102.072** | **98.571** | **96.385** |
| | | | *Entropy* ↑ | | | |
| MDLM | **4.405** | **4.379** | **4.366** | 4.357 | 4.353 | 4.347 |
| GIDD | 4.401 | 4.373 | 4.361 | **4.359** | **4.354** | 4.348 |
| XDLM(k=1e-3) | 4.382 | 4.363 | 4.357 | 4.358 | **4.354** | **4.351** |
| XDLM(k=0.1) | 4.326 | 4.329 | 4.334 | 4.342 | 4.345 | 4.346 |
| XDLM(k=0.5) | 4.309 | 4.315 | 4.325 | 4.335 | 4.340 | 4.342 |
| UDLM | 4.303 | 4.305 | 4.320 | 4.327 | 4.334 | 4.336 |

These findings are corroborated by the results on the LM1B dataset, detailed in Table 13. We observe a consistent trend where a higher $k$ dominates when the number of sampling steps is small. At 4 sampling steps, XDLM ($k = 0.5$) achieves

a perplexity of 232.702, effectively matching UDLM (232.403) and significantly outperforming MDLM (377.177). As the number of sampling steps increases, the performance gaps narrow, yet XDLM remains robust across the spectrum. Collectively, these results demonstrate that by tuning $k$, XDLM bridges the gap between the two paradigms: it captures the efficiency of uniform noise at few sampling steps while preserving the high-quality potential of masking at larger steps.

## J. Detailed Image Generation Results

We present an extended analysis of XDLM's image generation capabilities on ImageNet-1K and CIFAR-10, specifically examining the impact of the mixing ratio parameter $k$. As defined in our methodology, $k$ governs the balance between the mask-based and uniform-noise processes. A higher $k$ increases the proportion of uniform noise (effectively approaching the UDLM setting), while a lower $k$ retains more mask-like properties. The results in Tables 14, 15, 16, and 17 empirically validate the utility of this interpolation. We observe two key behaviors: first, a consistent convergence of XDLM towards UDLM behavior as $k$ increases; and second, the fact that the optimal performance often lies within the interpolated regime ($0 < k < 1$) rather than at the extremes (i.e., pure MDLM or UDLM).

Tables 14 and 15 detail the performance on ImageNet-1K, providing compelling evidence that the optimal $k$ varies depending on the generation setting. In the standard conditioning regime (Table 14), a balanced mix is preferred: XDLM with $k = 0.5$ achieves an FID of 23.417 at 16 steps, significantly outperforming both the low-mixing variant ($k = 10^{-3}$) and the specialized UDLM baseline (which achieves 26.242). However, when Classifier-Free Guidance (CFG) is applied (Table 15), the optimal operating point shifts. Here, $k = 0.1$ emerges as the "sweet spot," achieving an FID of 8.625, which surpasses both pure UDLM (8.980) and the $k = 0.5$ variant (8.790). These findings highlight the advantage of XDLM. By exploring the interpolation space between masking and uniform noise, one can identify configurations that outperform both individual baselines.

*Table 14.* **Comparison of FID (↓) and IS (↑) on ImageNet-1K.** For image generation, XDLM demonstrates competitive capabilities, achieving performance highly comparable to the specialized UDLM model. Notably, XDLM outperforms all baselines, including UDLM, at 16 generation steps, confirming its robust performance beyond the text domain. The best results are bolded.

| Model | FID↓ | | | | IS↑ | | | |
|---|---|---|---|---|---|---|---|---|
| | step 4 | step 8 | step 16 | step 32 | step 4 | step 8 | step 16 | step 32 |
| MDLM | 80.752 | 47.732 | 28.785 | **18.928** | 16.287 | 29.178 | 44.656 | **57.803** |
| GIDD | 86.842 | 54.933 | 35.403 | 24.588 | 14.559 | 24.297 | 35.698 | 46.376 |
| XDLM(k=1e-3) | 81.992 | 52.590 | 35.046 | 25.637 | 15.597 | 24.813 | 35.097 | 44.032 |
| XDLM(k=0.1) | 54.085 | 34.109 | 25.774 | 22.187 | 24.829 | 36.964 | 43.903 | 48.118 |
| XDLM(k=0.5) | **45.808** | **29.362** | 23.417 | 21.486 | **29.310** | **40.482** | **46.620** | 48.911 |
| UDLM | 49.861 | 30.144 | 26.242 | 25.661 | 27.049 | 38.832 | 41.801 | 41.850 |

*Table 15.* **Performance on ImageNet-1K with CFG (scale $= 2.0$).** The application of guidance shifts the optimal operating point. Here, $k = 0.1$ emerges as the "sweet spot," achieving an FID of 8.625 at 16 steps. This configuration surpasses both the pure UDLM baseline (8.980) and the higher-mixing variant ($k = 0.5$), demonstrating the importance of tuning $k$ for specific guidance regimes. The best results are bolded.

| Model | FID↓ | | | | IS↑ | | | |
|---|---|---|---|---|---|---|---|---|
| | step 4 | step 8 | step 16 | step 32 | step 4 | step 8 | step 16 | step 32 |
| MDLM | 33.468 | 11.144 | **6.725** | 7.485 | 54.740 | 119.150 | **172.664** | **203.722** |
| GIDD | 41.000 | 15.151 | 7.076 | **6.314** | 44.084 | 95.789 | 148.148 | 181.017 |
| XDLM(k=1e-3) | 32.876 | 12.872 | 7.612 | 7.588 | 53.351 | 102.601 | 144.019 | 167.660 |
| XDLM(k=0.1) | 13.550 | **8.956** | 8.625 | 8.914 | 107.403 | **148.723** | 165.916 | 171.046 |
| XDLM(k=0.5) | **11.945** | 9.038 | 8.790 | 8.939 | **114.135** | 145.718 | 156.339 | 159.829 |
| UDLM | 14.055 | 9.718 | 8.980 | 8.650 | 97.859 | 123.582 | 132.099 | 132.108 |

Tables 16 and 17 further illustrate the behavioral shift controlled by $k$ on CIFAR-10. On this benchmark, where UDLM demonstrates a distinct advantage, a clear monotonic improvement in generation quality is observed as $k$ increases. For

instance, under standard conditioning, increasing $k$ from $10^{-3}$ to $0.5$ improves the 32-step FID from 164.040 to 56.299, substantially closing the gap with UDLM (41.027). A similar trend holds with CFG, where $k = 0.5$ approaches the UDLM benchmark. Collectively, these results confirm that XDLM successfully bridges the gap between domains. Crucially, these findings demonstrate that no single fixed $k$ is optimal for all scenarios; instead, the flexibility to tune $k$ within the interval $(0, 1)$ is essential for maximizing performance across different datasets and guidance regimes.

*Table 16.* **Class-conditional generation on CIFAR-10.** On this benchmark, where UDLM holds a distinct advantage, we observe a clear monotonic improvement in XDLM quality as $k$ increases. Increasing $k$ from $10^{-3}$ to $0.5$ dramatically closes the performance gap with UDLM (e.g., improving the 32-step FID from 164.040 to 56.299), confirming that higher $k$ effectively aligns XDLM with uniform noise dynamics. The best results are bolded.

| Model | FID↓ | | | IS↑ | | |
|---|---|---|---|---|---|---|
| | step 32 | step 128 | step 512 | step 32 | step 128 | step 512 |
| MDLM | 211.983 | 69.151 | 33.961 | 2.685 | 5.615 | 6.801 |
| GIDD | 193.487 | 72.579 | 41.809 | 2.737 | 5.176 | 6.209 |
| XDLM(k=1e-3) | 164.040 | 50.638 | 29.079 | 3.427 | 6.223 | 7.079 |
| XDLM(k=0.1) | 77.985 | 48.702 | 43.210 | 5.316 | 6.284 | 6.513 |
| XDLM(k=0.5) | 56.299 | 37.307 | 33.030 | 6.077 | 6.801 | 6.956 |
| UDLM | **41.027** | **27.822** | **25.144** | **6.892** | **7.345** | **7.372** |

*Table 17.* **Performance of XDLM and baselines on image generation with CFG scale** $= 2.0$ **on CIFAR-10.** All mask-based methods (MDLM, GIDD, XDLM) show significant improvement when CFG is applied. The basic trend follows the results without CFG. The best results are bolded.

| Model | FID↓ | | | IS↑ | | |
|---|---|---|---|---|---|---|
| | step 32 | step 128 | step 512 | step 32 | step 128 | step 512 |
| MDLM | 181.453 | 51.113 | 22.762 | 3.329 | 6.969 | 8.078 |
| GIDD | 179.916 | 60.443 | 30.812 | 3.054 | 6.124 | 7.216 |
| XDLM(k=1e-3) | 130.106 | 35.980 | 19.877 | 4.559 | 7.575 | 8.186 |
| XDLM(k=0.1) | 59.600 | 36.889 | 32.444 | 6.699 | 7.596 | 7.670 |
| XDLM(k=0.5) | 40.850 | 27.445 | 24.445 | 7.560 | 7.881 | 7.893 |
| UDLM | **30.328** | **20.720** | **18.592** | **8.098** | **8.392** | **8.433** |

## K. Detailed LLaDA Continual Pretraining Results

To empirically validate the efficacy of the XDLM objective within the LLaDA framework, we conducted a continual pre-training experiment utilizing the LLaDA 8B Base checkpoints. We integrated the XDLM modeling definitions into the `transformers` codebase and trained on a 10 billion token sample of the FineWeb-Edu dataset (Penedo et al., 2024). The training process spanned 600 steps with a global batch size of 512 and a sequence length of 4,096. The learning rate was warmed up to $3 \times 10^{-4}$ over the first 100 steps and held constant for the subsequent 500 steps, with the XDLM mixing ratio set to 0.1. All experiments were conducted on a compute node equipped with $8\times$ Nvidia H800 GPUs. To rigorously attribute performance gains to the architectural paradigm rather than confounding variables, we established two robust baselines: LLaDA-MDLM, a control condition where the model is continually pre-trained for the same duration using the standard Masked Diffusion Language Model objective; and LLaDA-XDLM-infer, an ablation where the XDLM sampling strategy is applied to the base model without further training.

To deploy XDLM sampling within this architecture, we extended the standard low-confidence remasking strategy used in LLaDA. Algorithm 3 details the generation logic, highlighting the novel contributions in the conditional block controlled by the mixing ratio $k$. While standard LLaDA sampling operates via a monotonic reduction of uncertainty, prioritizing only the filling of existing [MASK] tokens, our implementation introduces a *refinement branch*. When $k > 0.0$, the algorithm calculates a `refine_priority` by identifying tokens that are currently unmasked but for which the model predicts a conflicting token ID with a confidence score exceeding the current token's confidence. These tokens are flagged in `update_mask` alongside the standard empty masks. This mechanism fundamentally transforms the generation from a

purely additive process into a dynamic error-correction loop, allowing the model to stochastically "change its mind" and revise previous outputs during the denoising trajectory.

---

**Algorithm 3** Generation Strategy of LLaDA-XDLM

```python
def generate(steps, seq_length, mask_id, topk_absorb, topk_uniform, k=0.1):
    """
    Args:
      steps (int): Total number of sampling steps.
      seq_length (int): Target sequence length to generate.
      mask_id (int): Vocabulary index of the [MASK] token.
      topk_absorb (list[int]): Schedule for the number of tokens to decode at each step.
      topk_uniform (list[int]): Schedule for the number of tokens to refine at each step.
      k (float): The mixing ratio.
    Returns:
      token_ids (torch.Tensor): The generated sequence of token IDs with shape (seq_length,).
    """
    token_ids = torch.full((seq_length,), fill_value=mask_id, dtype=torch.long)
    for i in range(steps):
        is_masked = (token_ids == mask_id)
        logits = model(token_ids).logits
        logits_with_noise = add_gumbel_noise(logits)
        pred_tokens = torch.argmax(logits_with_noise, dim=-1)
        probs = torch.softmax(logits, dim=-1)
        pred_confidence = torch.squeeze(torch.gather(probs, dim=-1, index=torch.unsqueeze(pred_tokens, -1)), -1)
        mask_priority = torch.where(is_masked, pred_confidence, -np.inf)

        if k > 0.0:
            current_confidence = torch.squeeze(torch.gather(probs, dim=-1, index=torch.unsqueeze(token_ids, -1)), -1)
            can_be_refined = (~is_masked) & (pred_tokens != token_ids) & (pred_confidence >= current_confidence)
            refine_priority = torch.where(can_be_refined, pred_confidence, -np.inf)

        update_mask = torch.zeros_like(pred_tokens, dtype=torch.bool)
        _, top_mask_indices = torch.topk(mask_priority, k=topk_absorb[i])
        update_mask[top_mask_indices] = True

        if k > 0.0:
            _, top_refine_indices = torch.topk(refine_priority, k=topk_uniform[i])
            update_mask[top_refine_indices] = True

        token_ids[update_mask] = pred_tokens[update_mask]

    return token_ids
```

---

The empirical results, summarized in Table 18, demonstrate the superiority of this formulation across reasoning (GSM8K, MATH, BBH) and code generation (HumanEval, MBPP) benchmarks, particularly in low-compute regimes. At 32 sampling steps, LLaDA-XDLM significantly outperforms all baselines. The most distinct advantage is observed in the MBPP code generation task, where LLaDA-XDLM achieves a score of 15.00, effectively doubling the performance of the base LLaDA (6.80) and the continued MDLM control (4.40). A granular analysis of the MBPP failure modes reveals that this improvement is driven by a fundamental enhancement in generative fidelity; LLaDA-XDLM substantially reduces the incidence of "Failed" cases, defined as non-compilable or syntactically invalid code, from 429 (LLaDA) to 304. By successfully converting these previously unproductive attempts into "Passed" outputs (75 vs. 34), the model confirms that the XDLM objective, combined with the refinement sampling strategy, provides the necessary structural coherence to self-correct syntactic errors that would otherwise lead to compilation failures. As sampling steps increase to 64, 128, and 256 (the total generation length), LLaDA-XDLM maintains a competitive edge, validating that the performance gains stem directly from the internal representations learned during XDLM training rather than being an artifact of the sampling algorithm alone.

Notably, we observe that the continual pretraining baseline (LLaDA-MDLM) exhibits a performance decline compared to the original LLaDA, likely due to the distribution shift between the tuning data and the original pre-training corpus. Consequently, the superior performance of LLaDA-XDLM over LLaDA-MDLM confirms that our gains stem from the effectiveness of the XDLM formulation rather than merely the additional 600 training steps.

Furthermore, LLaDA-XDLM significantly outperforms the ablation baseline LLaDA-XDLM-infer, demonstrating that the improvement is intrinsic to the learned model rather than being an artifact of sampling heuristic modifications.

*Table 18.* **Evaluation of LLaDA-XDLM on reasoning and code generation benchmarks.** We compare the proposed method with the base LLaDA, an inference-only ablation (LLaDA-XDLM-infer), and an MDLM control (LLaDA-MDLM) across varying sampling steps. MBPP scores are decomposed into failure modes to illustrate structural correctness. The best results are bolded.

| Dataset | BBH | GSM8K | MATH | HumanEval | MBPP | | | |
| | | | | | Score | Failed | Wrong | Pass |
|---|---|---|---|---|---|---|---|---|
| **32 steps / 256 tokens** | | | | | | | | |
| LLaDA | 42.68 | 24.49 | 4.80 | 5.49 | 6.80 | 429 | 37 | 34 |
| LLaDA-XDLM-infer | 41.74 | 24.26 | **4.86** | 1.22 | 5.40 | 438 | 35 | 27 |
| LLaDA-MDLM | **43.21** | 24.41 | 2.70 | 6.71 | 4.40 | 447 | 31 | 22 |
| LLaDA-XDLM | 42.76 | **29.26** | 4.72 | **10.98** | **15.00** | 304 | 121 | 75 |
| **64 steps / 256 tokens** | | | | | | | | |
| LLaDA | 47.16 | 57.16 | 16.10 | 12.80 | 17.80 | 328 | 83 | 89 |
| LLaDA-XDLM-infer | 46.76 | 53.75 | 16.12 | 4.88 | 17.60 | 328 | 84 | 88 |
| LLaDA-MDLM | **48.32** | 54.97 | 13.42 | 14.63 | 13.40 | 348 | 84 | 67 |
| LLaDA-XDLM | 45.61 | **57.85** | **16.48** | **15.85** | **23.60** | 197 | 182 | 118 |
| **128 steps / 256 tokens** | | | | | | | | |
| LLaDA | 47.49 | 67.93 | 26.12 | 24.39 | 27.60 | 161 | 199 | 138 |
| LLaDA-XDLM-infer | 47.89 | 66.41 | **26.70** | 8.54 | 27.40 | 187 | 174 | 137 |
| LLaDA-MDLM | **48.95** | **68.61** | 24.34 | 25.00 | 30.00 | 169 | 178 | 150 |
| LLaDA-XDLM | 46.41 | 68.01 | 25.06 | **25.00** | **30.80** | 130 | 215 | 154 |
| **256 steps / 256 tokens** | | | | | | | | |
| LLaDA | 47.77 | 72.25 | **29.98** | **33.54** | **40.40** | 48 | 248 | 202 |
| LLaDA-XDLM-infer | 47.93 | 71.34 | 29.34 | 6.71 | 37.00 | 74 | 238 | 185 |
| LLaDA-MDLM | **49.84** | 72.93 | 29.70 | 32.32 | 37.60 | 51 | 260 | 188 |
| LLaDA-XDLM | 46.07 | **73.16** | 29.22 | 31.71 | 34.60 | 90 | 236 | 173 |

## L. Detailed Computational Efficiency Analysis

We evaluate the computational efficiency of the proposed XDLM against baseline models by measuring speed and memory consumption across three distinct scenarios: the *forward* case (representing NELBO calculation for perplexity estimation), the *forward-backward* case (used in training), and the *sample* case (used for generation). For the *forward* and *forward-backward* evaluations, we utilize randomly generated sequences with a batch size of 32 and a sequence length of 1024 to simulate real data processing. To ensure robust measurements, models are warmed up for 10 iterations, and we report the mean value calculated over the subsequent 100 iterations. For the *sample* scenario, we employ standard ancestral sampling to generate sequences of length 1024 over 32 steps. The evaluation batch size is set to 32. Models are warmed up for 1 iteration, and we report the mean value calculated over the subsequent 10 iterations.

As detailed in Table 19, MDLM exhibits the highest throughput and lowest memory cost across all settings, which is expected due to the simplicity of its absorbing noise kernel. However, among models incorporating uniform noise distributions, XDLM achieves the highest throughput across the *forward*, *forward-backward*, and *sample* regimes. This performance is directly attributed to our scalar-reformulated efficient sampling and training strategy, which circumvents expensive matrix operations. In contrast, while UDLM maintains the third-highest throughput in *forward* and *forward-backward* modes, its sampling throughput is the lowest among the comparison group. Conversely, GIDD achieves the third-highest sampling throughput but underperforms in NELBO calculation, reaching only half the *forward* throughput of XDLM. This trend is mirrored in memory consumption. While MDLM is the most lightweight overall, XDLM ranks first among methods whose noise kernels include a uniform noise component, effectively avoiding the high memory overhead observed in GIDD and UDLM.

*Table 19.* **Computational Efficiency Comparison.** We report throughput (tokens/s, ↑) and peak memory usage (GB, ↓) for MDLM, GIDD, XDLM, and UDLM. The metrics cover three scenarios: *forward* (inference/perplexity estimation), *forward-backward* (training), and *sample* (generation). Leveraging the scalar-reformulated strategy, XDLM achieves highly competitive throughput and memory efficiency, outperforming other baselines incorporating uniform noise kernels (GIDD, UDLM) in most metrics. The best results among models incorporating uniform noise kernels are bolded.

| Model | Throughput (token/s) ↑ | | | Memory (GB) ↓ | | |
| --- | --- | --- | --- | --- | --- | --- |
| | forward | forward-backward | sample | forward | forward-backward | sample |
| MDLM | 424294 | 141990 | 8789 | 6.285 | 35.354 | 18.848 |
| GIDD | 199516 | 95395 | 6336 | 25.131 | 51.060 | 40.856 |
| XDLM (k=0.1) | **396398** | 137372 | **7108** | **18.850** | **41.634** | **31.414** |
| UDLM | 370952 | **142276** | 2882 | **18.850** | 44.777 | 59.683 |

## M. Detailed Training Dynamics

In this section, we provide the comprehensive numerical data corresponding to the training dynamics analysis discussed in Section 4.3 and visualized in Figure 4. These tables supplement the visual analysis by providing exact metric values for Perplexity, Entropy, FID, and IS across the training trajectory.

Table 20 details the training dynamics on the LM1B benchmark over the course of 1M training steps. The models are evaluated using a fixed budget of 128 sampling steps. As indicated in the main text, while MDLM exhibits superior performance in the initial phase, UDLM demonstrates strong scaling in later training stages, achieving the lowest perplexity among baselines at 1M steps (96.385). Furthermore, the table details the impact of varying the mixing ratio $k$ within our proposed XDLM method.

Complementing the text generation results, Table 21 presents the quantitative training dynamics for image generation on ImageNet-1K. Evaluations are conducted over 500k training steps using a fixed budget of 16 sampling steps. The numerical data corroborates the visual trends: XDLM establishes and maintains a substantial performance advantage. Notably, XDLM with $k = 0.5$ achieves the lowest reported FID of 23.417 and the highest IS of 46.620 at the final checkpoint, consistently outperforming the MDLM and GIDD baselines. While UDLM starts with a higher FID, it converges rapidly to a competitive score (26.242). However, XDLM remains the leading model throughout the majority of the training phases.

*Table 20.* **Quantitative training dynamics on the LM1B benchmark, supplementing the visual analysis in Figure 4.** We report Perplexity (↓) and Entropy (↑) over 1M training steps, utilizing a fixed budget of 128 sampling steps. The table details the impact of varying the mixing ratio $k$ in XDLM compared to MDLM, GIDD, and UDLM baselines. The best results are bolded.

| Train Steps | 100k | 200k | 300k | 400k | 500k | 600k | 700k | 800k | 900k | 1M |
| --- | --- | --- | --- | --- | --- | --- | --- | --- | --- | --- |
| | | | | | *Perplexity* ↓ | | | | | |
| MDLM | 113.218 | 107.864 | 106.735 | 105.839 | 104.594 | 104.106 | 103.559 | 103.174 | 102.769 | 102.142 |
| GIDD | **107.366** | **103.473** | **103.386** | **101.854** | 102.367 | 101.745 | 101.904 | 101.957 | 101.490 | 101.211 |
| XDLM(k=1e-3) | 112.628 | 108.381 | 106.406 | 104.913 | 105.156 | 103.817 | 103.624 | 102.731 | 103.638 | 103.544 |
| XDLM(k=0.1) | 119.632 | 110.803 | 109.017 | 106.265 | 105.428 | 104.572 | 103.645 | 103.348 | 102.223 | 101.983 |
| XDLM(k=0.5) | 117.846 | 111.024 | 108.130 | 105.955 | 104.953 | 103.048 | 102.464 | 101.040 | 102.015 | 100.754 |
| UDLM | 116.021 | 107.800 | 104.508 | 102.466 | **101.480** | **99.958** | **99.660** | **98.147** | **97.503** | **96.385** |
| | | | | | *Entropy* ↑ | | | | | |
| MDLM | 4.350 | 4.348 | 4.350 | 4.347 | 4.349 | 4.349 | 4.348 | 4.349 | 4.349 | 4.347 |
| GIDD | 4.352 | **4.350** | **4.351** | **4.349** | **4.351** | **4.349** | **4.350** | **4.350** | **4.351** | 4.348 |
| XDLM(k=1e-3) | 4.351 | 4.347 | 4.349 | 4.347 | 4.350 | 4.347 | 4.347 | 4.349 | **4.351** | **4.351** |
| XDLM(k=0.1) | **4.353** | 4.347 | 4.347 | 4.344 | 4.344 | 4.344 | 4.346 | 4.344 | 4.344 | 4.346 |
| XDLM(k=0.5) | 4.350 | 4.346 | 4.345 | 4.344 | 4.342 | 4.343 | 4.342 | 4.343 | 4.344 | 4.342 |
| UDLM | 4.349 | 4.343 | 4.341 | 4.340 | 4.338 | 4.336 | 4.338 | 4.336 | 4.335 | 4.336 |

*Table 21.* **Quantitative training dynamics on ImageNet-1K generation**, corresponding to the curves in Figure 4. We report FID (↓) and IS (↑) over 500k training steps, utilizing a fixed budget of 16 sampling steps. The numerical data corroborates the visual trends: XDLM establishes and maintains a decisive performance advantage. The best results are bolded.

| Train Steps | FID↓ | | | | | IS↑ | | | | |
|---|---|---|---|---|---|---|---|---|---|---|
| | **100k** | **200k** | **300k** | **400k** | **500k** | **100k** | **200k** | **300k** | **400k** | **500k** |
| MDLM | 40.060 | 33.723 | 31.090 | 29.436 | 28.785 | **28.978** | **36.136** | 39.804 | 42.893 | 44.656 |
| GIDD | 42.152 | 36.878 | 35.200 | 35.677 | 35.403 | 27.337 | 33.042 | 35.544 | 35.173 | 35.698 |
| XDLM(k=1e-3) | 45.381 | 38.967 | 36.061 | 36.287 | 35.046 | 24.967 | 30.512 | 33.948 | 34.031 | 35.097 |
| XDLM(k=0.1) | 38.358 | 31.464 | 27.524 | 26.853 | 25.774 | 27.949 | 35.839 | **41.150** | 42.222 | 43.903 |
| XDLM(k=0.5) | **37.567** | **30.098** | **26.687** | **23.790** | **23.417** | 27.854 | 35.682 | 40.724 | **45.976** | **46.620** |
| UDLM | 43.316 | 34.525 | 30.381 | 28.086 | 26.242 | 23.619 | 31.188 | 35.963 | 38.588 | 41.801 |

