# OpenReview forum: "Balancing Understanding and Generation in Discrete Diffusion Models"
_ICML.cc/2026/Conference — ICML 2026 spotlight_

### Official Review · Reviewer_2HPG · 2026-02-23

**Soundness:** 3
**Presentation:** 3
**Significance:** 2
**Originality:** 3
**Overall Recommendation:** 3
**Confidence:** 3

**Summary:**

The research intends to discuss the question of how to balance semantic understanding and generation quality in discrete generative modeling. The authors identify a distinct performance trade-off between two existing paradigms: Masked Diffusion Language Models (MDLM), which excel at semantic understanding and zero-shot generalization, and Uniform-noise Diffusion Language Models (UDLM), which dominate in few-step generation quality. To address this gap, the paper proposes XDLM, a mixed diffusion framework that theoretically unifies MDLM and UDLM via a stationary noise kernel. By introducing a mixing ratio hyperparameter ($k$) and utilizing an algebraically simplified scalar formulation to alleviate memory bottlenecks, XDLM aims to strike an optimal balance. Through extensive ablations and experiments on text (OpenWebText, LM1B) and image (ImageNet-1K) datasets, the authors demonstrate that XDLM can achieve a sweet spot that maintains strong zero-shot understanding while significantly improving few-step generation quality.

**Compliance With Llm Reviewing Policy:**

Affirmed.

**Final Justification:**

4 weak accept

**Key Questions For Authors:**

1. Could the authors provide a more rigorous theoretical intuition or analysis explaining the fundamental reasons *why* MDLM excels at understanding while UDLM excels at few-step generation?
2. Given that the optimal $k$ appears to shift depending on the modality (text vs. image) and the application of Classifier-Free Guidance, how sensitive is XDLM to this hyperparameter in a real-world, generalized setting?
3. Is it feasible that a single multimodal model trained jointly on text and images with a fixed $k$ would still exhibit this "sweet spot" balance, or would the competing objectives cause performance degradation?

**Limitations:**

Yes. The authors have adequately discussed the limitations of their work in the Conclusion section (Section 6), explicitly acknowledging the lack of large-scale from-scratch pretraining, the unexplored multimodal joint training, and the need for optimized domain-specific sampling strategies.

**Strengths And Weaknesses:**

**Paper Strengths:**
1. **Novel and Principled Perspective:** The paper is the first to explicitly propose an architectural interpolation between masked and uniform-noise diffusion models to balance understanding and generation. The theoretical unification under a stationary noise kernel is innovative and provides a fresh angle for discrete diffusion research.
2. **Good Empirical Validation:** The experimental results support the authors' claims. The extensive ablation studies effectively isolate the impact of the mixing ratio $k$, and the performance crossover phenomenon is well-documented, showing that XDLM successfully breaks the Pareto frontier of existing baselines.

**Paper Weaknesses:**
1. **Domain-Specific Tuning and Lack of Unified Multimodal Evidence:** Overall, the authors analyze a notable aspect of diffusion models regarding the understanding-generation trade-off. However, the evaluation strategy treats text and images as entirely disjointed pipelines. The paper demonstrates that tuning $k$ yields an optimal "sweet spot" (e.g., $k=0.1$ for text, but often different for images depending on CFG scales, as seen in Appendix I). If the model requires task-specific or modality-specific tuning of $k$ to achieve state-of-the-art results on entirely separate models (one for OpenWebText, one for ImageNet), the motivation for a "unified" balancing architecture is somewhat weakened. If the goal is simply to achieve the best performance on a specific single-domain dataset, one might argue for sticking to extreme ends (pure MDLM or UDLM) tailored with specific tricks. The paper lacks a joint training experiment (e.g., a unified multimodal model) to prove that a single XDLM setting can intrinsically balance understanding and generation across distinct data modalities simultaneously.
2. **Lack of Deep Theoretical Insight into the "Why":** While the paper provides robust empirical evidence and theoretical formulations on how to combine MDLM and UDLM, it lacks a profound theoretical analysis of why this inductive bias exists in the first place. Why exactly does a pure masking mechanism natively favor semantic understanding (zero-shot likelihood), and why does uniform noise inherently favor few-step generation? A deeper analytical discussion on the gradient dynamics or information-theoretic properties of these two noise distributions would significantly elevate the insights provided to the community.

---

> ### Author Rebuttal · Authors · 2026-03-30
>
> ## To Reviewer 2HPG
>
> We thank Reviewer 2HPG for recognizing our "novel and principled perspective" and the innovation of our stationary noise kernel in breaking the Pareto frontier. We have addressed the insightful comments regarding theoretical depth and multimodal generalization in the responses below.
>
> ### W1 & K2 & K3: Domain-specific tuning, and robustness of k without a unified multimodal model.
>
> We agree that a jointly trained multi-modal model would be a compelling demonstration that a single shared $k$ can simultaneously balance understanding and generation. While we have not yet conducted joint multi-modal training, our separate text and image experiments already provide consistent cross-modal evidence. XDLM's balancing behavior generalizes across modalities rather than being an artifact of any particular setting.
>
> Across tasks and step budgets, MDLM and UDLM exhibit clear endpoint biases, where each performs well in the regime it is designed for but shows a pronounced weakness in the complementary aspect. XDLM, by contrast, consistently avoids these failure modes, maintaining strong performance in both understanding (zero-shot perplexity) and generation (across step budgets, including with classifier-free guidance). This supports our claim that $k$ serves as an effective balancing parameter and motivates future evaluation within a unified multi-modal model.
>
> We summarize representative results below (**drawn from Table10-15 in Appendix G/H/I/L**, **bold** = best, underline = worst). The endpoint methods tend to excel in only a subset of settings, whereas XDLM provides a more consistent trade-off across tasks with across modalities and step budgets.
>
> Lastly, we note that the absence of joint multi-modal training is already acknowledged in the `Limitations' section of our submission, as such experiments require computational resources beyond our current capacity.
>
> |Model|MDLM|GIDD|XDLM(k=1e-3)|XDLM(k=0.1)|XDLM(k=0.5)|XDLM(k=0.9)|UDLM|
> |---|---|---|---|---|---|---|---|
> |OWT-ZS-PPL|53.650|**53.384**|53.722|54.110|56.236|58.829|$\underline{59.574}$|
> ||
> |Owt-Gen-PPL @ [1024/1024]|41.545|**39.981**|42.797|52.609|66.136|78.238|$\underline{106.666}$|
> |LM1B-Gen-PPL @ [128/128]|$\underline{102.142}$|101.211|103.544|101.983|100.754|-|**96.385**|
> |ImageNet-FID @ [32/256]|18.928|24.588|25.637|22.187|**21.486**|-|$\underline{27.049}$|
> ||
> |Owt-Gen-PPL @ [64/1024]|$\underline{106.098}$|102.558|104.090|94.837|86.956|84.068|**76.611**|
> |LM1B-Gen-PPL @ [32/128]|$\underline{124.731}$|124.014|124.444|113.006|106.182|-|**102.072**|
> |ImageNet-FID @ [8/256]|47.732|$\underline{54.933}$|52.590|34.109|**29.362**|-|30.144|
> ||
> |ImageNet-CFG-FID @ [8/256]|11.144|$\underline{15.151}$|12.872|**8.956**|9.038|-|9.718|
>
>
> ### W2 & K1: Theoretical Intuition: Why Masking Favors Understanding and Uniform Noise Favors Generation
>
> While our work primarily focuses on the formal unification via a stationary noise kernel, we offer the following intuition for why the empirical gap between endpoints arises.
>
> In MDLM, corrupted positions are replaced by an explicit [MASK] token, making the corruption directly observable. The model knows exactly which tokens are unreliable and can condition on the remaining trusted context. This produces a training signal closer to conditional modeling, which favors understanding tasks and lower zero-shot perplexity.
>
> In UDLM, corruption is implicit: any observed token may be either clean or corrupted, leaving the entire input ambiguous. The model must learn to recover from sequences where many positions are potentially wrong simultaneously. This aligns naturally with fast parallel denoising, where each step refines many tokens jointly, and therefore tends to favor few-step generation.
>
> XDLM interpolates these two signals through the stationary distribution, avoiding over-specialization to either endpoint and achieving a more balanced trade-off across step budgets.

---

> > ### Author Rebuttal · Reviewer_2HPG · 2026-04-03
> >
> > Thank you for the detailed rebuttal. Overall, my concerns are resolved, and I am raising my score to support the acceptance of this work.

---

> > > ### Author Response · Authors · 2026-04-03
> > >
> > > Thank you very much for your thoughtful review and for the time you spent evaluating our rebuttal. We are glad to hear that our responses have addressed your concerns, and we greatly appreciate your support for the acceptance of this work.
> > >
> > > We noticed that the numerical score in the system does not yet appear to reflect your updated evaluation. We wanted to bring this to your attention in case it was an oversight or a system issue. If needed, we believe the score can be updated by editing the original review (via the 'Edit' button) and resubmitting the form.
> > >
> > > Thank you again for your valuable and constructive feedback throughout the review process.

---

### Official Review · Reviewer_G7xm · 2026-03-02

**Soundness:** 4
**Presentation:** 4
**Significance:** 3
**Originality:** 2
**Overall Recommendation:** 5
**Confidence:** 5

**Summary:**

This work presents a discrete diffusion model that combines uniform categorical and masked absorbing state noising processes. In doing so, the authors demonstrate that their model (XDLM) combines the strong modeling capabilities of masked diffusion models with the low sampling-regime performance and error correction capabilities of uniform noise diffusion models. Importantly, the authors derive a computationally efficient form for the diffusion posteriors which accelerates their training process relative to previous works that incorporate uniform categorical noise. XDLM is validated across a wide range of downstream applications. including standard language modeling, image generation, and fine-tuning of an 8B large diffusion language model.

**Compliance With Llm Reviewing Policy:**

Affirmed.

**Final Justification:**

This paper is well executed and presents important results for the community. During the rebuttal, the authors answered my outstanding questions. I therefore maintain my support for this work.

**Key Questions For Authors:**

- Why was Duo (Sahoo et al., 2025) not included as a baseline?
- Why in Lemma 4 is the limiting case denoted with an approximately equal sign ($\approx$)? Is the limit as $s \rightarrow t$ not a true equality?
- I think I misunderstood the last step of the proof of Lemma 4. Why does $r(\mathbf{z}_t)/f_t(\mathbf{x}, \mathbf{z}_t)$ evaluate to $1$?
- Can the authors explain in more detail what it is about the GIDD formulation that prevents it from using efficient training?

**Limitations:**

Yes

**Strengths And Weaknesses:**

### **Strengths**

- The paper is well written and easy to follow.
- The experimental setup is robust and well executed. The results support the authors’ main claims.
- Experimental details are well laid out, which bolsters the reproducibility of this work.

### **Weaknesses**

- I have a concern with the presentation of the combined mask + uniform kernel: This exact kernel was already defined in D3PM (Austin et al., 2021; see section 4 first paragraph and Appendix A.2.6 there). However, in the current manuscript, the kernel is presented as a contribution of the current work under review. While I agree that XDLM thoroughly investigates the efficacy of using this limiting distribution, I believe that the presentation in Section 3.1 should be reworded to better reflect the fact that this particular contribution is not from the current work.
- Additionally, I think the novelty of the scalar formulation for the ELBO is not properly stated. In particular, I believe the derived formula is similar (equivalent in the case of pure UDLM, k=1) to that derived in Duo (Sahoo et al., 2025; see equation 17 and Appendix A.6 there). Applying this same idea to the joint masked + uniform kernel is a useful aspect of this work, but I believe the novelty of this should be better presented and that Sahoo et al., 2025 should more prominently be cited for this section. Please correct me if I am wrong in interpreting this equivalence.
- The authors should include the sampling procedure for XDLM in the methods section. Section 3.2 is titled “Efficient Sampling and Training via Scalar Formulation”, but I do not believe the sampling formulation, nor how it is accelerated, are properly discussed in this section. This is especially true for LLaDA-type models where the adaptation of confidence-based sampling is non-trivial for models that incorporate uniform noise; this should be presented as part of the method.

**Minor typos / suggestions**
- GIDD contains a hyperparameter that controls “the amount” of uniform categorical noise used ($p_u$, using the notation of Von Rütte et al., 2025). The authors should explicitly note the value for this hyperparameter that they used for each of their GIDD baselines.
- Throughout, to help with readability, I would recommend bolding the best values in tables.
- Eq. 4 is missing $\lim_{T \rightarrow \infty}$ before the $T D_{\text{KL}}$ I believe.
- In Lemma 3.1 (Line 111) should $\mathcal{R}^N$ be $\mathbb{R}^N$ instead?
- In Eq. 8 I believe $K$ should be bolded as it represented the limiting kernel from Equation 7
- Line 578 in Appendix B contains typos and what appears to be a malformed reference: $\texttt{lemma}$
- I believe Equations 56-60 in the proof of Lemma 4 should still be preceded by $\lim_{s \rightarrow t}$

---

**References**

Austin, Jacob, et al. "Structured denoising diffusion models in discrete state-spaces." Advances in neural information processing systems 34 (2021): 17981-17993.

Sahoo, Subham Sekhar, et al. "The diffusion duality." arXiv preprint arXiv:2506.10892 (2025).

Von Rütte, Dimitri, et al. "Generalized interpolating discrete diffusion." arXiv preprint arXiv:2503.04482 (2025).

---

> ### Author Rebuttal · Authors · 2026-03-30
>
> # To Reviewer G7xm
>
> We thank Reviewer G7xm for carefully checking our derivations and for acknowledging XDLM’s logical structure and reproducibility. We address the concerns on novelty, sampling, and baselines below.
>
> ### W1: Contribution of the mixed kernel regarding D3PM
>
> Although the noise kernel formulation in `Equation 7` matches the `mask + uniform` form in D3PM, our contribution lies in showing that this choice is not heuristic: it arises as the unique stationary kernel satisfying the instantaneous mixing condition established in Lemma 3.1.
>
> We will revise the text near `Equation 7` to explicitly note the connection:
>
> > ...the resulting noise kernel matrix naturally interpolates between uniform noise and the mask state as introduced by ~\cite{d3pm}:
>
> ### W2: Distinction of scalar ELBO formulation from Duo
>
> Our main contribution (`Appendix B`) is a scalarized ELBO computation, fundamentally different from the `Rao-Blackwellized NELBO` estimator used by Duo, with clear gains in throughput and memory (`Table 17`).
> Since our UDLM baseline is implemented from Duo's codebase (`DUO_BASE`) and uses the same `Rao–Blackwellized NELBO`, the remaining performance gap directly isolates the effect of our scalar ELBO formulation.
>
> ### W3: Sampling formulation and LLaDA sampling
>
> The sampling pseudocode was not included in `Section 3` for brevity, as our sampler derives directly from the posterior transition. In practice, its implementation further avoids instantiating additional full-vocabulary tensors. We will add the complete pseudocode to the appendix and reference it explicitly in the main text.
>
> For LLaDA, sampling is indeed non-trivial; we have provided the full procedure in `Appendix J`, `Algorithm 1` and will add a corresponding reference in the main text.
>
> ### Experimental setting and minor fixes
>
> All baselines (MDLM, GIDD, UDLM, and XDLM) were implemented and retrained in the Duo codebase, using official components where available. UDLM uses Duo’s `Rao-Blackwellized NELBO`implementation; GIDD uses official defaults (loss_weighting='dynamic', min_loss_weight=0.0, max_loss_weight=2.0, p_uniform=0.0, t_eps=1e-4).
> We will also incorporate all suggested typo and notation fixes in the revision.
>
> ### K1: Why Duo is not in a primary baseline
>
> Our goal is to isolate the effect of stationary-kernel interpolation (masking vs uniform noise). Duo’s strongest setting additionally incorporates `curriculum learning`($^\dagger$) and `distillation`, which are orthogonal techniques and would confound this comparison. Nevertheless, to ensure a strong and fair UDLM baseline, we implement UDLM using Duo’s `Rao–Blackwellized NELBO` from the Duo codebase.
>
> Below we provide representative comparisons. We will also add a brief discussion in the related work section.
>
> ***Table 7: validation perplexity on owt***
>
> |Model|MDLM|GIDD|XDLM|UDLM|**Duo $^\dagger$**|
> |---|---|---|---|---|---|
> |PPL|23.321|23.136|24.097|25.937|25.2|
>
> ***Table 8: zeroshot perplexity***
>
> |Dataset|AG News|LAMBADA|LM1B-GPT2|PTB|ArXiv|PubMed|WikiText|Average|
> |---|---|---|---|---|---|---|---|---|
> |XDLM|62.768|45.608|68.229|90.796|37.232|41.391|32.748|54.110|
> |UDLM|69.402|51.272|75.572|95.986|42.671|47.181|34.933|59.574|
> |**Duo $^\dagger$**|67.81|49.78|73.86|89.35|40.39|44.48|33.57|57.03|
>
> ***Table 10: generative ppl on owt***
>
> |Model|step 8|step 16|step 32|step 64|step 128|step 256|step 512|step 1024|
> |---|---|---|---|---|---|---|---|---|
> |XDLM|301.640|161.724|114.795|94.837|83.205|73.824|63.888|52.609|
> |**Duo $^\dagger$**|198.27|122.78|96.19|85.62|80.02|78.62|78.14|77.69|
>
> ### K2–K3: Lemma 4 notation and last-step clarification
>
> We confirm $\approx$ in Lemma 4 was a typo: as $s\to t$ it should be an equality. We will correct this and ensure that limit notation is used consistently throughout.
>
> For the last step, we do not claim $\frac{r(\mathbf{z}_t)}{f_t(\mathbf{x},\mathbf{z}_t)}$ equals $1$ in isolation; it cancels with the preceding factor $\frac{f_s(\mathbf{x},\mathbf{z}_t) \alpha _{t|s}}{r(\mathbf{z}_t)}$ (`Equation 65`). We will add an explicit intermediate line in `Appendix B` to clarify this multiplication and the $s\to t$ limit.
>
> ### K4: Why GIDD cannot directly use XDLM’s efficient training
>
> GIDD employs a time-varying, score-based framework and cannot directly inherit XDLM's efficient training, for two reasons.
>
> ***Objective mismatch***. GIDD's continuous-time, score-based objective with a time-varying kernel leads to a more complex estimator involving state-dependent weighting and additional correction terms. Applying XDLM's per-token scalar reductions would require fundamentally restructuring this estimator.
>
> ***Stationarity incompatibility***. XDLM’s scalarization and simplification rely on the time-independent $r(\cdot)$, which enables key cancellations in the $s \to t$ limits. With GIDD’s time-varying kernel, the effective prior becomes time-dependent, $r_t(\cdot)$, breaking these cancellations and preventing the same simplification.

---

> > ### Author Rebuttal · Reviewer_G7xm · 2026-04-01
> >
> > Thank you for the detailed response. My questions and comments have been addressed. I maintain my positive assessment of this work.

---

> > > ### Author Response · Authors · 2026-04-03
> > >
> > > Thank you for your time and for the positive evaluation of our rebuttal. We are glad that our responses addressed your concerns and will incorporate your feedback into the final manuscript. We truly appreciate your constructive comments throughout this process.

---

### Official Review · Reviewer_h4jz · 2026-03-02

**Soundness:** 3
**Presentation:** 3
**Significance:** 4
**Originality:** 4
**Overall Recommendation:** 4
**Confidence:** 3

**Summary:**

This paper introduces mixed Diffusion Language Modeling, which balances the semantic understanding of Masked Diffusion Language Models (MDLM) with the few-step generation capabilities of Uniform-noise Diffusion Language Models (UDLM). The main contribution is a noise kernel that blends absorbing and uniform noise, controlled by k. The authors also propose an efficient formulation for both the posterior probabilities and the KL-divergence objective to reduce computational cost.

**Compliance With Llm Reviewing Policy:**

Affirmed.

**Final Justification:**

The rebuttal has addressed all of my concerns, so I will keep my positive score.

**Key Questions For Authors:**

1. Please see the weakness part.
2. Can authors provide some guidance for selecting k?

**Limitations:**

Yes

**Strengths And Weaknesses:**

Strengths
1. The motivation of this paper is clear, and the paper targets an important and practical trade-off..
2. The proposed kernel clearly interpolates between uniform and absorbing noise. The proofs demonstrates that XDLM can be transferred to the MDLM and UDLM at extreme settings.
3. The evaluation is relatively comprehensive, covering both language modeling and image generation.

Weaknesses
1. The choice of k will influence the performance, and the selection of it is empirical.
2. The generation results are evaluated only on automatic metrics like Perplexity and Token Entropy or FID and IS, it lacks human evaluation or LLM-as-a-judge metrics to evaluate the real quality.

---

> ### Author Rebuttal · Authors · 2026-03-30
>
> ## To Reviewer h4jz
>
> We thank Reviewer h4jz for their positive assessment of our "clear" motivation and the robustness of our theoretical proofs across both text and image generation.
>
> ### W1 & K2: Guidance of the Hyper-parameter $k$ Selection
>
> In XLDM, $k$ serves as a single, interpretable hyperparameter that controls the trade-off between the MDLM and UDLM extremes. A smaller $k$ makes XDLM more MDLM-like, favoring language understanding and lower zero-shot perplexity, while a larger $k$ makes it more UDLM-like, favoring few-step generation. In practice, $k$ can therefore be selected based on the target step budget and evaluation objective, rather than through extensive hyperparameter search.
>
> In the `Impact of Mixing Ratio` section and `Appendix. G/H/I/L`, we sweep $k\in\{10^{-3},0.1,0.5, 0.9\}$ on text (OWT, LM1B) and image (ImageNet, CIFAR-10) benchmarks. While the optimal $k$ shifts slightly across settings, the overall pattern is consistent: MDLM and UDLM each specialize at opposite ends of the step-budget spectrum, whereas XDLM maintains more balanced performance across both step regimes and domains.
>
> (**collected from Appendix. G/H/I/L Tables 10-15, bold = best, underline = worst**)
>
> |Model|MDLM|GIDD|XDLM(k=1e-3)|XDLM(k=0.1)|XDLM(k=0.5)|XDLM(k=0.9)|UDLM|
> |---|---|---|---|---|---|---|---|
> |OWT-ZS-PPL|53.650|**53.384**|53.722|54.110|56.236|58.829|$\underline{59.574}$|
> ||
> |Owt-Gen-PPL @ [1024/1024]|41.545|**39.981**|42.797|52.609|66.136|78.238|$\underline{106.666}$|
> |OWT-Gen-PPL-[512/1024]|51.756|49.822|52.650|63.888|73.503|78.707|73.345|
> |OWT-Gen-PPL-[256/1024]|64.376|62.077|65.245|73.824|77.155|79.456|74.019|
> |LM1B-Gen-PPL @ [128/128]|$\underline{102.142}$|101.211|103.544|101.983|100.754|-|**96.385**|
> |ImageNet-FID @ [32/256]|18.928|24.588|25.637|22.187|**21.486**|-|$\underline{27.049}$|
> |ImageNet-FID-[16/256]|28.785|35.403|35.046|25.774|23.417|-|26.242|
> ||
> |Owt-Gen-PPL @ [64/1024]|$\underline{106.098}$|102.558|104.090|94.837|86.956|84.068|**76.611**|
> |LM1B-Gen-PPL @ [32/128]|$\underline{124.731}$|124.014|124.444|113.006|106.182|-|**102.072**|
> |ImageNet-FID @ [8/256]|47.732|$\underline{54.933}$|52.590|34.109|**29.362**|-|30.144|
> ||
> |ImageNet-CFG-FID @ [8/256]|11.144|$\underline{15.151}$|12.872|**8.956**|9.038|-|9.718|
>
>
> ### W2: Human-evaluation and LLM-as-judge
>
> Following GIDD, we additionally evaluate unconditional generation quality on OWT. Specifically, we sample 5,000 outputs using 128 decoding steps and score them via the GPT-5.2 API on five dimensions, i.e., clarity, grammaticality, factuality, writing style, and creativity, each on a 1-10 scale.  The grader is instructed to provide a brief justification before assigning a score for each dimension, and to return all results in JSON format for automated parsing. We adopt the same evaluation prompt as GIDD. The results are summarized below:
>
> |OWT@128|clarity|gramaticality|facuality|style|creativity|
> |---|---|---|---|---|---|
> |MDLM|2.02|2.06|1.93|2.43|4.24|
> |GIDD|2.02|2.05|**1.94**|2.44|4.24|
> |XDLM|2.02|2.10|1.92|**2.48**|**4.37**|
> |UDLM|2.02|**2.14**|1.84|2.46|4.33|
>
> A rigorous human evaluation would require careful questionnaire design, a sufficiently large annotator pool, and systematic collection and aggregation of responses; we consider this a valuable direction and leave it for future work.

---

> > ### Author Rebuttal · Reviewer_h4jz · 2026-04-01
> >
> > Thank you for the rebuttal and the additional clarifications. The response has addressed my main concerns. I will keep my current positive score.

---

> > > ### Author Response · Authors · 2026-04-03
> > >
> > > We sincerely appreciate your thorough evaluation of our rebuttal. We are gratified that our clarifications successfully addressed your concerns. Your insights have been instrumental in refining the paper, and we will carefully incorporate your suggestions into the final version of the paper. We sincerely appreciate your support throughout the review process.

---

### Official Review · Reviewer_hJW5 · 2026-03-13

**Soundness:** 3
**Presentation:** 4
**Significance:** 4
**Originality:** 4
**Overall Recommendation:** 4
**Confidence:** 3

**Summary:**

This paper proposes the XDLM model, which attempts to unify the understanding and generation tasks in the form of Diffusion Language Models (DLM). Through detailed theorem and formula derivation analysis, it achieves the algebraic optimization of training objectives after utilizing stationary noise kernels. This approach achieves synergistic performance in both understanding and generation within a few steps, while further elucidating the steady-state dynamics and scaling characteristics observed during the model's training process.

**Compliance With Llm Reviewing Policy:**

Affirmed.

**Key Questions For Authors:**

Regarding the content raised in the “weaknesses” section, I am perplexed by the emphasis on discussing the model's various performances within a relatively small step range, which results in settings that produce lower-quality content. I would like to see comparisons and explanations of generation quality under normal steps (eg. 256 or 512 steps).

Additionally, I hope the authors will further explore the impact of different k values across various domains and vocabularies, conducting a more comprehensive sensitivity analysis. Otherwise, the current k selection remains akin to a magic number approach.

I also hope the authors provide further training results for the scaling feature. While the current training demonstrates impressive performance on MBPP, I anticipate further attribution from the authors, along with an explanation for why the original LLaDA exhibits performance degradation under these training conditions.

**Limitations:**

Most experiments were conducted in scenarios with few steps, resulting in a rather limited demonstration of XDLM's overall performance. The selection of k values raises issues regarding their general applicability.

**Strengths And Weaknesses:**

**Strength:**

The derivation logic for the loss function and training objective after introducing the stationary noise kernel is sound and well-reasoned. The derivation demonstrating that UDLM and MDLM represent limiting cases of a unified model further substantiates the model's inherent unification capability.

The core principle of XDLM is presented in a clear, understandable, and effective manner. It achieves Pareto-frontier performance in scenarios with fewer steps and enhances model throughput while reducing GPU memory consumption through simplification techniques, demonstrating the model's scaling potential to some extent.

**Weakness:**

1. Clearly, the distributions jointly determined by MDLM and UDLM across different datasets are inconsistent (e.g., text-based OWT and LM1B show different trends of MDLM and UDLM). How is the $\pi$ distribution corresponding to XDLM's k setting justified? More evidence is needed regarding k's stability across tasks/vocab sizes.

2. XDLM's performance gains appear concentrated in low-step scenarios. Does it retain advantages for multi-step generation (e.g., 256 steps or more)? The image quality in the appendix for fewer steps is also noticeably poorer. Is comparison at this scale necessary?

3. The scope of comparison is rather limited. Beyond contrasting with the strong adversarial model GIDD, comparisons with other models attempting to unify generation and understanding tasks should be added. Furthermore, can simple loss weighting alone meet the corresponding requirements? Or could more prominent flow matching achieve a similar effect?

---

> ### Author Rebuttal · Authors · 2026-03-30
>
> ## To Reviewer hJW5
>
> We thank the reviewer for the constructive feedback and for recognizing the sound derivation and Pareto trade-off of XDLM. We address the concerns below.
>
> ### W1 & K2: How $\pi$ corresponds to $k$; robustness of $k$
> In XDLM, the stationary distribution is explicitly $\pi = k\mathbf{u}+(1-k)\mathbf{m}$,
> where $\mathbf{u}\in\mathbb{R}^V$ is uniform and $\mathbf{m}\in\mathbb{R}^V$ is the one-hot distribution of $[\mathrm{MASK}]$. Thus, $k$ is an explicit mixing coefficient between MDLM-like masking bias and UDLM-like global stochasticity.
>
> For robustness, we sweep $k\in\{10^{-3},0.1,0.5,0.9\}$ on text (OWT, LM1B) and images (ImageNet, CIFAR-10) in the `Impact of Mixing Ratio` section and `Appendices G/H/I/L`, with different task and vocabulary size. The results are shown in `Tables 10-15`, whereas XDLM is more balanced within each modality across understanding and generation. Due to limited response space, we refer the reviewer to the representative results summarized in our response to **Reviewer h4jz** **W1 & K2**.
>
> ### W2 & K1: Multi-step evaluation; is few-step scale necessary?
>
> We report results across the full step spectrum, not only few-step, for the text (Figure 2; Tables 10-11), LLaDA tuning (Table 16), and images (Tables 12-15).
> On LM1B, we observe a consistent cross-over over training: the performance of XDLM improves relative to MDLM, and with sufficient training they can match or even surpass MDLM at high-step sampling (Figure 4).
>
> We focus on the few-step regime because diffusion language models are designed to generate multiple tokens per step (D3PM), so the low-step setting directly reflects their intended use case. This is also consistent with common practice in image generation, where few-step budgets (e.g., 8 steps) are widely adopted operating points (MaskGIT, Consistency Models, SDXL-Turbo).
>
> ### W3: Related models; simple loss weighting; flow matching
>
> ***Related mixing approaches.***
> Several prior works incorporate uniform noise into masking-based diffusion, spanning both non-distilled training (e.g., GIDD) and distillation-oriented approaches (e.g., ReMDM, DiDi-Instruct). While distillation can improve inference speed, it often comes at the cost of language understanding capability, as reflected by zero-shot perplexity. To illustrate this trade-off, we include the zero-shot perplexity of DiDi-Instruct for comparison in the following table.
>
> |Dataset|AG News|LAMBADA|LM1B-GPT2|PTB|ArXiv|PubMed|WikiText|Average|
> |---|---|---|---|---|---|---|---|---|
> |XDLM|62.768|45.608|68.229|90.796|37.232|41.391|32.748|54.110|
> |MDLM + DiDi-Instruct|78.36|53.62|80.58|107.03|45.35|47.56|35.20|63.96|
>
> ***Simple loss weighting.*** A weighted sum of MDLM and UDLM losses is heuristic and lacks a corresponding noise kernel. In contrast, XDLM is derived from an explicit stationary distribution, providing a principled foundation. This heuristic baseline (denoted as SimpMix) also underperforms XDLM across all step budgets on LM1B as shown below:
>
> |Model|step 4|step 8|step 16|step 32|step 64|step 128|
> |----|---|---|---|---|---|---|
> |Perplexity $\downarrow$|
> |XDLM(k=0.1)|246.134|155.487|123.129|113.006|105.509|101.983|
> |SimpMix(k=0.1)|324.830|207.235|165.450|150.812|143.689|137.545|
>
> ***Flow matching.*** Discrete flow matching (e.g., DFM) typically employs a cross-entropy objective that, as also noted in MDLM, is not weighted to form a proper ELBO on log-likelihood. Since our theoretical framework, comparisons, and efficiency claims are all grounded in likelihood-based evaluation, including our scalarized ELBO derivation, a fair incorporation of DFM would require re-deriving both the training objective and the evaluation protocol under a likelihood-aligned bound, rather than simply substituting the loss function. We leave this extension to future work.
>
> ### K3: Further LLaDA results; why LLaDA-MDLM degrades
> Full tuning details and results are in `Appendix J`. We attribute the performance degradation of LLaDA-MDLM primarily to data distribution mismatch: training is conducted on a small FineWeb-Edu subset under a limited compute budget, which may be insufficient for MDLM to adapt without losing prior capabilities. In contrast, XDLM remains robust under the same budget: at small steps it surpasses both the untuned LLaDA and the tuned LLaDA-MDLM, and on GSM8K it consistently outperforms both across 32$\rightarrow$256 steps. We report the results of GSM8K with LLaDA-XDLM from `Tables 16` below.
>
> |GSM8K|[32/256]|[64/256]|[128/256]|[256/256]|
> |---|---|---|---|---|
> |LLaDA|24.49|57.16|67.93|72.25|
> |LLaDA-MDLM|24.41|54.97|68.61|72.93|
> |LLaDA-XDLM|29.26|57.85|68.01|73.16|
>
> ```citations:
> D3PM: Austin et al., NeurIPS 2021.
> MaskGIT: Chang et al., CVPR 2022.
> Consistency Models: Song et al., ICML 2023.
> SDXL-Turbo: Surkov et al., NeurIPS 2025.
> DFM: Gat et al., NeurIPS 2024.
> MDLM: Sahoo et al., NeurIPS 2024.
> ReMDM: Wang et al., arXiv 2025.
> DiDi-Instruct: Zheng et al., arXiv 2025.
> ```

---

> > ### Author Rebuttal · Reviewer_hJW5 · 2026-04-02
> >
> > Thanks for providing detailed explanations on my questions, I will keep the positive view of this paper.

---

> > > ### Author Response · Authors · 2026-04-03
> > >
> > > Thank you for your thorough review and for the time and effort you devoted to evaluating our rebuttal. We are glad that our responses have helped address your concerns. We will carefully take your feedback into account as we prepare the final version of the paper. We sincerely appreciate your valuable and constructive comments throughout the review process.

---

### Decision · Program_Chairs · 2026-04-30

**Decision:**

Accept (spotlight)

**Comment:**

The paper proposes XDLM, a discrete diffusion language model that unifies understanding and generation by combining uniform and masked absorbing noise processes within a single framework, supported by theoretical derivations showing these as limiting cases under a stationary noise kernel. This method demonstrates strong performance while offering a principled training objective and unified formulation. As supported by all reviewers, the paper provides values with unified perspective, solid theoretical grounding, and effective experiment setting and empirical validation. Besides, reviewers also raise the questions about the stability and selection of key parameters across datasets and vocabularies, differences compared with related works in literature, lack of validation in multimodal joint training scenarios, and deeper analytical discussion. The author provides a detailed response in the rebuttal to clarify these questions or discuss the limitations in the paper. All reviewers’ concerns are addressed. Overall, this paper provides valuable insights to the community for discrete diffusion models. Thus, I recommend Accept.